# GraphSTAGE: Channel-Preserving Graph Neural Networks for Time Series Forecasting

## Abstract

Recent advancements in multivariate time series forecasting (MTSF) have increasingly focused on the core challenge of learning dependencies within sequences, specifically intra-series (temporal), inter-series (spatial), and cross-series dependencies. While extracting multiple types of dependencies can theoretically enhance the richness of learned correlations, it also increases computational complexity and may introduce additional noise. The trade-off between the variety of dependencies extracted and the potential interference has not yet been fully explored. To address this challenge, we propose GRAPHSTAGE, a purely graph neural network (GNN)-based model that decouples the learning of intra-series and inter-series dependencies. GRAPHSTAGE features a minimal architecture with a specially designed embedding and patching layer, along with the STAGE (Spatial-Temporal Aggregation Graph Encoder) blocks. Unlike channel-mixing approaches, GRAPH-STAGE is a channel-preserving method that maintains the shape of the input data throughout training, thereby avoiding the interference and noise typically caused by channel blending. Extensive experiments conducted on 13 real-world datasets demonstrate that our model achieves performance comparable to or surpassing state-of-the-art methods. Moreover, comparative experiments between our channel-preserving framework and channel-mixing designs show that excessive dependency extraction and channel blending can introduce noise and interference. As a purely GNN-based model, GRAPHSTAGE generates learnable graphs in both temporal and spatial dimensions, enabling the visualization of data periodicity and node correlations to enhance model interpretability.

**Resources**: `https://anonymous.4open.science/r/GraphSTAGE`

## 1 Introduction

Multivariate time series forecasting (MTSF) is pivotal in various domains such as traffic flow prediction and energy consumption forecasting. A key consideration in MTSF is effectively modeling the dependencies within the sequences—specifically the intra-series (temporal), inter-series (spatial), and potentially cross-series dependencies (Liu et al., 2024a), as shown in Figure 2. Capturing these dependencies is crucial for understanding the underlying spatial and temporal relationships in the data, which directly impacts the accuracy of predictions.

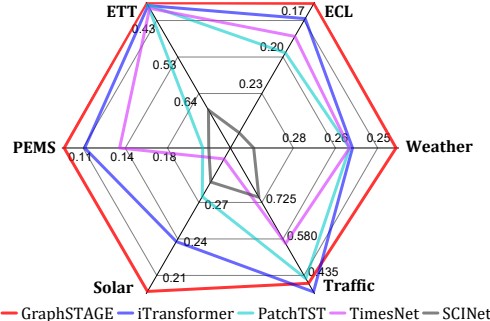

Figure 1: Performance of GRAPHSTAGE on average results (MSE).

However, many existing models focus on only one type of dependency. Common approaches employ channel-mixing techniques that project the original time series data $X_{\text{in}} \in \mathbb{R}^{N \times T}$ (where $N$ is the number of nodes and $T$ is the length of time series) into different representations. For instance, some methods transform $X_{\text{in}}$ into $H_S \in \mathbb{R}^{N \times D}$ (Liu et al., 2024c), which captures spatial dependencies among nodes, while others project it into $H_T \in \mathbb{R}^{T \times D}$ (Zhou et al., 2022; Li et al., 2021; Wu et al., 2021), emphasizing on temporal dependencies across time steps. These transformations

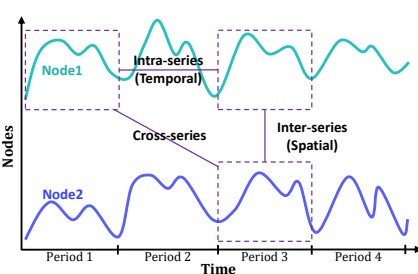

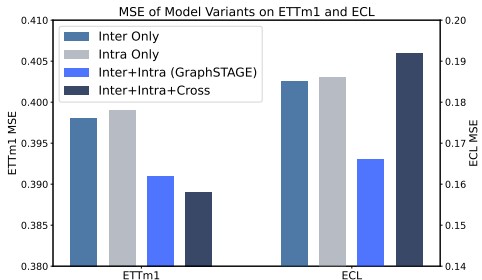

Figure 2: Dependencies between two sub-series in a multivariate time series.

Figure 3: Performance of GRAPHSTAGE Variants on ETTm1 and ECL Datasets.

often overlook at least one kind of dependency and fail to learn the underlying spatial or temporal graph structures (Yu et al., 2024), limiting the models' ability to extract inter-series or intra-series correlations effectively.

Recent models such as UniTST (Liu et al., 2024a) and FourierGNN (Yi et al., 2024) attempt to capture multiple types of dependencies, including cross-series dependencies, by blending the temporal and spatial dimensions. They reshape the input data $X_{\text{in}}$ from $\mathbb{R}^{N \times T}$ into a $\mathbb{R}^{NT \times 1}$ structure. While this approach theoretically allows for the simultaneous modeling of all dependencies, it also presents two significant challenges: (1) increased computational complexity and (2) a heightened risk of introducing additional noise.

First, mixing the channels may increases computational complexity. The complexity of weight multiplication operations escalates from $O(N^2)$ to $O((NT)^2)$ (Liu et al., 2024a; Yi et al., 2024), leading to exponentially higher computational costs. Consequently, these models often implement some compression mechanisms, such as router mechanism (Zhang & Yan, 2023), to mitigate the computational burden. Despite these efforts, a trade-off between model size and performance persists. Achieving better performance frequently requires larger models, indicating that compression techniques may not fully address the efficiency concerns. To further illustrate this point, we conducted model variants experiments in Section 4.3. As shown in Table 4, our model outperforms VarC — a channel-mixing model similar to UniTST (Liu et al., 2024a) and FourierGNN (Yi et al., 2024), as depicted in Figure 7, while also reducing memory usage by 83%.

Second, while blending channels allows these models to account for cross-series dependencies, it may introduce additional noise into the modeling process. Existing studies have often emphasized the benefits of capturing cross-series dependencies without fully considering the potential downsides of added noise. As shown in Figure 3, aggregating all dependencies may enhance predictive accuracy to some extent (as demonstrated by the improvement of performance on the ETTm1 dataset). However, it can also lead to overly complex models that struggle to compensate for the interference caused by the introduced noise, resulting in a sharp reduction in performance on the ECL dataset. This raises a crucial question: *Is it truly necessary to model all these dependencies?*

We argue that modeling either a single type of dependency or multiple dependencies in a coupled manner is inefficient. Recently, channel-preserving approaches have demonstrated efficiency and effectiveness (Liu et al., 2024b; Wang et al., 2024). To address the challenges of computational inefficiency and noise introduced by channel-mixing, we propose GRAPHSTAGE, a purely GNN-based model that decouples the learning of inter-series and intra-series dependencies while preserving the original channel structures. Unlike existing channel-mixing approaches, GRAPHSTAGE maintains the shape of the input data throughout the training process, thereby avoiding the interference caused by channel blending. To our knowledge, GRAPHSTAGE is the first purely graph-based, channel-preserving model. This design not only enhances computational efficiency but also reduces the noise associated with channel blending. Our contributions are threefold:

- We reflect on the extraction of dependencies in current time series models and emphasize that existing methods tend to overlook certain dependencies. Furthermore, we highlight that channel blending and excessive correlation extraction can introduce noise, and propose a channel-preserving framework to enable more accurate and robust dependencies modeling.

- We propose GRAPHSTAGE, a fully GNN-based method to effectively capture intra-series and inter-series dependencies, respectively, while generating interpretable correlation graphs. Moreover, its decoupled design allows for the independent extraction of specific dependencies as required.
- Experimentally, despite GRAPHSTAGE is structurally simple, it performs comparably to or surpasses state-of-the-art models across 13 MTSF benchmark datasets, as shown in Figure 1. Notably, GRAPHSTAGE ranks top-1 among 8 advanced models in 22 out of 30 comparisons, with results averaged across various prediction lengths.

By preserving the original data channels and decoupling dependencies learning, GRAPHSTAGE overcomes the key limitations of existing methods, providing a more efficient and interpretable solution for MTSF.

## 2    RELATED WORKS

**Single Dependency Modeling.**    Traditional multivariate time series forecasting methods often focus on capturing a single type of dependency—either temporal (intra-series) or spatial (inter-series). Deep learning models such as CNNs, RNNs, GRUs and Formers (Hochreiter, 1997; Chung et al., 2014; Rangapuram et al., 2018; Wu et al., 2021; Li et al., 2021; Zhou et al., 2022; Liu et al., 2021; Zhang et al., 2024) excel at modeling sequential data by capturing temporal dynamics within each series. However, these models typically treat each spatial node independently, failing to account for inter-series dependency. On the other hand, models that focus solely on inter-series dependency, such as GNNs (Bai et al., 2020) and Formers (Kitaev et al., 2020; Liu et al., 2024c; Cai et al., 2024), while effective at capturing spatial correlations, may not adequately model the temporal correlations within each series. Consequently, methods that concentrate on one type of dependency may fail to fully capture the complex correlations inherent in multivariate time series data.

**Modeling Combined Dependencies.**    To address the limitations of single-dependency extracting models, several GNNs (Kipf et al., 2018; Wu et al., 2019; 2020; Shang et al., 2021; Xu et al., 2023) have attempted to extract dependencies in both the temporal and spatial domains. However, these models often ignore global information extraction in either the spatial or temporal domain, focusing instead on local neighborhood information. Recent approaches have explored to capture multiple types of dependencies simultaneously by blending the temporal and spatial dimensions. FourierGNN (Yi et al., 2024) and UniTST (Liu et al., 2024a) construct hypervariate graph as input embeddings to represent time series with a unified view of spatial and temporal dynamics but overlook the potential interference caused by channel-mixing. Recognizing this issue, DGCformer (Liu et al., 2024b) identifies irrelevant nodes in channel-mixing and adopts a grouping mechanism to focus attention on relevant nodes. Crossformer (Zhang & Yan, 2023) and CARD (Wang et al., 2024) propose a two-stage framework to extract inter-series and intra-series dependencies, applying attention across both dimensions and then fuses the results. Building on these insights, we propose GRAPHSTAGE, a purely GNN-based model that decouples the learning of inter-series and intra-series dependencies while preserving the original input channels to avoid the interference introduced by channel blending.

## 3    GRAPHSTAGE

**Problem Definition.**    Given the historical data $\mathbf{X} = \{\mathbf{x}_1, \ldots, \mathbf{x}_T\} \in \mathbb{R}^{N \times T}$ with $N$ nodes and $T$ time steps, the multivariate time series forecasting task is to predict the future $K$ time steps $\mathbf{Y} = \{\mathbf{x}_{T+1}, \ldots, \mathbf{x}_{T+K}\} \in \mathbb{R}^{N \times K}$. This process can be given by:

$$\hat{\mathbf{Y}} = F_\theta(\mathbf{X}) = F_{\theta_t, \theta_s}(\mathbf{X}), \tag{1}$$

where $\hat{\mathbf{Y}}$ are the predictions corresponding to the ground truth $\mathbf{Y}$. The forecasting function is denoted as $F_\theta$ parameterized by $\theta$. In practice, the channel-preserving model will be decoupled leverage a temporal network (parameterized by $\theta_t$) to learn the intra-series dependency and a spatial network (parameterized by $\theta_s$) to learn the inter-series dependency, respectively (Wang et al., 2024).

**Overall Structure.**    Based on the motivation of using channel-preserving strategy to avoid interference introduced by channel-mixing, we propose GRAPHSTAGE—a purely GNN-based model with

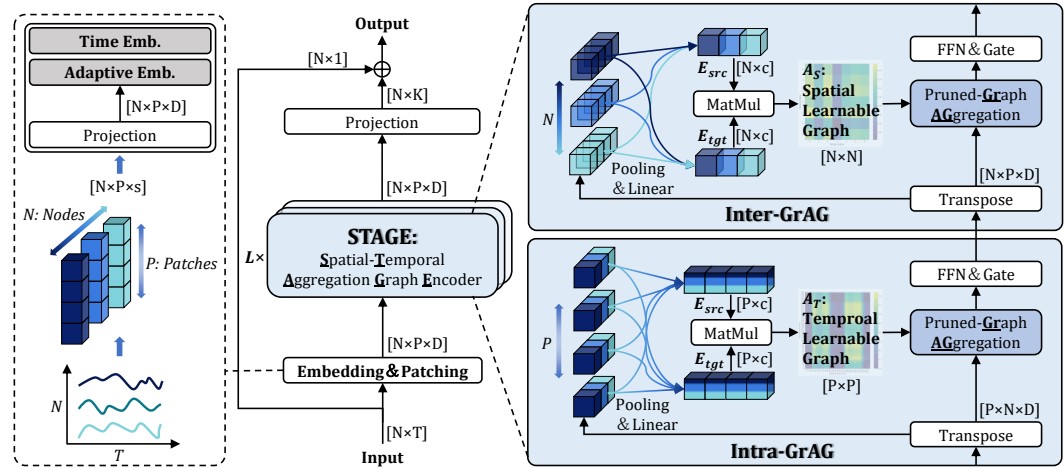

Figure 4: Overall Structure of GRAPHSTAGE. The model is composed of an Embedding & Patching layer followed by $L$ stacked STAGE blocks. Each STAGE block employs a decoupled yet unified architecture integrating two key modules: the Intra-GrAG (Intra-series Pruned-Graph Aggregation), which captures temporal dependency and generates the temporal learnable graph $A_T$; the Inter-GrAG (Inter-series Pruned-Graph Aggregation), which captures spatial dependency and generates the spatial learnable graph $A_S$. The pseudo-code of GRAPHSTAGE can be found in Algorithm 1.

an architecture that decouples the learning of intra-series and inter-series dependencies, as illustrated in Figure 4. Our model comprises two key components: (1) a specially designed embedding and patching layer; and (2) the Spatial-Temporal Aggregation Graph Encoder (STAGE) block. In the embedding and patching layer, we introduce a more fine-grained time embedding to fully utilize the relative positions of data points within an hour as prior knowledge. In the STAGE block, we design a decoupled framework to respectively extract temporal and spatial dependencies, with corresponding learnable graphs that can be visualized to enhance interpretability.

## 3.1 TOKENIZATION VIA EMBEDDING AND PATCHING

**Channel-preserving Embedding Strategy.** Most signal intra-series dependency modeling models regard multiple nodes of the same time as the (temporal) token. As a result, they project the input data shaped as $X_{\text{in}} \in \mathbb{R}^{N \times T}$ into $\mathbb{R}^{T \times D}$, where $D$ is the hidden dimension, and the original spatial dimension $N$ is not preserved. Inspired by inter-series oriented models (Liu et al., 2024c) in MTSF, we preserve the nodes dimension throughout the model, which proven competent by previous works (Cai et al., 2024). Given a time series with $N$ nodes, $X \in \mathbb{R}^{N \times T}$, we divide each univariate time series $x_i$ into patches $x_p^i \in \mathbb{R}^{P \times s}$, with stride $s$ and number of patches $P$ (Nie et al., 2023). A projection layer is then applied to map all the series into $X_p \in \mathbb{R}^{N \times P \times D}$, where $D$ is the embedding dimension.

**Refined Time Embedding to Enhance Relative Positioning.** The effectiveness of static covariates that are available in advance has been validated in several MTSF models (Lim et al., 2021; Jiang et al., 2023; Huang & Xiao, 2024). However, for datasets with a fixed sampling frequency below one hour (e.g., five minutes or fifteen minutes), previous models only embedded the 'Hour of Day' and 'Day of Week' information (Cai et al., 2024), which is insufficient to reflect the relative position within an hour. To address this limitation, we modify existing embedding methods by replacing the 'Hour of Day' embedding with a 'Timestamp of Day' embedding. This allows the embedding layer to adapt to the sample frequency, providing a more fine-grained time embedding that fully utilize the relative positions of data points within an hour as prior knowledge. Additionally, we introduce an learnable embedding to adaptively capture underlying dependencies. The process is presented below:

$$H = \text{Embedding}(X_p) = X_p + \mathbf{e}_{tod} + \mathbf{e}_{dow} + \mathbf{e}_{adp}[1], \tag{2}$$

---

[1]The process utilizes the broadcasting mechanism in PyTorch.

where $H \in \mathbb{R}^{N \times P \times D}$ contains $N$ embedded tokens of dimension $D$, $\mathbf{e}_{tod} \in \mathbb{R}^{P \times D}$ and $\mathbf{e}_{dow} \in \mathbb{R}^{P \times D}$ are learnable embeddings for 'Timestamp of Day' and 'Day of Week', respectively. $\mathbf{e}_{adp} \in \mathbb{R}^{P \times D}$ is generated using a random tensor method.

## 3.2 SPATIAL-TEMPORAL AGGREGATION GRAPH ENCODER

Our proposed STAGE block is illustrated in Figure 4. STAGE employs a decoupled yet unified architecture to aggregate information learned by Temporal Learnable Graph ($A_T$) and Spatial Learnable Graph ($A_S$). The Intra-series Pruned-Graph AGgregation module (Intra-GrAG) is responsible for extracting intra-series (temporal) dependencies and generating the $A_T$. Similarly, the Inter-series Pruned-Graph AGgregation module (Inter-GrAG) extracts inter-series (spatial) dependencies and generates the $A_S$.

**Decoupled Spatial-Temporal Extraction with Unified Aggregation.** STAGE is capable of learning intra-series and inter-series dependencies separately within a single block by utilizing a decoupled architecture composed of Intra-GrAG and Inter-GrAG modules. In STAGE block, the input tensor has dimensions $H \in \mathbb{R}^{N \times P \times D}$, where $N$ is the number of nodes, $P$ is the number of patches, and $D$ is the embedding dimension. To learn intra-series dependencies, we first transpose the input tensor to shape $\mathbb{R}^{P \times N \times D}$, swapping the spatial and temporal dimensions. This restructure allows the model to focus on temporal relationships within each node across different time steps. After learning the intra-dependencies, we transpose the tensor back to its original shape $\mathbb{R}^{N \times P \times D}$ to learn inter-series dependencies, concentrating on the relationships between different nodes at each time step. By adopting this approach, we can employ a unified architecture for both intra-dependency and inter-dependency learning, simply by changing the order of the input dimensions.

Furthermore, since STAGE is a purely GNN-based method, the correlations among nodes or patches (time steps) learned by the model can be directly visualized, enhancing interpretability and providing insights into the data periodicity and node correlations.

**Learnable Graph Generator for Temporal and Spatial Dimensions.** Learnable Graphs are essential for characterizing both temporal and spatial similarities. STAGE adaptively learns the graph structures by generating separate adjacency matrices: $A_T$ for patches (temporal dimension) and $A_S$ for nodes (spatial dimension).

Since STAGE employs a unified aggregation mechanism, the principles of the Inter-GrAG and Intra-GrAG modules are analogous. Therefore, to avoid redundancy, the subsequent discussion will focus only on the components of the Inter-GrAG module. First, a Pooling layer downsamples the extracted temporal information. We can choose any pooling mechanisms in the temporal dimension as the $\mathrm{Pool}$ operation, such as max-pooling and mean-pooling. To capture directed similarities among nodes, we apply two Linear mappings to each node:

$$E_{src} = \mathrm{L2Norm}(H_{pool}W_{p1}), E_{tgt} = \mathrm{L2Norm}(H_{pool}W_{p2}), H_{pool} = \mathrm{Pool}(\mathrm{H_{in}}), \qquad (3)$$

where $H_{pool} \in \mathbb{R}^{N \times D}$. Here, $H_{in} \in \mathbb{R}^{N \times P \times D}$ is obtained by transposing the output of intra-GrAG module, which originally has the shape $\mathbb{R}^{P \times N \times D}$. $W_{p1} \in \mathbb{R}^{D \times c}, W_{p2} \in \mathbb{R}^{D \times c}$ are two trainable matrices, and $E_{src} \in \mathbb{R}^{N \times c}$ and $E_{tgt} \in \mathbb{R}^{N \times c}$ are the source and target embedding matrices of all nodes, respectively. The L2 normalization ensures that each embedding matrices has a unit norm, facilitating stable training and enhancing model performance.

The directed similarities between each pair of nodes can be extracted as follows (Wu et al., 2020):

$$A_S = \mathrm{SoftMax}(\mathrm{ReLU}(E_{src} \cdot E_{tgt}^T)). \qquad (4)$$

The $\mathrm{ReLU}$ activation is used to avoid negative values. $\mathrm{SoftMax}$ function is employed to normalize values in the matrix. In this way, we obtain the spatial learnable graph $A_S \in \mathbb{R}^{N \times N}$, which serves as a global similarities matrix. It should be noted that the parameters of this similarity matrix are derived for each individual sample. Consequently, when the sample changes, the similarity weights among different nodes also change.

**Pruned-Graph Aggregation Mechanism.** In the Intra-GrAG module, this mechanism performs graph convolutions on the learned graph $A_T$. In the Inter-GrAG module, it performs graph convolutions on the learned graph $A_S$, aggregating information from global nodes while pruning irrelevant or

weak connections. The pruning operation reduces noise and enhances the model's ability to focus on the most significant correlations. To avoid redundancy and for simplicity, the subsequent discussion will focus only on the components of the Inter-GrAG module.

Graph attention network (GAT) (Velickovic et al., 2017) is a powerful model for extracting spatial dependencies, allocating different weights to neighbor nodes. Pruned-Graph Aggregation (PGA) can be regarded as a Special GAT with three specific improvements: 1) input embeddings are the extracted temporal embeddings rather than the original features; 2) the input nodes learnable graph will be pruned to make the model concentrate on the most significant connections; 3) the spatial dependencies among nodes is global rather than localized in neighborhoods. In this way, PGA incorporates spatial information effectively and aggregates global information without any prior knowledge, such as pre-defined static graph. The whole process can be formulated as below:

$$H_{ag} = H_{in}W_1 + \text{Prune}(A_S)H_{in}W_2 + \text{Prune}(A_S)^T H_{in}W_3, \tag{5}$$

where $W_1, W_2, W_3 \in \mathbb{R}^{D \times D}$ are trainable matrices and $H_{ag} \in \mathbb{R}^{N \times P \times D}$. The $\text{Prune}$ operation retains the top-$k$ values to focus on the most significant connections, where $k = N \times \alpha$ for Inter-GrAG module and $k = P \times \alpha$ for Intra-GrAG module, with a coefficient $\alpha$ between 0 and 1 (e.g., 0.7). After that, a Feed-Forward Network (FFN) and $\text{Gate}$ is employed to obtain the output of Encoders $H_E$. The FFN processes the aggregated features to capture nonlinear transformations, while the gating mechanism controls the flow of information. This gating enhances the model's capacity to capture complex dependencies by adaptively weighing the importance of different features. The detailed implement about the FFN and $\text{Gate}$ layer can be found in Appendix A.

In summary, STAGE decouples intra-series and inter-series dependencies within a unified pruned-graph aggregation mechanism, avoiding computational overhead and potential noise introduced by channel blending. Its fully graph-based mechanism enhances interpretability. Further discussion about the variants of STAGE will be delivered in the Section 4.3.

## 4 EXPERIMENTS

### 4.1 EXPERIMENTAL SETUP

**Datasets.** To validate the performance of GRAPHSTAGE, we conduct extensive benchmarks on 13 real-world datasets, including ETT (4 subsets), ECL, Exchange, Traffic, Weather, Solar-Energy datasets proposed in LSTNet (Lai et al., 2018a), and PEMS (4 subsets) collected by the Performance Measurement System (PeMS) (Choe et al., 2002) and proposed in ASTGCN (Guo et al., 2019). Detailed dataset descriptions are provided in Appendix B, and the hyperparameters and settings can be found in Appendix C.

**Baselines.** We have selected seven well-known forecasting models as our benchmarks, including (1) Transformer-based methods: iTransformer (Liu et al., 2024c), Crossformer (Zhang & Yan, 2023), PatchTST (Nie et al., 2023); (2) Linear-based methods: DLinear (Zeng et al., 2023), RLinear (Li et al., 2023); and (3) TCN-based methods: SCINet (Liu et al., 2022), TimesNet (Wu et al., 2023). Additional comparisons with four advanced GNNs are provided in Table 9 of Appendix D.

### 4.2 MAIN RESULTS

**Outstanding Performance of GRAPHSTAGE Across 13 Datasets: Ranking First in 22 out of 30 Comparisons.** Comprehensive forecasting results are presented in Table 1, with the best performances in **red** and the second in blue. Full forecasting results are provided in Appendix D. Lower MSE/MAE values indicate better prediction performance. The quantitative results reveal that GRAPHSTAGE demonstrates outstanding performance across all datasets, including node-based multivariate time series datasets (e.g., PEMS, Solar-Energy) and attribute-based multivariate time series datasets (e.g., ETT, Weather, ECL). GRAPHSTAGE achieves the best performance in 22 out of 30 cases, significantly outperforming the recent state-of-the-art (SOTA) iTransformer, which ranks first in only 4 instances. Compared to iTransformer, the MSE on the ECL, ETT (AVG), Weather, Solar-Energy, and PEMS (AVG) datasets is significantly reduced by 6.7%, 2.1%, 5.8%, 17.6%, and 14.3%, respectively. Specifically, on the PEMS07 dataset, which has the largest number of nodes, GRAPHSTAGE outperforms the recent SOTA iTransformer by 20.8%, indicating its potential

Table 1: Multivariate forecasting results with prediction lengths $K \in \{12, 24, 48, 96\}$ for PEMS and $K \in \{96, 192, 336, 720\}$ for others and fixed lookback length $T = 96$. Results are averaged from all prediction lengths. *AVG* means further averaged by subsets. Full results are listed in Appendix D.

| Models | Ours | | iTransformer | | RLinear | | PatchTST | | Crossformer | | TimesNet | | DLinear | | SCINet | |
|---|---|---|---|---|---|---|---|---|---|---|---|---|---|---|---|---|
| Metric | MSE | MAE | MSE | MAE | MSE | MAE | MSE | MAE | MSE | MAE | MSE | MAE | MSE | MAE | MSE | MAE |
| ECL | **0.166** | **0.263** | 0.178 | 0.270 | 0.219 | 0.298 | 0.205 | 0.290 | 0.244 | 0.334 | 0.192 | 0.295 | 0.212 | 0.300 | 0.268 | 0.365 |
| ETTm1 | 0.391 | **0.394** | 0.407 | 0.410 | 0.414 | 0.407 | **0.387** | 0.400 | 0.513 | 0.496 | 0.400 | 0.406 | 0.403 | 0.407 | 0.485 | 0.481 |
| ETTm2 | **0.278** | **0.325** | 0.288 | 0.332 | 0.286 | 0.327 | 0.281 | 0.326 | 0.757 | 0.610 | 0.291 | 0.333 | 0.350 | 0.401 | 0.571 | 0.537 |
| ETTh1 | **0.445** | **0.430** | 0.454 | 0.447 | 0.446 | 0.434 | 0.469 | 0.454 | 0.529 | 0.522 | 0.458 | 0.450 | 0.456 | 0.452 | 0.747 | 0.647 |
| ETTh2 | 0.387 | 0.407 | 0.383 | 0.407 | **0.374** | **0.398** | 0.387 | 0.407 | 0.942 | 0.684 | 0.414 | 0.427 | 0.559 | 0.515 | 0.954 | 0.723 |
| ETT (AVG) | **0.375** | **0.388** | 0.383 | 0.399 | 0.380 | 0.392 | 0.381 | 0.397 | 0.685 | 0.578 | 0.391 | 0.404 | 0.442 | 0.444 | 0.689 | 0.597 |
| Exchange | 0.376 | 0.409 | 0.360 | **0.403** | 0.378 | 0.417 | 0.367 | 0.404 | 0.940 | 0.707 | 0.416 | 0.443 | **0.354** | 0.414 | 0.750 | 0.626 |
| Traffic | 0.462 | 0.294 | **0.428** | **0.282** | 0.626 | 0.378 | 0.481 | 0.304 | 0.550 | 0.304 | 0.620 | 0.336 | 0.625 | 0.383 | 0.804 | 0.509 |
| Weather | **0.243** | **0.274** | 0.258 | 0.278 | 0.272 | 0.291 | 0.259 | 0.281 | 0.259 | 0.315 | 0.259 | 0.287 | 0.265 | 0.317 | 0.292 | 0.363 |
| Solar-Energy | **0.192** | 0.267 | 0.233 | **0.262** | 0.369 | 0.356 | 0.270 | 0.307 | 0.641 | 0.639 | 0.301 | 0.319 | 0.330 | 0.401 | 0.282 | 0.375 |
| PEMS03 | **0.097** | **0.210** | 0.113 | 0.221 | 0.495 | 0.472 | 0.180 | 0.291 | 0.169 | 0.281 | 0.147 | 0.248 | 0.278 | 0.375 | 0.114 | 0.224 |
| PEMS04 | **0.090** | **0.200** | 0.111 | 0.221 | 0.526 | 0.491 | 0.195 | 0.307 | 0.209 | 0.314 | 0.129 | 0.241 | 0.295 | 0.388 | 0.092 | 0.202 |
| PEMS07 | **0.080** | **0.179** | 0.101 | 0.204 | 0.504 | 0.478 | 0.211 | 0.303 | 0.235 | 0.315 | 0.124 | 0.225 | 0.329 | 0.395 | 0.119 | 0.234 |
| PEMS08 | **0.139** | **0.220** | 0.150 | 0.226 | 0.529 | 0.487 | 0.280 | 0.321 | 0.268 | 0.307 | 0.193 | 0.271 | 0.379 | 0.416 | 0.158 | 0.244 |
| PEMS (AVG) | **0.102** | **0.203** | 0.119 | 0.218 | 0.514 | 0.482 | 0.217 | 0.305 | 0.220 | 0.304 | 0.148 | 0.246 | 0.320 | 0.394 | 0.121 | 0.222 |
| $1^{st}$ **Count** | **22** | | 4 | | 2 | | 1 | | 0 | | 0 | | 1 | | 0 | |

for application to larger-scale MTSF tasks, such as extensive grid management. Moreover, the recent SOTA iTransformer performs poorly on attribute-based multivariate time series datasets (e.g., ETT) because it is a single-dependency learning model that focuses solely on inter-series (spatial) dependencies. In attribute-based datasets, there is generally no strong direct interaction or correlation between the attributes (e.g., temperature, wind speed), which makes it more necessary to extract intra-series (temporal) dependencies. This observation further validates the effectiveness of GRAPHSTAGE in capturing both intra-series and inter-series dependencies, leading to superior forecasting accuracy across diverse types of multivariate time series data.

**Model Efficiency and Increasing lookback length.** We conducted a comprehensive comparison of the performance, training speed, and memory usage of GRAPHSTAGE against other models on the ECL dataset, as shown in Figure 5. While GRAPHSTAGE may not achieve the best results in terms of training speed and memory usage, it delivers the best predictive performance. To

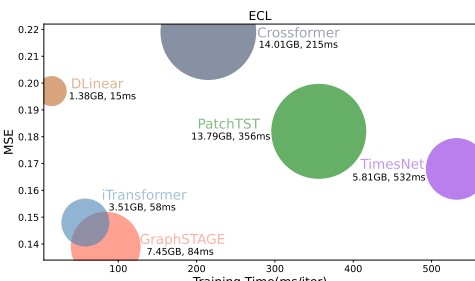

Figure 5: Model efficiency comparison on ECL dataset with input length 96 and output length 96.

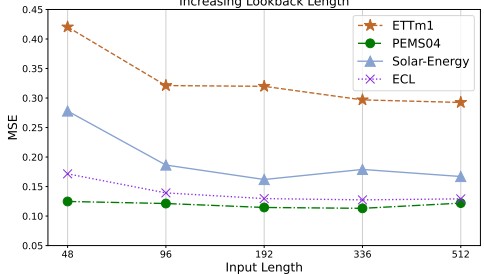

Figure 6: Forecasting results with output length 96 and input length in {48, 96, 192, 336, 512} across four datasets.

ensure a fair comparison, we followed the settings in (Cai et al., 2024) and set the batch size of GRAPHSTAGE to 32. Compared with Crossformer (Zhang & Yan, 2023), the only baseline model that learns multiple dependencies, GRAPHSTAGE's memory usage decreased by 47.0%, training time decreased by 60.9%, and predictive performance improved by 36.5%. This significant reduction in computational resources, combined with an improvement in accuracy, highlights GRAPHSTAGE's efficiency. Therefore, GRAPHSTAGE effectively balances model size, computational speed, and predictive accuracy. Our model achieves superior performance at an acceptable computational cost, demonstrating its practicality for real-world MTSF tasks.

Additionally, to evaluate the ability of GRAPHSTAGE to leverage increasing lookback length, we conducted experiments on the ETTm1, PEMS04, Solar-Energy, and ECL datasets. The input lengths were varied from shorter to longer as 48, 96, 192, 336, 512, while the forecasting horizon was fixed at the next 96 time steps. As shown in Figure 6, the model's performance steadily improves as the input length increases. Notably, when the input length expands from 48 to 96, the MSE decreases most significantly. This demonstrates that the *Intra-GrAG* module of GRAPHSTAGE effectively captures intra-series dependencies, enabling it to learn more temporal correlations from longer input series.

## 4.3 MODEL ANALYSIS

**Ablation on Correlation Learning Mechanism.** To verify the effectiveness of GRAPHSTAGE components, we provide detailed ablation studies covering both removing components (w/o) and replacing components (Replace) experiments. The averaged results are listed in Table 2. In the replacement experiments, we use the attention from Crossformer (Zhang & Yan, 2023), which has been proved more accurate than vanilla Transformer (Vaswani et al., 2017). Removing any component from GRAPHSTAGE results in performance degradation. GRAPHSTAGE utilizes Inter-GrAG module on the spatial dimension and Intra-GrAG module on the time dimension, generally achieving better performance than when replaced by the specially designed attention from Crossformer.

Table 2: Ablations on the Correlation Learning Mechanism. We remove or replace components along spatial and temporal dimensions to learn multivariate correlations. The average results of all predicted lengths are listed here, with full results provided in Appendix G.

| Design | Spatial | Temporal | ETTm1 | | ECL | | Traffic | | Solar-Energy | |
|---|---|---|---|---|---|---|---|---|---|---|
| | | | MSE | MAE | MSE | MAE | MSE | MAE | MSE | MAE |
| GRAPHSTAGE | Inter-GrAG | Intra-GrAG | **0.391** | **0.394** | **0.166** | **0.263** | 0.462 | **0.294** | **0.192** | 0.267 |
| w/o | Inter-GrAG | w/o | 0.398 | 0.400 | 0.185 | 0.277 | 0.478 | 0.312 | 0.225 | 0.292 |
| | w/o | Intra-GrAG | 0.399 | 0.400 | 0.186 | 0.276 | 0.509 | 0.320 | 0.239 | 0.294 |
| Replace | Inter-GrAG | Attention | 0.395 | 0.401 | 0.168 | 0.265 | 0.478 | 0.303 | 0.206 | 0.270 |
| | Attention | Intra-GrAG | 0.403 | 0.406 | 0.171 | 0.268 | 0.459 | 0.305 | 0.206 | 0.276 |
| | Attention | Attention | 0.395 | 0.404 | 0.171 | 0.269 | **0.453** | 0.300 | 0.204 | **0.264** |

**Ablation on Embedding&Patching Mechanism.** As shown in Table 3, we test the components of the Embedding&Patching module through three ablation studies: w/o Patching, w/o Time Embedding, and w/o Adaptive Embedding. The performance of GRAPHSTAGE consistently surpasses all of the ablation variants, indicating that accurate prediction relies not only on the dependency extraction module but also importantly on the use of prior knowledge. Full results are provided in Appendix G.

Table 3: Ablations on the Embedding&Patching Mechanism. The average results are listed here.

| Design | PEMS03 | | PEMS04 | | PEMS07 | | PEMS08 | |
|---|---|---|---|---|---|---|---|---|
| | MSE | MAE | MSE | MAE | MSE | MAE | MSE | MAE |
| GRAPHSTAGE | **0.097** | **0.210** | **0.090** | **0.200** | **0.080** | **0.179** | **0.139** | **0.220** |
| w/o Patching | 0.110 | 0.222 | 0.100 | 0.215 | 0.096 | 0.199 | 0.176 | 0.253 |
| w/o Time Emb. | 0.114 | 0.223 | 0.099 | 0.211 | 0.091 | 0.193 | 0.199 | 0.264 |
| w/o Adaptive Emb. | 0.121 | 0.257 | 0.098 | 0.211 | 0.116 | 0.221 | 0.203 | 0.260 |

**Variants Comparison.** We designed three model variants to validate the effectiveness of our framework. As illustrated in Figure 7, the proposed GRAPHSTAGE model is referred to as **Orig**.

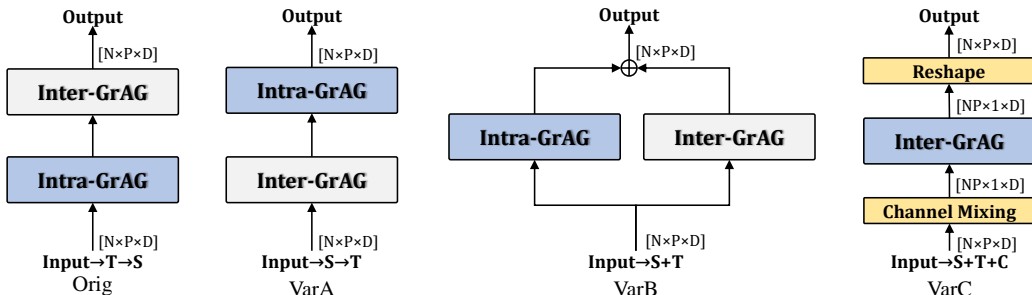

Figure 7: Model Variants. **Orig** (GRAPHSTAGE) follows an input→T→S structure, sequentially extracting temporal and then spatial dependencies. **VarA** uses input→S→T, reversing the order but remaining sequential. **VarB** employs input→S+T, a parallel structure that decouples temporal and spatial extraction before fusion. **VarC** utilizes input→S+T+C (C represents cross-series dependency as shown in Figure 2), incorporating channel-mixing with a unified architecture similar to FourierGNN (Yi et al., 2024), extracting all three types of dependencies within a unified framework.

In Variant **VarA**, we swapped the positions of the *Inter-GrAG* and *Intra-GrAG* modules. The *Inter-GrAG* module now processes the original features, rather than the temporal embeddings extracted by the *Intra-GrAG* module. The swap aims to validate the rationale of the proposed sequential architecture. VarA's performance in Table 4, shows that the original sequence—inputting the extracted temporal embeddings into the *Inter-GrAG*—contributes positively to the model's effectiveness.

In Variant **VarB**, the *Inter-GrAG* and *Intra-GrAG* modules are connected in parallel rather than sequentially. This configuration investigates whether simultaneous processing of inter-series and intra-series dependencies impacts model performance compared to the original sequential architecture. VarB's performance in Table 4 confirms the sequential structure is more effective than the parallel.

In Variant **VarC**, we adopt the same channel-mixing architecture as UniTST (Liu et al., 2024a) and FourierGNN (Yi et al., 2024), which reshapes the input data $X_{\text{in}}$ from $\mathbb{R}^{N \times T}$ to a $\mathbb{R}^{NT \times 1}$ structure. This reshaping enables the coupled learning of three types of dependencies within a unified structure. By comparing Orig with VarC, we are able to evaluate the effectiveness of our proposed channel-preserving framework. From the results in Table 4, we observe that although channel-mixing demonstrates stronger results in some cases—e.g., on the ETTm1 dataset with an input length of 96 and forecast length of 720, it outperforms Orig by 5.8%—this improvement comes at the cost of increased memory usage. Moreover, on larger datasets like ECL, channel blending leads to an exponential increase in parameters and a sharp decrease in prediction accuracy. By treating the original multivariate time series as a univariate time series of length $N \times T$, the coupled dependencies learning introduces more interference and noise compared to the proposed decoupled framework. This highlights the advantages of our channel-preserving strategy, which maintains computational efficiency and reduces noise while effectively capturing the essential dependencies.

The comparisons among these variants validate the design of GRAPHSTAGE. The sequential structure in **Orig** (GRAPHSTAGE) proves to be more effective than altering the module order (**VarA**) or processing dependencies in parallel (**VarB**). Additionally, our channel-preserving framework demonstrates superior scalability and efficiency compared to the channel-mixing strategy in **VarC**, especially on larger datasets. This underscores the importance of preserving the original data structure and decoupling the learning of inter-series and intra-series dependencies in MTSF models.

**Visualization of Learned Dependencies.** We conducted heatmap visualizations of dependencies on three datasets with different sampling frequencies: ETTm1, ECL, and PEMS04. For ETTm1, the input length is set to 288, corresponding to 3 days of data, as the sampling frequency is 15 minutes ($288 \times 15$ minutes = 3 days). For ECL, the input length is 96, meaning each sample contains 4 days of data, given the sampling frequency of 1 hour ($96 \times 1$ hour = 4 days). For PEMS04 with 5-minute intervals, the input length is set to 576, meaning each sample contains 2 days of input data.

Table 4: Model variants. All models are evaluated on 4 different predication lengths. The best results are in **red**, the second results are in blue, and the highest memory usage is in **bold**.

| Models | Orig (GRAPHSTAGE) | | | VarA | | | VarB | | | VarC | | |
|---|---|---|---|---|---|---|---|---|---|---|---|---|
| **Metric** | MSE | MAE | Mem (GB) | MSE | MAE | Mem (GB) | MSE | MAE | Mem (GB) | MSE | MAE | Mem (GB) |
| **ETTm1** 96 | 0.319 | **0.356** | 0.522 | 0.326 | 0.361 | 0.522 | **0.316** | 0.357 | 0.522 | 0.325 | 0.361 | **0.558** |
| 192 | 0.367 | 0.381 | 0.522 | **0.365** | **0.383** | 0.522 | 0.373 | 0.390 | 0.522 | 0.370 | 0.387 | **0.578** |
| 336 | **0.394** | **0.400** | 0.522 | 0.403 | 0.413 | 0.522 | 0.401 | 0.409 | 0.522 | 0.402 | 0.410 | **0.578** |
| 720 | 0.482 | **0.441** | 0.544 | **0.456** | 0.444 | 0.544 | 0.476 | 0.450 | 0.544 | 0.458 | 0.443 | **0.597** |
| AVG | 0.391 | **0.394** | 0.528 | **0.388** | 0.400 | 0.528 | 0.392 | 0.402 | 0.528 | 0.389 | 0.400 | **0.578** |
| **ECL** 96 | **0.139** | **0.237** | 4.066 | 0.166 | 0.257 | 3.920 | 0.156 | 0.250 | 4.110 | 0.170 | 0.265 | **23.703** |
| 192 | **0.155** | **0.251** | 4.080 | 0.172 | 0.265 | 3.920 | 0.169 | 0.262 | 4.124 | 0.175 | 0.267 | **23.725** |
| 336 | **0.175** | **0.272** | 4.086 | 0.193 | 0.285 | 4.100 | 0.184 | 0.277 | 4.186 | 0.192 | 0.285 | **23.749** |
| 720 | **0.196** | **0.292** | 4.144 | 0.235 | 0.319 | 4.120 | 0.225 | 0.313 | 4.200 | 0.231 | 0.317 | **23.794** |
| AVG | **0.166** | **0.263** | 4.094 | 0.192 | 0.282 | 4.015 | 0.184 | 0.276 | 4.155 | 0.192 | 0.284 | **23.743** |

In experiments, we set the patch stride to 2 and randomly selected one Temporal Learnable Graph ($A_T$) for each dataset, as shown in Figure 8. In ETTm1's $A_T^{(1)}$, peaks occur every 48 patches, corresponding to 24 hours. Similarly, ECL's $A_T^{(2)}$ shows peak every 12 patches (24 hours), and PEMS04's $A_T^{(3)}$ peaks every 144 patches (24 hours). These visualizations demonstrate that the periodicity extracted by the Inter-GrAG module matches the inherent daily periodicity of each dataset. This match confirms our method effectively captures and visualizes the daily patterns in the data. Appendix H provides additional $A_T$ visualizations and the analysis of Spatial Learnable Graph ($A_S$).

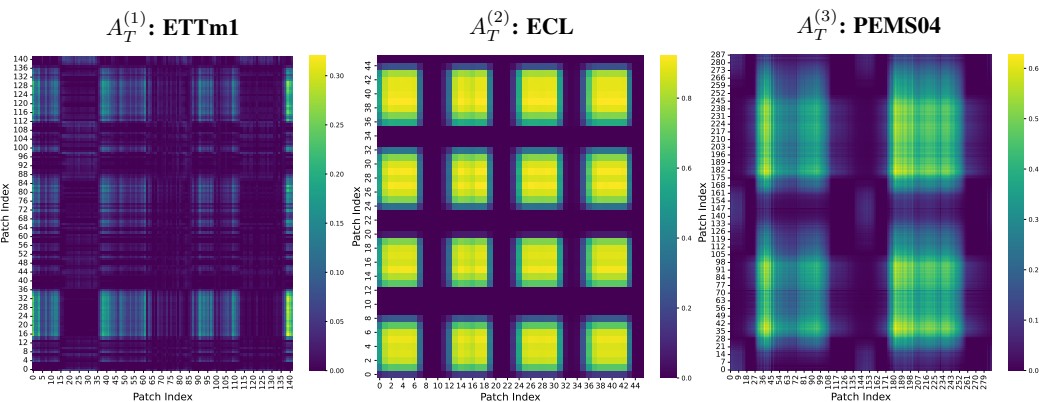

Figure 8: Visualization of Temporal Learnable Graphs ($A_T$) across different datasets (ETTm1, ECL, PEMS04). Each column represents a randomly selected $A_T$ from the results of GRAPHSTAGE.

## 5 CONCLUSION

Current models primarily focus on the advantages of channel-mixing methods for extracting multiple dependencies, often neglecting the noise these approaches can introduce. GRAPHSTAGE is the first model to directly address this issue. Through the model variants experiments in Section 4.3, we validated the presence of such interference, underscoring the limitations of excessive dependency extraction. To mitigate these challenges, GRAPHSTAGE utilizes a decoupled architecture that independently extracts inter-series and intra-series dependencies. As a fully graph-based, channel-preserving framework, GRAPHSTAGE maintains the integrity of the original channel structures, effectively avoiding the interference and noise associated with channel blending. Extensive experiments conducted on 13 real-world datasets demonstrate that GRAPHSTAGE achieves performance on par with, or surpassing, state-of-the-art methods. Future research could explore decoupled extraction of cross-series dependencies and develop inductive models that maintain channel preservation.

# 6 ETHICS STATEMENT

Our work focuses solely on scientific challenges and does not involve human subjects, animals, or environmentally sensitive materials. We foresee no ethical risks or conflicts of interest. We are committed to upholding the highest standards of scientific integrity and ethical conduct to ensure the validity and reliability of our findings.

# 7 REPRODUCIBILITY STATEMENT

We provide detailed implementation information in Appendix A, B, and C, including additional model details, descriptions of the datasets, hyperparameters, and experiment settings. For reproducibility, the source code is made available through an anonymous link: `https://anonymous.4open.science/r/GraphSTAGE`.

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

## A    IMPLEMENTATION DETAILS

The detailed implementation for the Feed-Forward Network (FFN) and Gate layers are presented below. Since the Spatial-Temporal Aggregation Graph Encoder (STAGE) block employs a unified aggregation mechanism, the principles of the Inter-GrAG and Intra-GrAG modules are analogous. Therefore, to avoid redundancy, we focus on the components of the Inter-GrAG module. The transposed input to the Inter-GrAG module is $H_{in} \in \mathbb{R}^{N \times P \times D}$, and the output of the Pruned-Graph Aggregation (PGA) is $H_{ag} \in \mathbb{R}^{N \times P \times D}$. The module employs a FFN and a Gate layer to generate the encoder output $H_E$.

**Feed-Forward Network (FFN).**    The FFN is responsible for processing the aggregated features to capture nonlinear transformations. It introduces nonlinearity and enhances the model's capacity to learn complex representations. As formulated in Equation 6, the FFN consists of two linear layers with ReLU activation functions. To facilitate better gradient flow and mitigate the vanishing gradient problem, residual connections are employed. Specifically, after the FFN processes the features, a residual connection adds the dropped $H_{\text{FFN}}$ back to the original input $H_{in}$, followed by layer normalization.

$$H_{res} = \text{LayerNorm}\left(\text{Dropout}(H_{\text{FFN}}) + H_{in}\right), \tag{6a}$$

$$H_{\text{FFN}} = \text{ReLU}\left(\text{Linear}\left(\text{ReLU}\left(\text{Linear}(H_{ag})\right)\right)\right). \tag{6b}$$

**Gate Layer.**    We use the same Gate layer as UniTS (Gao et al., 2024). The Gate layer is placed at the output of each Inter-GrAG and Intra-GrAG module within the STAGE blocks to regulate the flow of information. Specifically, given an input $H_{res} \in \mathbb{R}^{N \times P \times D}$, a linear layer maps the input to a scaling factor $H_l \in \mathbb{R}^{N \times P \times 1}$ along the embedding dimension. This is followed by a Sigmoid function to ensure the scaling factor lies between 0 and 1. The final gating operation involves element-wise multiplication of the input with the Sigmoid-activated scaling factor, as formulated in Equation 7.

$$H_E = \text{Sigmoid}(H_l) \odot H_{res}, \quad H_l = \text{Linear}(H_{res}). \tag{7}$$

This gating mechanism enhances the model's ability to capture complex dependencies by adaptively weighing the importance of different features.

Additionally, the pseudocode of GRAPHSTAGE, which outlines the key steps and components, is provided in Algorithm 1. This serves as a comprehensive guide to understanding the implementation details of our proposed model.

## B    DATASETS DETAILS FOR MULTIVARIATE TIME SERIES FORECASTING

We conduct experiments on 13 real-world datasets, covering a diverse range of application scenarios and facilitating a comprehensive evaluation of the model. The details of the datasets are as follows: (1) ETT (Li et al., 2021) records 7 features of electricity transformer at two time scales: hourly and every 15 minutes. The data are sourced from two regions, resulting in four subsets: ETTh1, ETTh2, ETTm1, and ETTm2. (2) ECL (Wu et al., 2021) records the hourly electricity consumption data of 321 customers. (3) Exchange (Lai et al., 2018b) collects the data of daily exchange rates for 8 countries from 1990 to 2016. (4) Traffic (Wu et al., 2023) contains hourly road occupancy rates measured by 862 sensors on San Francisco Bay area freeways in two years. (5) Weather (Liu et al., 2024c) records 21 meteorological indicators at 10-minute intervals. (6) Solar-Energy (Lai et al., 2018a) includes solar power production data from 137 photovoltaic plants in 2006, with recording taken every 10 minutes. (7) PEMS (Choe et al., 2002) collects traffic network data in California through multiple detection instruments. We adopt four subsets—PEMS03, PEMS04, PEMS07, and PEMS08 used by ASTGCN (Guo et al., 2019). The details of datasets are provided in Table 5.

Table 5: Detailed dataset descriptions. *Nodes* denote the node numbers of each dataset. *Prediction Length* denotes the future time points to be predicted and four prediction settings are included in each dataset. *Dataset Size* refers to the total number of time points in (Train, Validation, Test) split respectively. *Frequency* denotes the sampling frequency of time points.

| Dataset | Nodes | Prediction Length | Dataset Size | Frequency |
|---|---|---|---|---|
| ETTh1 | 7 | {96, 192, 336, 720} | (8545, 2881, 2881) | Hourly |
| ETTh2 | 7 | {96, 192, 336, 720} | (8545, 2881, 2881) | Hourly |
| ETTm1 | 7 | {96, 192, 336, 720} | (34465, 11521, 11521) | 15min |
| ETTm2 | 7 | {96, 192, 336, 720} | (34465, 11521, 11521) | 15min |
| Exchange | 8 | {96, 192, 336, 720} | (5120, 665, 1422) | Daily |
| Weather | 21 | {96, 192, 336, 720} | (36792, 5271, 10540) | 10min |
| ECL | 321 | {96, 192, 336, 720} | (18317, 2633, 5261) | Hourly |
| Traffic | 862 | {96, 192, 336, 720} | (12185, 1757, 3509) | Hourly |
| Solar-Energy | 137 | {96, 192, 336, 720} | (36601, 5161, 10417) | 10min |
| PEMS03 | 358 | {12, 24, 48, 96} | (15617, 5135, 5135) | 5min |
| PEMS04 | 307 | {12, 24, 48, 96} | (10172, 3375, 3375) | 5min |
| PEMS07 | 883 | {12, 24, 48, 96} | (16911, 5622, 5622) | 5min |
| PEMS08 | 170 | {12, 24, 48, 96} | (10690, 3548, 3548) | 5min |

---

**Algorithm 1** The learning algorithm of GRAPHSTAGE.

---

**Require:** Input historical time series $\mathbf{X} \in \mathbb{R}^{N \times T}$; input length $T$; prediction length $K$; nodes number $N$; patches number $P$; patch stride $s$; embedding dimension $D$; STAGE block number $L$.

1: $\mathbf{Base} = \texttt{Mean}(\mathbf{X})$ ▷ $\mathbf{Base} \in \mathbb{R}^{N \times 1}$

2: $\mathbf{X} = \texttt{Patching}(\mathbf{X})$ ▷ $\mathbf{X} \in \mathbb{R}^{N \times P \times s}$

3: ▷ Projecton works on the last dimension to map series into embedding dimension $D$.

4: $\mathbf{X_p} = \texttt{Projecton}(\mathbf{X})$ ▷ $\mathbf{X_p} \in \mathbb{R}^{N \times P \times D}$

5: ▷ Refined time embedding to enhance relative positioning.

6: $\mathbf{H}^0 = \texttt{Embedding}(\mathbf{X_p})$ ▷ $\mathbf{H}^0 \in \mathbb{R}^{N \times P \times D}$

7: **for** $l$ **in** $\{1, \ldots, L\}$**:** ▷ Run through stacked STAGE blocks.

8:     ▷ Intra-GrAG module to capture temporal dependency.

9:     $\mathbf{H_t}^{l-1} = \texttt{IntraGrAG}(\mathbf{H}^{l-1}.\texttt{transpose})$ ▷ $\mathbf{H_t}^{l-1} \in \mathbb{R}^{P \times N \times D}$

10:     ▷ Inter-GrAG module to capture spatial dependency.

11:     $\mathbf{H}^l = \texttt{InterGrAG}(\mathbf{H_t}^{l-1}.\texttt{transpose})$ ▷ $\mathbf{H}^l \in \mathbb{R}^{N \times P \times D}$

12: **End for**

13: $\hat{\mathbf{Y}} = \texttt{Projecton}(\mathbf{H}^L)$ ▷ Project tokens back to predicted series, $\hat{\mathbf{Y}} \in \mathbb{R}^{N \times K}$

14: $\hat{\mathbf{Y}} = \hat{\mathbf{Y}} + \mathbf{Base}$ ▷ $\hat{\mathbf{Y}} \in \mathbb{R}^{N \times K}$

15: **Return** $\hat{\mathbf{Y}}$ ▷ Return the prediction result $\hat{\mathbf{Y}}$

## C   HYPERPARAMETERS AND SETTINGS

All experiments are conducted on a single RTX 4090 24GB GPU, and we utilize the Adam (Kingma & Ba, 2015) optimizer to optimize the training process. All experiments are repeated five times and we report the averaged results. The batch size is consistently set to 16, and the number of training epochs is fixed to 10. We conduct a grid search to determine the best configuration. We consistently set the embedding dimension $D$ to 64, and the number of STAGE layers between 1 and 2. Normalization is skipped before the embedding process for the PEMS and Solar-Energy datasets, and performed in advance for all other datasets. Table 6 outlines the specific hyperparameters used for each dataset.

We partition the dataset for train-validation-test following the methodology established in Times-Net (Wu et al., 2023), to ensure the comparability of subsequent experiments. For the forecasting settings, the lookback length for all datasets is set to 96. The prediction horizon varies across $\{12, 24, 48, 96\}$ for the PEMS datasets and $\{96, 192, 336, 720\}$ for the other datasets.

Table 6: Hyperparameters of GRAPHSTAGE on different datasets.

| Dataset | ETTm1 | ETTm2 | ETTh1 | ETTh2 | ECL | Exchange | Weather | Traffic | Solar-Energy | PEMS03 | PEMS04 | PEMS07 | PEMS08 |
|---|---|---|---|---|---|---|---|---|---|---|---|---|---|
| Epochs | | | | | | | 10 | | | | | | |
| Batch | | | | | | | 16 | | | | | | |
| Loss | | | | | | | MSE | | | | | | |
| Learning Rate | 1e-3 | | | | 2e-3 | 2e-4 | 5e-4 | 5e-3 | 5e-4 | | | 2e-3 | |
| Layers | | | 1 | | | | | | 2 | | | 1 | |
| Use Norm | | | 1 | | | | | | | 0 | | | |
| $D$ | | | | | | | 64 | | | | | | |
| $c$ | | | | | | | 12 | | | | | | |
| Optimizer | | | | | | | Adam | | | | | | |

## D   FULL RESULT ACROSS 13 REAL-WORLD DATASETS

In this section, we provide detailed multivariate prediction results across 13 real-world datasets.

Table 7 summarizes the results for various prediction lengths across 9 benchmark datasets. The results indicate that GRAPHSTAGE consistently compares to or outperforms other models across all datasets, securing the highest rank in MSE and MAE 26 and 25 times, respectively.

Table 8 presents the forecasting results for the four subsets of the PEMS dataset. Notably, GRAPH-STAGE achieves the best MSE in 20 out of 21 comparisons and the best MAE in 19 out of 21 comparisons across the PEMS datasets. Specifically on PEMS07, the model achieves a significant improvement over the recent state-of-the-art iTransformer, with a margin of 20.8%.

Table 9 contains comparison results with advanced GNNs, including four well-known models: FourierGNN (Yi et al., 2024), CrossGNN (Huang et al., 2023), StemGNN (Cao et al., 2020), and MTGNN (Wu et al., 2020). We reproduce the result of FourierGNN (Yi et al., 2024) and StemGNN (Cao et al., 2020), while collecting the other baseline results from TimesNet (Wu et al., 2023). All experiments are repeated five times and we report the averaged results. The results indicate that GRAPHSTAGE achieves top-1 performance in most cases. Notably, on the largest-scale dataset (ECL with 321 nodes), it outperforms the second-best model (CrossGNN) by significant margins, with reductions in MSE and MAE exceeding 17.4% and 12.3%, respectively.

Table 7: Full results of the long-term forecasting task. We compare extensive competitive models under different prediction lengths following the setting of iTransformer (Liu et al., 2024c). The input sequence length is set to 96 for all baselines. *AVG* means the average results from all four prediction lengths: $\{96, 192, 336, 720\}$.

| Models | | Ours | | iTransformer | | RLinear | | PatchTST | | Crossformer | | TimesNet | | DLinear | | SCINet | |
|---|---|---|---|---|---|---|---|---|---|---|---|---|---|---|---|---|---|
| Metric | | MSE | MAE | MSE | MAE | MSE | MAE | MSE | MAE | MSE | MAE | MSE | MAE | MSE | MAE | MSE | MAE |
| ETTm1 | 96 | **0.319** | **0.356** | 0.334 | 0.368 | 0.355 | 0.376 | 0.329 | 0.367 | 0.404 | 0.426 | 0.338 | 0.375 | 0.345 | 0.372 | 0.418 | 0.438 |
| | 192 | **0.367** | **0.381** | 0.377 | 0.391 | 0.391 | 0.392 | 0.367 | 0.385 | 0.450 | 0.451 | 0.374 | 0.387 | 0.380 | 0.389 | 0.439 | 0.450 |
| | 336 | **0.394** | **0.400** | 0.426 | 0.420 | 0.424 | 0.415 | 0.399 | 0.410 | 0.532 | 0.515 | 0.410 | 0.411 | 0.413 | 0.413 | 0.490 | 0.485 |
| | 720 | 0.482 | 0.441 | 0.491 | 0.459 | 0.487 | 0.450 | **0.454** | **0.439** | 0.666 | 0.589 | 0.478 | 0.450 | 0.474 | 0.453 | 0.595 | 0.550 |
| | AVG | 0.391 | **0.394** | 0.407 | 0.410 | 0.414 | 0.407 | **0.387** | 0.400 | 0.513 | 0.496 | 0.400 | 0.406 | 0.403 | 0.407 | 0.485 | 0.481 |
| ETTm2 | 96 | **0.174** | **0.259** | 0.180 | 0.264 | 0.182 | 0.265 | 0.175 | 0.259 | 0.287 | 0.366 | 0.187 | 0.267 | 0.193 | 0.292 | 0.286 | 0.377 |
| | 192 | 0.241 | 0.304 | 0.250 | 0.309 | 0.246 | 0.304 | **0.241** | **0.302** | 0.414 | 0.492 | 0.249 | 0.309 | 0.284 | 0.362 | 0.399 | 0.445 |
| | 336 | **0.301** | **0.341** | 0.311 | 0.348 | 0.307 | 0.342 | 0.305 | 0.343 | 0.597 | 0.542 | 0.321 | 0.351 | 0.369 | 0.427 | 0.637 | 0.591 |
| | 720 | **0.397** | **0.398** | 0.412 | 0.407 | 0.407 | 0.398 | 0.402 | 0.400 | 1.730 | 1.042 | 0.408 | 0.403 | 0.554 | 0.522 | 0.960 | 0.735 |
| | AVG | **0.278** | **0.325** | 0.288 | 0.332 | 0.286 | 0.327 | 0.281 | 0.326 | 0.757 | 0.610 | 0.291 | 0.333 | 0.350 | 0.401 | 0.571 | 0.537 |
| ETTh1 | 96 | **0.384** | **0.395** | 0.386 | 0.405 | 0.386 | 0.395 | 0.414 | 0.419 | 0.423 | 0.448 | 0.384 | 0.402 | 0.386 | 0.400 | 0.654 | 0.599 |
| | 192 | **0.435** | 0.426 | 0.441 | 0.436 | 0.437 | **0.424** | 0.460 | 0.445 | 0.471 | 0.474 | 0.436 | 0.429 | 0.437 | 0.432 | 0.719 | 0.631 |
| | 336 | **0.476** | **0.441** | 0.487 | 0.458 | 0.479 | 0.446 | 0.501 | 0.466 | 0.570 | 0.546 | 0.491 | 0.469 | 0.481 | 0.459 | 0.778 | 0.659 |
| | 720 | 0.487 | **0.460** | 0.503 | 0.491 | **0.481** | 0.470 | 0.500 | 0.488 | 0.653 | 0.621 | 0.521 | 0.500 | 0.519 | 0.516 | 0.836 | 0.699 |
| | AVG | **0.445** | **0.430** | 0.454 | 0.447 | 0.446 | 0.434 | 0.469 | 0.454 | 0.529 | 0.522 | 0.458 | 0.450 | 0.456 | 0.452 | 0.747 | 0.647 |
| ETTh2 | 96 | 0.292 | 0.341 | 0.297 | 0.349 | **0.288** | **0.338** | 0.302 | 0.348 | 0.745 | 0.584 | 0.340 | 0.374 | 0.333 | 0.387 | 0.707 | 0.621 |
| | 192 | 0.380 | 0.395 | 0.380 | 0.400 | **0.374** | **0.390** | 0.388 | 0.400 | 0.877 | 0.656 | 0.402 | 0.414 | 0.477 | 0.476 | 0.860 | 0.689 |
| | 336 | 0.424 | 0.431 | 0.428 | 0.432 | **0.415** | **0.426** | 0.426 | 0.433 | 1.043 | 0.731 | 0.452 | 0.452 | 0.594 | 0.541 | 1.000 | 0.744 |
| | 720 | 0.453 | 0.459 | 0.427 | 0.445 | **0.420** | **0.440** | 0.431 | 0.446 | 1.104 | 0.763 | 0.462 | 0.468 | 0.831 | 0.657 | 1.249 | 0.838 |
| | AVG | 0.387 | 0.407 | 0.383 | 0.407 | **0.374** | **0.398** | 0.387 | 0.407 | 0.942 | 0.684 | 0.414 | 0.427 | 0.559 | 0.515 | 0.954 | 0.723 |
| ECL | 96 | **0.139** | **0.237** | 0.148 | 0.240 | 0.201 | 0.281 | 0.181 | 0.270 | 0.219 | 0.314 | 0.168 | 0.272 | 0.197 | 0.282 | 0.247 | 0.345 |
| | 192 | **0.155** | **0.251** | 0.162 | 0.253 | 0.201 | 0.283 | 0.188 | 0.274 | 0.231 | 0.322 | 0.184 | 0.289 | 0.196 | 0.285 | 0.257 | 0.355 |
| | 336 | **0.175** | 0.272 | 0.178 | **0.269** | 0.215 | 0.298 | 0.204 | 0.293 | 0.246 | 0.337 | 0.198 | 0.300 | 0.209 | 0.301 | 0.269 | 0.369 |
| | 720 | **0.196** | **0.292** | 0.225 | 0.317 | 0.257 | 0.331 | 0.246 | 0.324 | 0.280 | 0.363 | 0.220 | 0.320 | 0.245 | 0.333 | 0.299 | 0.390 |
| | AVG | **0.166** | **0.263** | 0.178 | 0.270 | 0.219 | 0.298 | 0.205 | 0.290 | 0.244 | 0.334 | 0.192 | 0.295 | 0.212 | 0.300 | 0.268 | 0.365 |
| Exchange | 96 | **0.084** | **0.203** | 0.086 | 0.206 | 0.093 | 0.217 | 0.088 | 0.205 | 0.256 | 0.367 | 0.107 | 0.234 | 0.088 | 0.218 | 0.267 | 0.396 |
| | 192 | 0.186 | 0.306 | 0.177 | 0.299 | 0.184 | 0.307 | **0.176** | **0.299** | 0.470 | 0.509 | 0.226 | 0.344 | 0.176 | 0.315 | 0.351 | 0.459 |
| | 336 | 0.339 | 0.420 | 0.331 | 0.417 | 0.351 | 0.432 | **0.301** | **0.397** | 1.268 | 0.883 | 0.367 | 0.448 | 0.313 | 0.427 | 1.324 | 0.853 |
| | 720 | 0.898 | 0.710 | 0.847 | **0.691** | 0.886 | 0.714 | 0.901 | 0.714 | 1.767 | 1.068 | 0.964 | 0.746 | **0.839** | 0.695 | 1.058 | 0.797 |
| | AVG | 0.376 | 0.409 | 0.360 | **0.403** | 0.378 | 0.417 | 0.367 | 0.404 | 0.940 | 0.707 | 0.416 | 0.443 | **0.354** | 0.414 | 0.750 | 0.626 |
| Traffic | 96 | 0.438 | 0.281 | **0.395** | **0.268** | 0.649 | 0.389 | 0.462 | 0.295 | 0.522 | 0.290 | 0.593 | 0.321 | 0.650 | 0.396 | 0.788 | 0.499 |
| | 192 | 0.442 | 0.282 | **0.417** | **0.276** | 0.601 | 0.366 | 0.466 | 0.296 | 0.530 | 0.293 | 0.617 | 0.336 | 0.598 | 0.370 | 0.789 | 0.505 |
| | 336 | 0.461 | 0.292 | **0.433** | **0.283** | 0.609 | 0.369 | 0.482 | 0.304 | 0.558 | 0.305 | 0.629 | 0.336 | 0.605 | 0.373 | 0.797 | 0.508 |
| | 720 | 0.509 | 0.322 | **0.467** | **0.302** | 0.647 | 0.387 | 0.514 | 0.322 | 0.589 | 0.328 | 0.640 | 0.350 | 0.645 | 0.394 | 0.841 | 0.523 |
| | AVG | 0.462 | 0.294 | **0.428** | **0.282** | 0.626 | 0.378 | 0.481 | 0.304 | 0.550 | 0.304 | 0.620 | 0.336 | 0.625 | 0.383 | 0.804 | 0.509 |
| Weather | 96 | 0.159 | **0.208** | 0.174 | 0.214 | 0.192 | 0.232 | 0.177 | 0.218 | **0.158** | 0.230 | 0.172 | 0.220 | 0.196 | 0.255 | 0.221 | 0.306 |
| | 192 | 0.207 | **0.251** | 0.221 | 0.254 | 0.240 | 0.271 | 0.225 | 0.259 | **0.206** | 0.277 | 0.219 | 0.261 | 0.237 | 0.296 | 0.261 | 0.340 |
| | 336 | **0.263** | **0.292** | 0.278 | 0.296 | 0.292 | 0.307 | 0.278 | 0.297 | 0.272 | 0.335 | 0.280 | 0.306 | 0.283 | 0.335 | 0.309 | 0.378 |
| | 720 | **0.344** | **0.345** | 0.358 | 0.347 | 0.364 | 0.353 | 0.354 | 0.348 | 0.398 | 0.418 | 0.365 | 0.359 | 0.345 | 0.381 | 0.377 | 0.427 |
| | AVG | **0.243** | **0.274** | 0.258 | 0.278 | 0.272 | 0.291 | 0.259 | 0.281 | 0.259 | 0.315 | 0.259 | 0.287 | 0.265 | 0.317 | 0.292 | 0.363 |
| Solar-Energy | 96 | **0.172** | 0.258 | 0.203 | **0.237** | 0.322 | 0.339 | 0.234 | 0.286 | 0.310 | 0.331 | 0.250 | 0.292 | 0.290 | 0.378 | 0.237 | 0.344 |
| | 192 | **0.183** | **0.259** | 0.233 | 0.261 | 0.359 | 0.356 | 0.267 | 0.310 | 0.734 | 0.725 | 0.296 | 0.318 | 0.320 | 0.398 | 0.280 | 0.380 |
| | 336 | **0.205** | 0.278 | 0.248 | **0.273** | 0.397 | 0.369 | 0.290 | 0.315 | 0.750 | 0.735 | 0.319 | 0.330 | 0.353 | 0.415 | 0.304 | 0.389 |
| | 720 | **0.211** | **0.273** | 0.249 | 0.275 | 0.397 | 0.356 | 0.289 | 0.317 | 0.769 | 0.765 | 0.338 | 0.337 | 0.356 | 0.413 | 0.308 | 0.388 |
| | AVG | **0.192** | 0.267 | 0.233 | **0.262** | 0.369 | 0.356 | 0.270 | 0.307 | 0.641 | 0.639 | 0.301 | 0.319 | 0.330 | 0.401 | 0.282 | 0.375 |
| Average | | **0.327** | **0.340** | 0.332 | 0.343 | 0.376 | 0.367 | 0.345 | 0.353 | 0.597 | 0.512 | 0.372 | 0.366 | 0.395 | 0.399 | 0.573 | 0.514 |
| 1st Count | | **26** | **25** | 5 | 11 | 6 | 6 | 5 | 4 | 2 | 0 | 0 | 0 | 2 | 0 | 0 | 0 |

Table 8: Full results of the PEMS forecasting task. We compare extensive competitive models under different prediction lengths following the setting of SCINet (Liu et al., 2022). The input length is set to 96 for all baselines. *AVG* means the average results from all four prediction lengths: $\{12, 24, 48, 96\}$.

| Models | | Ours | | iTransformer | | RLinear | | PatchTST | | Crossformer | | TimesNet | | DLinear | | SCINet | |
|---|---|---|---|---|---|---|---|---|---|---|---|---|---|---|---|---|---|
| Metric | | MSE | MAE | MSE | MAE | MSE | MAE | MSE | MAE | MSE | MAE | MSE | MAE | MSE | MAE | MSE | MAE |
| PEMS03 | 12 | 0.065 | 0.170 | 0.071 | 0.174 | 0.126 | 0.236 | 0.099 | 0.216 | 0.090 | 0.203 | 0.085 | 0.192 | 0.122 | 0.243 | 0.066 | 0.172 |
| | 24 | 0.082 | 0.193 | 0.093 | 0.201 | 0.246 | 0.334 | 0.142 | 0.259 | 0.121 | 0.240 | 0.118 | 0.223 | 0.201 | 0.317 | 0.085 | 0.198 |
| | 48 | 0.106 | 0.219 | 0.125 | 0.236 | 0.551 | 0.529 | 0.211 | 0.319 | 0.202 | 0.317 | 0.155 | 0.260 | 0.333 | 0.425 | 0.127 | 0.238 |
| | 96 | 0.136 | 0.253 | 0.164 | 0.275 | 1.057 | 0.787 | 0.269 | 0.370 | 0.262 | 0.367 | 0.228 | 0.317 | 0.457 | 0.515 | 0.178 | 0.287 |
| | AVG | 0.097 | 0.210 | 0.113 | 0.221 | 0.495 | 0.472 | 0.180 | 0.291 | 0.169 | 0.281 | 0.147 | 0.248 | 0.278 | 0.375 | 0.114 | 0.224 |
| PEMS04 | 12 | 0.070 | 0.174 | 0.078 | 0.183 | 0.138 | 0.252 | 0.105 | 0.224 | 0.098 | 0.218 | 0.087 | 0.195 | 0.148 | 0.272 | 0.073 | 0.177 |
| | 24 | 0.082 | 0.190 | 0.095 | 0.205 | 0.258 | 0.348 | 0.153 | 0.275 | 0.131 | 0.256 | 0.103 | 0.215 | 0.224 | 0.340 | 0.084 | 0.193 |
| | 48 | 0.096 | 0.207 | 0.120 | 0.233 | 0.572 | 0.544 | 0.229 | 0.339 | 0.205 | 0.326 | 0.136 | 0.250 | 0.355 | 0.437 | 0.099 | 0.211 |
| | 96 | 0.113 | 0.228 | 0.150 | 0.262 | 1.137 | 0.820 | 0.291 | 0.389 | 0.402 | 0.457 | 0.190 | 0.303 | 0.452 | 0.504 | 0.114 | 0.227 |
| | AVG | 0.090 | 0.200 | 0.111 | 0.221 | 0.526 | 0.491 | 0.195 | 0.307 | 0.209 | 0.314 | 0.129 | 0.241 | 0.295 | 0.388 | 0.092 | 0.202 |
| PEMS07 | 12 | 0.056 | 0.152 | 0.067 | 0.165 | 0.118 | 0.235 | 0.095 | 0.207 | 0.094 | 0.200 | 0.082 | 0.181 | 0.115 | 0.242 | 0.068 | 0.171 |
| | 24 | 0.072 | 0.175 | 0.088 | 0.190 | 0.242 | 0.341 | 0.150 | 0.262 | 0.139 | 0.247 | 0.101 | 0.204 | 0.210 | 0.329 | 0.119 | 0.225 |
| | 48 | 0.087 | 0.179 | 0.110 | 0.215 | 0.562 | 0.541 | 0.253 | 0.340 | 0.311 | 0.369 | 0.134 | 0.238 | 0.398 | 0.458 | 0.149 | 0.237 |
| | 96 | 0.105 | 0.209 | 0.139 | 0.245 | 1.096 | 0.795 | 0.346 | 0.404 | 0.396 | 0.442 | 0.181 | 0.279 | 0.594 | 0.553 | 0.141 | 0.234 |
| | AVG | 0.080 | 0.179 | 0.101 | 0.204 | 0.504 | 0.478 | 0.211 | 0.303 | 0.235 | 0.315 | 0.124 | 0.225 | 0.329 | 0.395 | 0.119 | 0.234 |
| PEMS08 | 12 | 0.085 | 0.175 | 0.079 | 0.182 | 0.133 | 0.247 | 0.168 | 0.232 | 0.165 | 0.214 | 0.112 | 0.212 | 0.154 | 0.276 | 0.087 | 0.184 |
| | 24 | 0.111 | 0.205 | 0.115 | 0.219 | 0.249 | 0.343 | 0.224 | 0.281 | 0.215 | 0.260 | 0.141 | 0.238 | 0.248 | 0.353 | 0.122 | 0.221 |
| | 48 | 0.155 | 0.230 | 0.186 | 0.235 | 0.569 | 0.544 | 0.321 | 0.354 | 0.315 | 0.355 | 0.198 | 0.283 | 0.440 | 0.470 | 0.189 | 0.270 |
| | 96 | 0.207 | 0.270 | 0.221 | 0.267 | 1.166 | 0.814 | 0.408 | 0.417 | 0.377 | 0.397 | 0.320 | 0.351 | 0.674 | 0.565 | 0.236 | 0.300 |
| | AVG | 0.139 | 0.220 | 0.150 | 0.226 | 0.529 | 0.487 | 0.280 | 0.321 | 0.268 | 0.307 | 0.193 | 0.271 | 0.379 | 0.416 | 0.158 | 0.244 |
| Average | | 0.102 | 0.203 | 0.119 | 0.218 | 0.514 | 0.482 | 0.217 | 0.305 | 0.220 | 0.304 | 0.148 | 0.246 | 0.320 | 0.394 | 0.121 | 0.222 |
| 1st Count | | 20 | 19 | 1 | 1 | 0 | 0 | 0 | 0 | 0 | 0 | 0 | 0 | 0 | 0 | 0 | 1 |

Table 9: Additional comparison with advanced GNNs on long-term forecasting tasks, following the setting of TimesNet (Wu et al., 2023). The input sequence length is set to 96 for all baselines. *AVG* means the average results from all four prediction lengths: $\{96, 192, 336, 720\}$.

| Models | | Ours | | FourierGNN | | CrossGNN | | StemGNN | | MTGNN | |
|---|---|---|---|---|---|---|---|---|---|---|---|
| Metric | | MSE | MAE | MSE | MAE | MSE | MAE | MSE | MAE | MSE | MAE |
| ETTm1 | 96 | 0.319 | 0.356 | 0.389 | 0.409 | 0.335 | 0.373 | 0.470 | 0.491 | 0.379 | 0.446 |
| | 192 | 0.367 | 0.381 | 0.427 | 0.429 | 0.372 | 0.390 | 0.497 | 0.504 | 0.470 | 0.428 |
| | 336 | 0.394 | 0.400 | 0.459 | 0.451 | 0.403 | 0.411 | 0.578 | 0.557 | 0.473 | 0.430 |
| | 720 | 0.482 | 0.441 | 0.535 | 0.502 | 0.461 | 0.442 | 0.653 | 0.596 | 0.553 | 0.479 |
| | AVG | 0.391 | 0.394 | 0.453 | 0.448 | 0.393 | 0.404 | 0.550 | 0.537 | 0.469 | 0.446 |
| ETTh2 | 96 | 0.292 | 0.341 | 0.398 | 0.432 | 0.309 | 0.359 | 0.599 | 0.571 | 0.354 | 0.454 |
| | 192 | 0.380 | 0.395 | 0.556 | 0.518 | 0.390 | 0.406 | 1.296 | 0.886 | 0.457 | 0.464 |
| | 336 | 0.424 | 0.431 | 0.630 | 0.566 | 0.426 | 0.444 | 1.189 | 0.843 | 0.515 | 0.540 |
| | 720 | 0.453 | 0.459 | 0.587 | 0.551 | 0.445 | 0.464 | 1.549 | 0.946 | 0.532 | 0.576 |
| | AVG | 0.387 | 0.407 | 0.543 | 0.517 | 0.393 | 0.418 | 1.158 | 0.812 | 0.465 | 0.509 |
| Weather | 96 | 0.159 | 0.208 | 0.189 | 0.248 | 0.159 | 0.218 | 0.188 | 0.261 | 0.230 | 0.329 |
| | 192 | 0.207 | 0.251 | 0.226 | 0.283 | 0.211 | 0.266 | 0.239 | 0.306 | 0.263 | 0.322 |
| | 336 | 0.263 | 0.292 | 0.274 | 0.320 | 0.267 | 0.310 | 0.315 | 0.367 | 0.354 | 0.396 |
| | 720 | 0.344 | 0.345 | 0.339 | 0.369 | 0.352 | 0.362 | 0.412 | 0.432 | 0.409 | 0.371 |
| | AVG | 0.243 | 0.274 | 0.257 | 0.305 | 0.247 | 0.289 | 0.289 | 0.342 | 0.314 | 0.355 |
| ECL | 96 | 0.139 | 0.237 | 0.202 | 0.299 | 0.173 | 0.275 | 0.188 | 0.288 | 0.217 | 0.318 |
| | 192 | 0.155 | 0.251 | 0.207 | 0.305 | 0.195 | 0.288 | 0.194 | 0.296 | 0.238 | 0.352 |
| | 336 | 0.175 | 0.272 | 0.220 | 0.319 | 0.206 | 0.300 | 0.224 | 0.326 | 0.260 | 0.348 |
| | 720 | 0.196 | 0.292 | 0.254 | 0.349 | 0.231 | 0.335 | 0.255 | 0.352 | 0.290 | 0.369 |
| | AVG | 0.166 | 0.263 | 0.221 | 0.318 | 0.201 | 0.300 | 0.215 | 0.316 | 0.251 | 0.347 |
| Average | | 0.297 | 0.335 | 0.369 | 0.397 | 0.309 | 0.353 | 0.553 | 0.502 | 0.375 | 0.414 |
| 1st Count | | 18 | 21 | 1 | 0 | 2 | 0 | 0 | 0 | 0 | 0 |

# E Visualization of 96-to-96 Forecasting Across Datasets

In order to better compare the models, we present supplementary prediction results for four representative datasets in Figures 9, 10, 11, and 12, generated by the following models: GRAPHSTAGE, iTransformer (Liu et al., 2024c), PatchTST (Nie et al., 2023), Crossformer (Zhang & Yan, 2023), TimesNet (Wu et al., 2023), DLinear (Zeng et al., 2023) and SCINet (Liu et al., 2022). For all baselines, the input length is set to 96, with a forecasting horizon of 96 time steps.

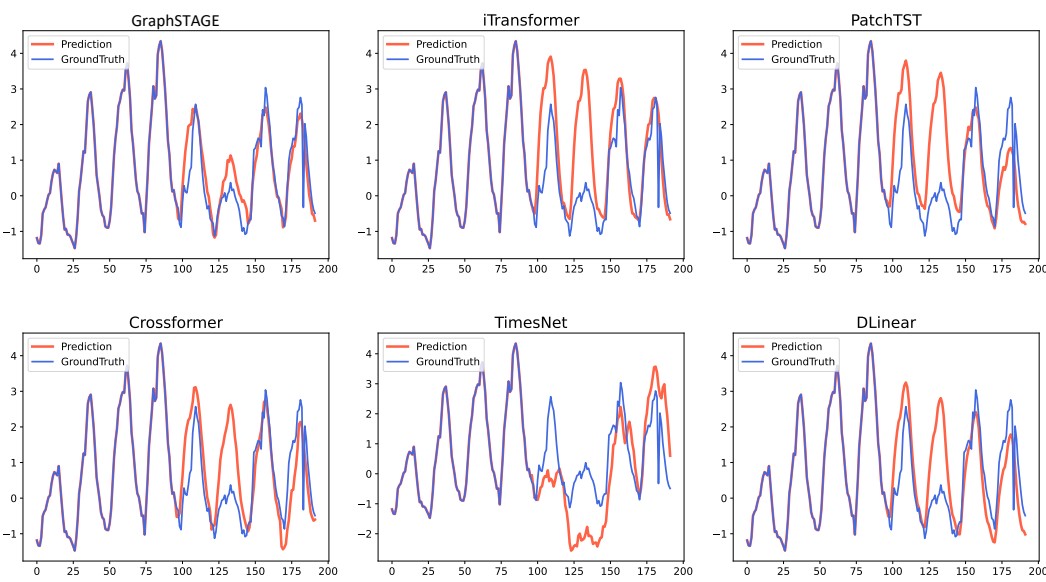

Figure 9: Sample visualization across models on ECL dataset, with forecast horizon 96.

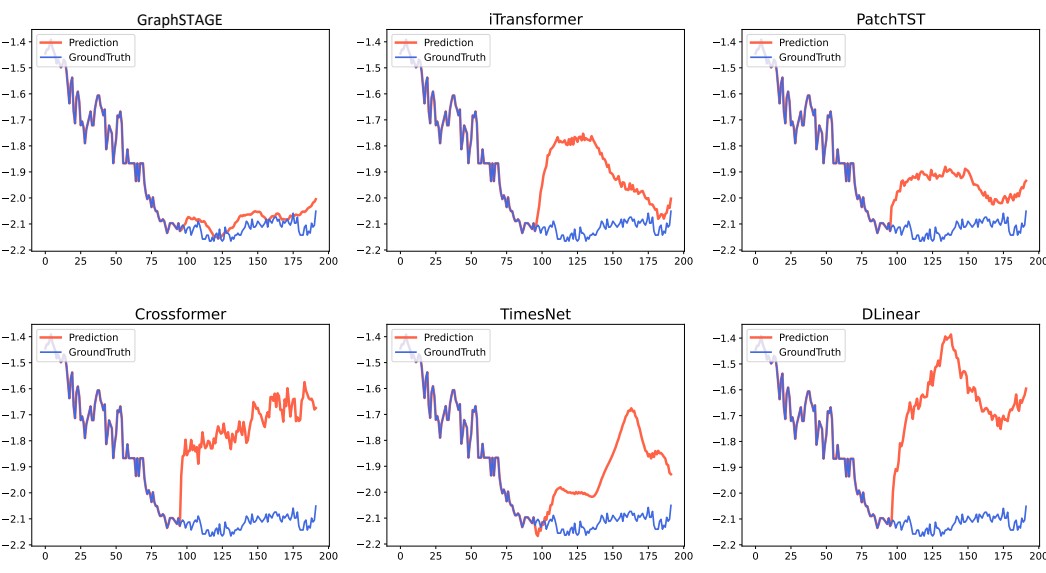

Figure 10: Sample visualization across models on ETTm1 dataset, with forecast horizon 96.

In Figure 9, GRAPHSTAGE predicts the values for time steps 125 to 150 more accurately than the other models. In Figure 10, only GRAPHSTAGE's predictions closely follow the trend of the GroundTruth, while the other models deviate significantly. In Figure 11, our model is the only one to accurately predict the peak at the 160th time step. Finally, in Figure 12, our predictions perfectly match the trend of the GroundTruth, with Crossformer (Zhang & Yan, 2023) coming in as the second best.

Overall, GRAPHSTAGE consistently delivers the most accurate predictions of future series variations, demonstrating outstanding performance across all datasets.

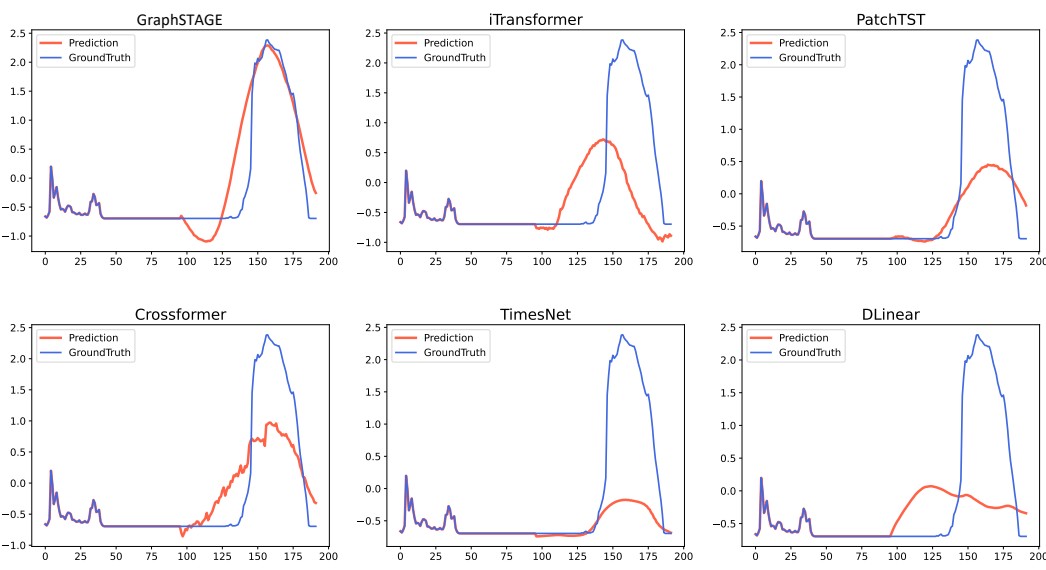

Figure 11: Sample visualization across models on Solar-Energy dataset, with forecast horizon 96.

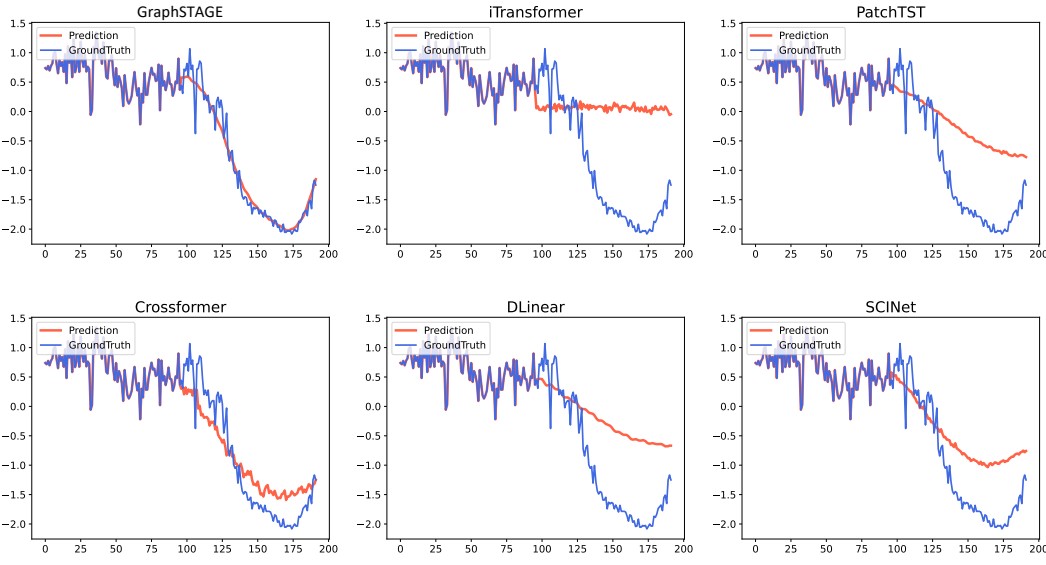

Figure 12: Sample visualization across models on PEMS07 dataset, with forecast horizon 96.

# F ROBUSTNESS EXPERIMENTS

In this section, we present the standard deviations of GRAPHSTAGE across all multivariate time series forecasting tasks, as shown in Table 11 and Table 10. These results were obtained using five random seeds.

Additionally, we conducted a separate set of robustness tests with varying input lengths. The results of these experiments are presented in Figure 13. Experiments were performed on the ETTm1 and ECL datasets, with each configuration run 10 times to assess whether the results remained stable within a consistent range. Furthermore, as the model is able to leverage longer historical input data, both the MSE and MAE consistently decreased. The reductions in MSE and MAE are most significant when the input length increases from 48 to 96. These findings highlight the model's strong robustness and its effective capability in extracting intra-series (temporal) correlations.

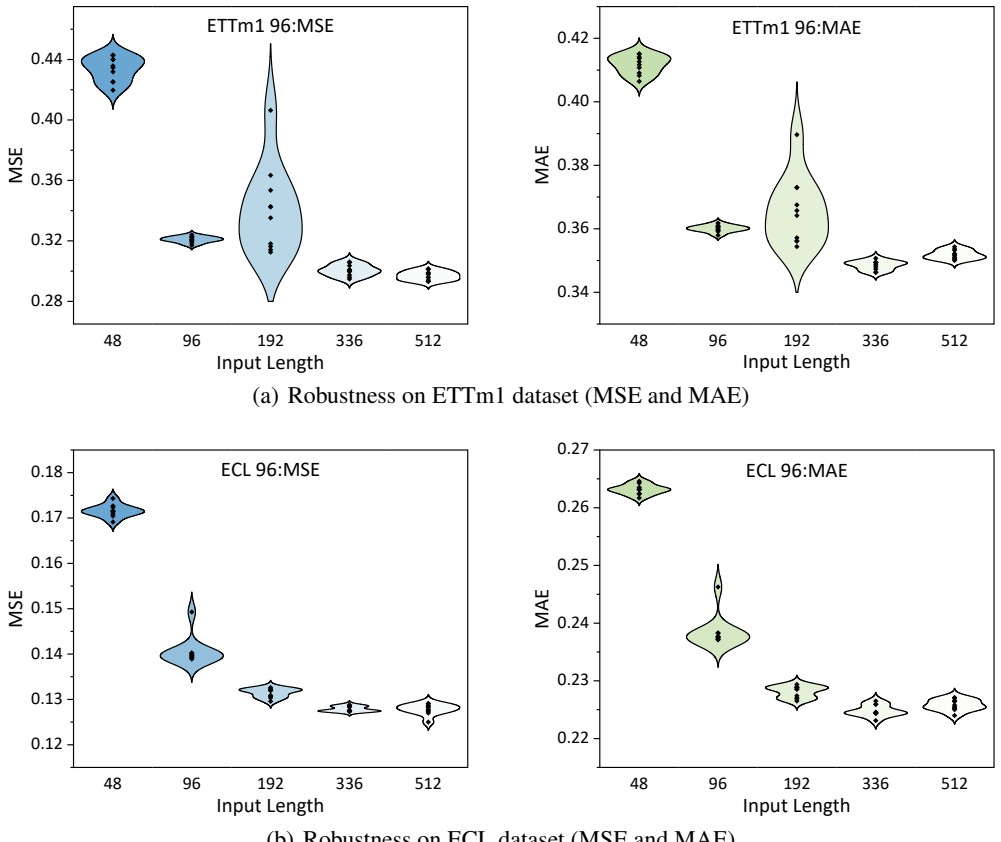

(a) Robustness on ETTm1 dataset (MSE and MAE)

(b) Robustness on ECL dataset (MSE and MAE)

Figure 13: Robustness Experiments with increasing input lengths: $\{48, 96, 192, 336, 512\}$, and fixed output length: 96.

Table 10: Standard deviations of GRAPHSTAGE on 9 time series datasets for long-term forecasting tasks. The results are obtained from five random seeds.

| Dataset | ETTm1 | | ETTm2 | | ETTh1 | |
|---|---|---|---|---|---|---|
| Horizon | MSE | MAE | MSE | MAE | MSE | MAE |
| 96 | 0.319±0.004 | 0.356±0.003 | 0.174±0.002 | 0.259±0.001 | 0.384±0.003 | 0.395±0.007 |
| 192 | 0.367±0.002 | 0.381±0.002 | 0.241±0.007 | 0.304±0.006 | 0.435±0.007 | 0.426±0.005 |
| 336 | 0.394±0.003 | 0.400±0.002 | 0.301±0.001 | 0.341±0.000 | 0.476±0.005 | 0.441±0.001 |
| 720 | 0.482±0.005 | 0.441±0.002 | 0.397±0.005 | 0.398±0.002 | 0.487±0.003 | 0.460±0.003 |
| Dataset | ETTh2 | | ECL | | Exchange | |
| Horizon | MSE | MAE | MSE | MAE | MSE | MAE |
| 96 | 0.292±0.002 | 0.341±0.002 | 0.139±0.001 | 0.237±0.001 | 0.084±0.003 | 0.203±0.004 |
| 192 | 0.380±0.008 | 0.395±0.004 | 0.155±0.004 | 0.251±0.002 | 0.186±0.002 | 0.306±0.002 |
| 336 | 0.424±0.006 | 0.431±0.007 | 0.175±0.003 | 0.272±0.002 | 0.339±0.003 | 0.420±0.002 |
| 720 | 0.453±0.004 | 0.459±0.002 | 0.196±0.005 | 0.292±0.004 | 0.898±0.014 | 0.710±0.012 |
| Dataset | Traffic | | Weather | | Solar-Energy | |
| Horizon | MSE | MAE | MSE | MAE | MSE | MAE |
| 96 | 0.438±0.005 | 0.281±0.007 | 0.159±0.001 | 0.208±0.001 | 0.172±0.003 | 0.258±0.004 |
| 192 | 0.442±0.005 | 0.282±0.002 | 0.207±0.001 | 0.251±0.001 | 0.183±0.002 | 0.259±0.001 |
| 336 | 0.461±0.004 | 0.292±0.004 | 0.263±0.001 | 0.292±0.001 | 0.205±0.003 | 0.278±0.005 |
| 720 | 0.509±0.006 | 0.322±0.007 | 0.344±0.001 | 0.345±0.001 | 0.211±0.001 | 0.273±0.001 |

Table 11: Standard deviations of GRAPHSTAGE on the PEMS forecasting tasks. The results are obtained from five random seeds.

| Dataset | PEMS03 | | PEMS04 | | PEMS07 | | PEMS08 | |
|---|---|---|---|---|---|---|---|---|
| Horizon | MSE | MAE | MSE | MAE | MSE | MAE | MSE | MAE |
| 12 | 0.065±0.002 | 0.170±0.002 | 0.070±0.001 | 0.174±0.002 | 0.056±0.001 | 0.152±0.001 | 0.085±0.007 | 0.175±0.006 |
| 24 | 0.082±0.004 | 0.193±0.004 | 0.082±0.001 | 0.190±0.001 | 0.072±0.001 | 0.175±0.003 | 0.111±0.002 | 0.205±0.003 |
| 48 | 0.106±0.005 | 0.219±0.005 | 0.096±0.005 | 0.207±0.005 | 0.087±0.007 | 0.179±0.004 | 0.155±0.010 | 0.230±0.009 |
| 96 | 0.136±0.007 | 0.253±0.005 | 0.113±0.004 | 0.228±0.003 | 0.105±0.005 | 0.209±0.006 | 0.207±0.006 | 0.270±0.007 |

# G   FULL RESULTS OF ABLATION STUDY

In this section, we provide the detail results of our ablation studies to offer deeper insights into the effectiveness of each component in GRAPHSTAGE. Table 12 displays the full results of the ablation study on the Correlation Learning Mechanism for each prediction length. The experiments include both component removal (w/o) and component replacement (Replace) using the attention mechanism from Crossformer (Zhang & Yan, 2023). Detailed results of the ablation study on the Embedding & Patching mechanism are presented in Table 13. We investigate the impact of removing the Patching module (w/o Patching), the Time Embedding (w/o Time Embedding), and the Adaptive Embedding (w/o Adaptive Embedding) individually.

The performance degradation observed in all ablated variants across different prediction lengths underscores the significant role of the components in GRAPHSTAGE. Our comprehensive ablation studies confirm that each component contributes to the model's overall performance. The Inter-GrAG and Intra-GrAG modules are essential for learning spatial and temporal dependencies, while the Embedding & Patching mechanism effectively incorporates prior knowledge. These findings underscore the importance of each design in GRAPHSTAGE, collectively leading to its superior performance in MTSF tasks.

Table 12: Full Results of Ablation Study on Correlation Learning Mechanism. The input sequence length is set to 96. *AVG* means the average results from all four prediction lengths.

| Design | Spatial | Temporal | Prediction Lengths | ETTm1 MSE | ETTm1 MAE | ECL MSE | ECL MAE | Traffic MSE | Traffic MAE | Solar-Energy MSE | Solar-Energy MAE |
|---|---|---|---|---|---|---|---|---|---|---|---|
| GRAPHSTAGE | Inter-GrAG | Intra-GrAG | 96 | 0.319 | 0.356 | 0.139 | 0.237 | 0.438 | 0.281 | 0.172 | 0.258 |
| | | | 192 | 0.367 | 0.381 | 0.155 | 0.251 | 0.442 | 0.282 | 0.183 | 0.259 |
| | | | 336 | 0.394 | 0.400 | 0.175 | 0.272 | 0.461 | 0.292 | 0.205 | 0.278 |
| | | | 720 | 0.482 | 0.441 | 0.196 | 0.292 | 0.509 | 0.322 | 0.211 | 0.273 |
| | | | AVG | **0.391** | **0.394** | **0.166** | **0.263** | 0.462 | **0.294** | **0.192** | 0.267 |
| w/o | Inter-GrAG | w/o | 96 | 0.319 | 0.360 | 0.160 | 0.253 | 0.455 | 0.308 | 0.177 | 0.268 |
| | | | 192 | 0.377 | 0.389 | 0.168 | 0.260 | 0.452 | 0.292 | 0.223 | 0.297 |
| | | | 336 | 0.410 | 0.410 | 0.183 | 0.276 | 0.475 | 0.307 | 0.226 | 0.301 |
| | | | 720 | 0.486 | 0.441 | 0.230 | 0.318 | 0.529 | 0.340 | 0.274 | 0.300 |
| | | | AVG | 0.398 | 0.400 | 0.185 | 0.277 | 0.478 | 0.312 | 0.225 | 0.292 |
| | w/o | Intra-GrAG | 96 | 0.328 | 0.364 | 0.167 | 0.257 | 0.488 | 0.307 | 0.241 | 0.305 |
| | | | 192 | 0.374 | 0.386 | 0.169 | 0.259 | 0.515 | 0.320 | 0.226 | 0.282 |
| | | | 336 | 0.398 | 0.404 | 0.190 | 0.279 | 0.495 | 0.308 | 0.245 | 0.299 |
| | | | 720 | 0.496 | 0.448 | 0.219 | 0.307 | 0.536 | 0.343 | 0.243 | 0.291 |
| | | | AVG | 0.399 | 0.400 | 0.186 | 0.276 | 0.509 | 0.320 | 0.239 | 0.294 |
| Replace | Inter-GrAG | Attention | 96 | 0.323 | 0.363 | 0.143 | 0.240 | 0.448 | 0.286 | 0.183 | 0.259 |
| | | | 192 | 0.374 | 0.388 | 0.165 | 0.258 | 0.462 | 0.297 | 0.210 | 0.276 |
| | | | 336 | 0.401 | 0.410 | 0.170 | 0.267 | 0.468 | 0.299 | 0.208 | 0.272 |
| | | | 720 | 0.481 | 0.443 | 0.195 | 0.296 | 0.533 | 0.329 | 0.221 | 0.274 |
| | | | AVG | 0.395 | 0.401 | 0.168 | 0.265 | 0.478 | 0.303 | 0.206 | 0.270 |
| | Attention | Inter-GrAG | 96 | 0.339 | 0.373 | 0.144 | 0.242 | 0.436 | 0.305 | 0.177 | 0.256 |
| | | | 192 | 0.375 | 0.389 | 0.161 | 0.257 | 0.445 | 0.291 | 0.206 | 0.278 |
| | | | 336 | 0.414 | 0.413 | 0.176 | 0.274 | 0.461 | 0.298 | 0.225 | 0.288 |
| | | | 720 | 0.486 | 0.449 | 0.202 | 0.297 | 0.494 | 0.325 | 0.214 | 0.280 |
| | | | AVG | 0.403 | 0.406 | 0.171 | 0.268 | 0.459 | 0.305 | 0.206 | 0.276 |
| | Attention | Attention | 96 | 0.316 | 0.361 | 0.144 | 0.243 | 0.414 | 0.284 | 0.181 | 0.253 |
| | | | 192 | 0.385 | 0.398 | 0.160 | 0.257 | 0.444 | 0.292 | 0.205 | 0.265 |
| | | | 336 | 0.393 | 0.410 | 0.177 | 0.276 | 0.461 | 0.298 | 0.210 | 0.268 |
| | | | 720 | 0.486 | 0.447 | 0.203 | 0.299 | 0.494 | 0.325 | 0.218 | 0.270 |
| | | | AVG | 0.395 | 0.404 | 0.171 | 0.269 | **0.453** | 0.300 | 0.204 | **0.264** |

Table 13: Full Results of Ablation Study on Embedding&Patching Mechanism. The input sequence length is set to 96. *AVG* means the average results from all four prediction lengths.

| Design | Prediction Lengths | PEMS03 | | PEMS04 | | PEMS07 | | PEMS08 | |
|---|---|---|---|---|---|---|---|---|---|
| | | MSE | MAE | MSE | MAE | MSE | MAE | MSE | MAE |
| GRAPHSTAGE | 12 | 0.065 | 0.170 | 0.070 | 0.174 | 0.056 | 0.152 | 0.085 | 0.175 |
| | 24 | 0.082 | 0.193 | 0.082 | 0.190 | 0.072 | 0.175 | 0.111 | 0.205 |
| | 48 | 0.106 | 0.219 | 0.096 | 0.207 | 0.087 | 0.179 | 0.155 | 0.230 |
| | 96 | 0.136 | 0.253 | 0.113 | 0.228 | 0.105 | 0.209 | 0.207 | 0.270 |
| | **AVG** | **0.097** | **0.210** | **0.090** | **0.200** | **0.080** | **0.179** | **0.139** | **0.220** |
| w/o Patching | 12 | 0.071 | 0.179 | 0.075 | 0.183 | 0.058 | 0.157 | 0.105 | 0.191 |
| | 24 | 0.091 | 0.205 | 0.089 | 0.202 | 0.078 | 0.181 | 0.127 | 0.216 |
| | 48 | 0.118 | 0.231 | 0.103 | 0.217 | 0.101 | 0.203 | 0.175 | 0.258 |
| | 96 | 0.160 | 0.272 | 0.134 | 0.259 | 0.146 | 0.257 | 0.295 | 0.348 |
| | **AVG** | 0.110 | 0.222 | 0.100 | 0.215 | 0.096 | 0.199 | 0.176 | 0.253 |
| w/o Time Emb. | 12 | 0.070 | 0.177 | 0.071 | 0.176 | 0.063 | 0.162 | 0.108 | 0.199 |
| | 24 | 0.093 | 0.203 | 0.088 | 0.198 | 0.077 | 0.179 | 0.179 | 0.254 |
| | 48 | 0.132 | 0.242 | 0.115 | 0.231 | 0.095 | 0.201 | 0.195 | 0.269 |
| | 96 | 0.162 | 0.269 | 0.122 | 0.239 | 0.128 | 0.231 | 0.315 | 0.335 |
| | **AVG** | 0.114 | 0.223 | 0.099 | 0.211 | 0.091 | 0.193 | 0.199 | 0.264 |
| w/o Adaptive Emb. | 12 | 0.071 | 0.180 | 0.076 | 0.185 | 0.060 | 0.161 | 0.100 | 0.188 |
| | 24 | 0.089 | 0.304 | 0.086 | 0.196 | 0.079 | 0.184 | 0.116 | 0.206 |
| | 48 | 0.122 | 0.239 | 0.107 | 0.220 | 0.131 | 0.239 | 0.166 | 0.250 |
| | 96 | 0.202 | 0.306 | 0.125 | 0.242 | 0.194 | 0.299 | 0.429 | 0.395 |
| | **AVG** | 0.121 | 0.257 | 0.098 | 0.211 | 0.116 | 0.221 | 0.203 | 0.260 |

# H  VISUALIZATION OF TEMPORAL AND SPATIAL LEARNABLE GRAPHS.

GRAPHSTAGE is a fully graph-based model that decouples the learning of inter-series (spatial) and intra-series (temporal) dependencies. Consequently, it can generate two learnable graphs in the spatial and temporal dimensions, respectively.

Figure 14 presents additional visualizations of the Temporal Learnable Graphs ($A_T$). Each column displays a randomly selected $A_T$ from the results of GRAPHSTAGE, with experiments conducted on the ETTm1, ECL, and PEMS04 datasets.

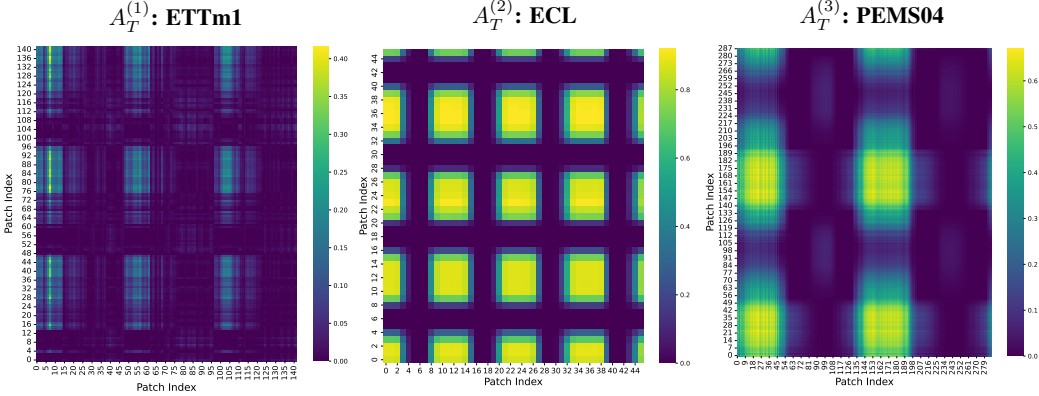

Figure 14: Supplementary visualization of Temporal Learnable Graphs ($A_T$) across datasets (ETTm1, ECL, PEMS04). Each column represents a randomly selected $A_T$ from the results of GRAPHSTAGE.

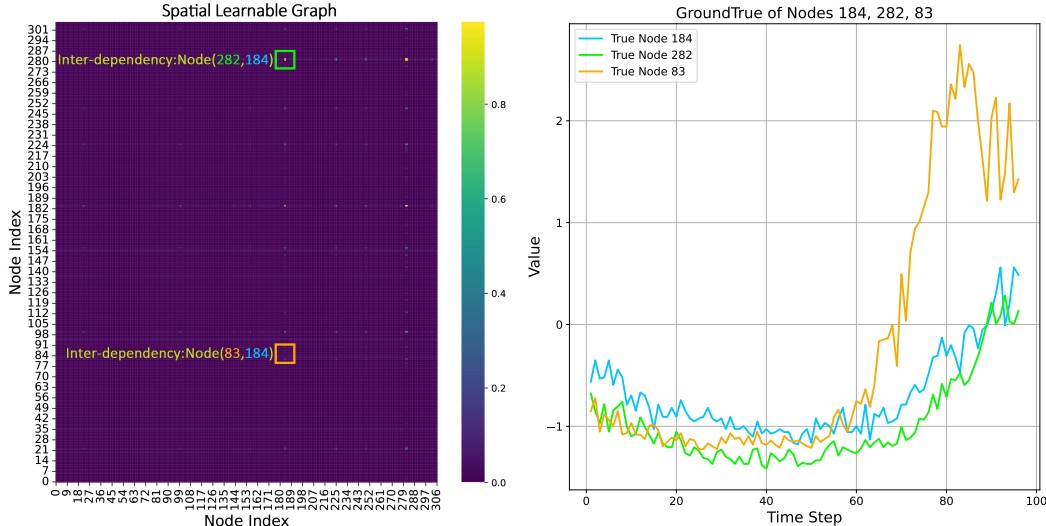

Figure 15: A randomly selected sample of the Spatial Learnable Graph ($A_S$) on the PEMS04 dataset (left), along with the corresponding GroundTruth of the nodes (right).

Figure 15 presents a randomly selected sample of the Spatial Learnable Graph ($A_S$) along with the corresponding ground truth of the nodes.

In $A_S$, we observe that nodes 184 and 282 exhibit a high correlation—as indicated by a bright spot within the green square in Figure 15 (left), representing a correlation coefficient close to 1. Conversely, nodes 184 and 83 show almost zero correlation—there is no bright spot within the orange square in Figure 15 (left), indicating a correlation coefficient close to 0. Correspondingly, as shown in Figure 15 (right), the ground truth for nodes 184 and 282 behaves very similarly, whereas node 83 displays completely different trends.

This randomly selected visualization demonstrates that the correlations among nodes in $A_S$ learned by the Intra-GrAG module match the ground truth, confirming the effectiveness of GRAPHSTAGE in capturing inter-series dependencies.

