# OpenReview forum: "GraphSTAGE: Channel-Preserving Graph Neural Networks for Time Series Forecasting"
_ICLR.cc/2025/Conference — Submitted to ICLR 2025_

### Official Review · Reviewer_RvJQ · 2024-10-28

**Soundness:** 3
**Presentation:** 3
**Contribution:** 2
**Rating:** 6
**Confidence:** 3

**Summary:**

The paper presents GraphSTAGE, a novel GNN-based model for multivariate time series forecasting (MTSF). By separating the learning of temporal (intra-series) and spatial (inter-series) dependencies while preserving the original channel structures, GraphSTAGE minimizes noise and computational overhead caused by channel mixing. It incorporates the Spatial-Temporal Aggregation Graph Encoder (STAGE), which enhances interpretability by visualizing temporal and spatial patterns. Experiments across 13 real-world datasets demonstrate that GraphSTAGE outperforms state-of-the-art models by up to 20.8%.

**Strengths:**

Overall, good empirical result: the proposed model outperforms state-of-the-art models by up to 20.8%. The experiments in this paper focus on evaluating the proposed GRAPHSTAGE model for multivariate time series forecasting (MTSF). Here’s a summary of the experiments and results:

Experimental Setup
* Datasets: The model is tested on 13 real-world datasets, including ETT, ECL, Exchange, Traffic, Weather, Solar-Energy, and PEMS (with subsets PEMS03, PEMS04, PEMS07, and PEMS08). These datasets represent diverse MTSF tasks, covering domains like energy, weather, and traffic forecasting. See Section 4.1 in Page 6.

* Baselines: Seven well-known models, including iTransformer, PatchTST, Crossformer, DLinear, RLinear, SCINet, and TimesNet, are used for comparison​. See Section 4.1 in Page 6.

Main Results
* Overall Performance: GRAPHSTAGE demonstrates up to a 20.8% improvement in performance and achieves first place in 22 out of 30 comparisons across all datasets. It achieves lower Mean Squared Error (MSE) and Mean Absolute Error (MAE) compared to state-of-the-art (SOTA) models, particularly excelling in datasets like ECL, ETT, Weather, Solar-Energy, and PEMS​. See Section 4.2 in Pages 6 and 7.

* Model Efficiency: Compared to models like Crossformer, GRAPHSTAGE reduces memory usage by 47.0% and training time by 60.9%, while achieving a 36.5% improvement in predictive accuracy. See Page 8, Lines from 378 to 385.

Ablation Studies
* Correlation Learning Mechanism: Ablations reveal that removing or replacing the Inter-GrAG and Intra-GrAG modules degrades performance, confirming the importance of the decoupled spatial-temporal extraction in GRAPHSTAGE​. See Section 4.3 and Table 2 in Page 8.

* Embedding & Patching Mechanism: Ablations also show that removing patching or adaptive embedding reduces accuracy, indicating the critical role of these components in effective forecasting. See Section 4.3 and Table 3 in Page 8.

**Weaknesses:**

Incremental idea & Lack of novelty: I think the core idea of decoupling the learning of intra-series and inter-series instead of channel mixing is incremental and not significant.

* By decoupling intra-series and inter-series dependencies, GRAPHSTAGE enables more efficient modeling without the interference and noise associated with channel blending. See Section 3.2, Lines from 228 to 242 in Page 5.

* GRAPHSTAGE’s decoupled architecture reduces computational overhead and enhances model interpretability by generating separate learnable graphs for temporal and spatial dimensions​. See Section 3.2, Lines from 243 to 266 in Page 5.

**Questions:**

Please address the concern regarding novelty that I have mentioned in weaknesses!

---

> ### Author Response · Authors · 2024-11-21
>
> # Response to Reviewer RvJQ (1/1)
>
> > **W1 ＆ Q1 : Incremental idea & Lack of novelty: I think the core idea of decoupling the learning of intra-series and inter-series instead of channel mixing is incremental and not significant.**
>
>
>
> Thank you for pointing out this. We would like to highlight the unique aspects of our approach and clarify the distinctions between our work and the recent models (iTransformer [1] and FourierGNN [2]).
>
> 1. **Novelty in Research Motivation:**
>    We are the first to study the trade-off between **the variety of dependencies extracted** and **the noise that may be introduced by channel-mixing**. As we stated in our introduction: *"Is it truly necessary to model all these dependencies?"*      Previous works [2] [3] often adopt a "brute-force modeling" approach, stacking parameters to capture as many dependencies as possible. While this might seem effective, it overlooks a critical issue—the potential noise introduced by this process.
> 2. **Innovation in Model Framework:**
>    We propose a **novel channel-preserving framework**: GraphSTAGE. Through fair model variant experiments in *Table 4 on Page 10*, we validate the presence of such noise, underscoring the limitations of excessive dependency extraction. GraphSTAGE is a pure graph paradigm that decouples the extraction of **global** **temporal** dependencies and **global** **spatial** dependencies, **rather than being limited to local** neighbor information.
> 3. **Advancements in Model Performance:**
>    Despite its structural simplicity, our model performs comparably to or surpasses state-of-the-art models across 13 MTSF benchmark datasets. It **ranks first among 8 advanced models** in 22 out of 30 comparisons in *Table 1 on Page 7*. Specifically, on the PEMS07 dataset—which has the largest number of nodes—GraphSTAGE outperforms the recent SOTA iTransformer by 20.8%, indicating its **potential for application to larger-scale MTSF tasks**, such as extensive grid management.
>
> ---
>
> **$\triangleright$Differences with iTransformer:**
>
> Conceptually, **iTransformer** is not a channel-preserving model; it **overlooks temporal channel information.** iTransformer projects the original time series data $X_{\text{in}} \in \mathbb{R}^{N \times T}$ (where $N$ is the number of nodes and $T$ is the length of the time series) into $H_S \in \mathbb{R}^{N \times D}$ to capture spatial dependencies among nodes. However, this transformation ignores temporal dependencies and fails to learn the underlying temporal graph structures.
>
> In contrast, our GraphSTAGE embeds the input data $X_{\text{in}} \in \mathbb{R}^{N \times T}$ into $H \in \mathbb{R}^{N \times T \times D}$, where $D$ is the embedding dimension. The original node and time dimensions are preserved. This channel-preserving framework enables the model to incorporate both spatial (inter-series) and temporal (intra-series) dependencies by decoupling them. This separation allows GraphSTAGE to capture temporal dynamics more effectively, which is a significant extension beyond what iTransformer offers.

---

> ### Author Response · Authors · 2024-11-21
>
> **$\triangleright$Differences with FourierGNN:**
>
> Although both are pure graph methods, Graph Neural Networks (GNNs) are a very broad concept. **Being graph-based does not make our work incremental.**
>
> 1. **Dependency Modeling Approach:**
>    **FourierGNN** represents a trend toward modeling **three kinds of dependencies** within a unified framework, as shown in *Figure 2 on Page 2*. **GraphSTAGE**, however, focuses on decoupling and **modeling two types of dependencies.**
>
> 2. **Research Focus:**
>    The drawbacks of using hypervariate graphs to unify multi-dependency modeling have not been studied in previous research. There are two main drawbacks: **High Memory Consumption:** This issue has also been reported in other studies [4]. **Potential Noise:** Modeling extensive cross-dependencies may lead the model to learn many pseudo connections (as we explained in our response to Q1). These pseudo connections can weaken the influence of genuine connections (inter-series and intra-series dependencies), potentially diminishing the model's predictive performance.
>
>    Additionally, instead of focusing solely on FourierGNN, we aimed to broadly address hypervariate graph-based multi-dependency modeling methods. In the Variants Comparison section, we included a variant called **VarC** (shown in *Figure 7 on Page 9*) to encompass these methods and compared it with our channel-preserving GraphSTAGE. Our findings indicate that **modeling only two types of dependencies can achieve excellent predictive performance while also reducing memory usage.**
>
> ---
>
> **$\triangleright$Addition of Graph-Based Baselines:**
>
> To **provide a more direct and comprehensive comparison with FourierGNN** and other graph-based models, we have included four additional graph-based baselines in the revised paper. The detailed results are presented in *Table 9 on Page 18* of the revised paper. For your convenience, we provide a summary of the relevant averaged results below:
>
> | **Model**          | GraphSTAGE          |                     | FourierGNN[2]   |                 | CrossGNN[5] |       | StemGNN[6]      |                 | MTGNN[7] |       |
> | ------------------ | ------------------- | ------------------- | --------------- | --------------- | ----------- | ----- | --------------- | --------------- | -------- | ----- |
> | Metric             | MSE                 | MAE                 | MSE             | MAE             | MSE         | MAE   | MSE             | MAE             | MSE      | MAE   |
> | ECL (321 Nodes)    | **0.166**$\pm$0.003 | **0.263**$\pm$0.002 | 0.221$\pm$0.001 | 0.318$\pm$0.001 | 0.201       | 0.300 | 0.215$\pm$0.002 | 0.316$\pm$0.002 | 0.251    | 0.347 |
> | ETTm1 (7 Nodes)    | **0.391**$\pm$0.003 | **0.394**$\pm$0.002 | 0.453$\pm$0.002 | 0.448$\pm$0.003 | 0.393       | 0.404 | 0.550$\pm$0.004 | 0.537$\pm$0.005 | 0.469    | 0.446 |
> | ETTh2 (7 Nodes)    | **0.387**$\pm$0.005 | **0.407**$\pm$0.004 | 0.543$\pm$0.006 | 0.517$\pm$0.004 | 0.393       | 0.418 | 1.158$\pm$0.010 | 0.812$\pm$0.011 | 0.465    | 0.509 |
> | Weather (21 Nodes) | **0.243**$\pm$0.001 | **0.274**$\pm$0.001 | 0.257$\pm$0.002 | 0.305$\pm$0.003 | 0.247       | 0.289 | 0.289$\pm$0.005 | 0.342$\pm$0.007 | 0.314    | 0.355 |
>
> The results indicate that GraphSTAGE achieved the **top-1** rank in terms of MSE and MAE metrics across four datasets **compared with four graph-based models.** Although CrossGNN performs comparably to our GraphSTAGE on smaller datasets (e.g., ETT and Weather), GraphSTAGE notably outperforms CrossGNN on large-scale datasets (e.g., ECL with 321 nodes). These findings further validate the effectiveness of GraphSTAGE in capturing both intra-series and inter-series dependencies, leading to **superior forecasting accuracy across datasets of varying scales.**
>
> We believe these clarifications address your concerns regarding the novelty and contributions of our work. Current models primarily focus on the advantages of channel-mixing methods for extracting multiple dependencies, often **neglecting the noise these approaches can introduce.** **We are the first to directly address this issue.** Thank you again for your valuable feedback, which has helped us improve our paper.
>
> ---
>
> [1] *iTransformer: Inverted Transformers Are Effective for Time Series Forecasting.* ICLR, 2024
>
> [2] *FourierGNN: Rethinking multivariate time series forecasting from a pure graph perspective.* NeurIPS, 2024
>
> [3] *UniTST: Effectively Modeling Inter-Series and Intra-Series Dependencies for Multivariate Time Series Forecasting.* arXiv
>
> [4] *ForecastGrapher: Redefining Multivariate Time Series Forecasting with Graph Neural Networks.* arXiv
>
> [5] *Crossgnn: Confronting noisy multivariate time series via cross interaction refinement*. NeurIPS, 2023
>
> [6] *Spectral temporal graph neural network for multivariate time-series forecasting.* NeurIPS, 2020
>
> [7] *Connecting the dots: Multivariate time series forecasting with graph neural networks.* KDD, 2020

---

> > ### Comment · Reviewer_RvJQ · 2024-11-25
> > **Upgrade the score**
> >
> > Dear Authors,
> >
> > To be honest, I am still not convinced that your idea of decoupling the learning of intra-series and inter-series is significant. I don't really see it from the rebuttal.
> >
> > However, since the authors updated with more experimental results and the new results look reasonably good, so I updated the score to 6: marginally accepted.
> >
> > Best,
> > Reviewer

---

> > > ### Author Response · Authors · 2024-12-03
> > >
> > > Thank you for your positive feedback and for taking the time to carefully review our responses. We deeply appreciate your thoughtful evaluation and support!

---

> ### Author Response · Authors · 2024-11-24
> **Request of Reviewer's attention and feedback**
>
> Thank you for your valuable and constructive feedback, which has inspired further improvements to our paper. As a gentle reminder, it has been more than 2 days since we submitted our rebuttal. We would like to know whether our response addressed your concerns.
>
> Following your comments and suggestions, we have **answered your concerns** and **made the following revisions**:
>
> - To address your concerns, we have clarified the unique aspects of our approach, particularly highlighting its **motivation and framework**. Additionally, we included a **detailed comparison with recent models**. To further enhance the validity of our work, we added four graph-based baselines.
> - We have expanded our experiments to **include additional four graph-based baseline methods** for multivariate time series forecasting tasks: (1) FourierGNN, (2) CrossGNN, (3) StemGNN, and (4) MTGNN. For complete results, refer to *Table 9 in Appendix D on Page 18*.
>
> Once again, we sincerely thank you for your insightful review. We look forward to your feedback and are prepared to address any further questions or concerns you may have.

---

> ### Author Response · Authors · 2024-11-25
> **Request of Reviewer's attention and feedback**
>
> Dear Reviewer RvJQ,
>
> We sincerely thank you for your valuable and constructive feedback. Since the End of author/reviewer discussions is coming soon, may we know if our response addresses your main concerns? If so, we kindly ask for your reconsideration of the score. Should you have any further advice on the revised paper and/or our rebuttal, please let us know and we will be more than happy to engage in more discussion and paper improvements.
>
> Thank you so much for devoting time to improving our paper!

---

### Official Review · Reviewer_7NU8 · 2024-10-31

**Soundness:** 2
**Presentation:** 2
**Contribution:** 2
**Rating:** 5
**Confidence:** 4

**Summary:**

This study focuses on graph structure learning for time series forecasting based on modeling inter-series and intra-series dependencies for complex multivariate time series. To enable efficient modeling of dependencies with controlled interference between variables and reduced noise, the authors propose GraphStage architecture, a GNN-based approach that, contrary to previous approaches, is channel-preserving. To achieve this, the GraphStage model is built upon patching and embedding layers, followed by decoupled inter-series and intra-series correlation graph structure modules that produce spatial and temporal graphs fine-tuned with the task. Finally, GraphStage is evaluated for time series forecasting against recent state-of-the-art architectures, showing competitive performance. Several ablation studies and visualizations of the learnable correlation provide additional insights into the potential of the proposed method.

**Strengths:**

The authors tackle a long-standing problem in time series forecasting, which is inferring a graph capturing evolving dependencies in terms of multiple variables. The main strengths of this work and the proposed method are summarized as follows:
- The authors combine embedding and patching mechanisms recently proposed for time series forecasting with the challenging task of graph structure learning, enabling channel independence.
- Correlations are captured in terms of a graph structure separately for the spatial and temporal dimensions, which remains a relatively novel design choice.
- Experimental results show that the proposed graph-based method outcompetes, in several cases, recent baselines in time series forecasting. Additionally, the visualizations of the learned temporal correlations are very interesting qualitative aspects of the presented study.

**Weaknesses:**

1. **Poor Experimental Evaluation and Unclear Performance Significance:** (W1) Chosen baseline methods for comparisons are limited to non-graph-based models. In contrast, several methods leveraging GNNs for time series forecasting can be found in the literature [1]. (W2) Additionally, the performance improvements achieved by the proposed method are, in most cases, very small compared to the best competitor (for instance, the biggest difference achieved by GraphStage from its best competitor in Table 1 is for Solar-Energy: from MSE equal to 0.233 to MSE 0.192, yet GraphStage is outcompeted in terms of MAE). (W3) It is unclear if the authors have performed multiple runs with random seeds to capture the model’s performance variability. The lack of standard deviations makes it impossible to assess whether the results' difference is statistically significant.
2. **Limited Related Works Presented:** (W4) Graph structure learning for time series forecasting is a long-term studied problem in the literature, entirely before the pure graph paradigm (Yi et al., 2024). Several methods that capture dependencies in the spatial and temporal domain jointly (Shang et al., 2021; Wu et al., 2020) or separately (Kipf et al., 2018; Xu et al., 2023) have been proposed, but are rather not mentioned in the paper and not considered in the experiments. (W5) Similarly the graph learning module (softmax on dot products between pairs) is a standard choice in relevant works (Wu et al., 2020), yet not correctly cited in this work.
3. **Lack of Clarity in the Methodological Presentation and Inflated Claims:** (W6) Section 3 is not very easy to follow since it has a great amount of text, making it hard to understand how the two graph learning blocks interact and how the final output is derived and optimized. (W7) Some claims are slightly exaggerated, e.g., that the proposed components “effectively capture intra-series and inter-series dependencies, respectively, while generating interpretable correlation graphs”. Additionally, performance improvements are, in the average case, around 6% (and no improvement for four datasets), thus the statement “with improvements up to 20.8%” is slightly misleading.

[1] Yi, K., Zhang, Q., Fan, W., He, H., Hu, L., Wang, P., ... & Niu, Z. (2024). FourierGNN: Rethinking multivariate time series forecasting from a pure graph perspective. Advances in Neural Information Processing Systems, 36.

[2] Shang, C., Chen, J., & Bi, J. (2021). Discrete graph structure learning for forecasting multiple time series. arXiv preprint arXiv:2101.06861.

[3] Kipf, T., Fetaya, E., Wang, K. C., Welling, M., & Zemel, R. (2018, July). Neural relational inference for interacting systems. In International conference on machine learning (pp. 2688-2697). PMLR.

[4] Xu, N., Kosma, C., & Vazirgiannis, M. (2023, November). TimeGNN: Temporal Dynamic Graph Learning for Time Series Forecasting. In International Conference on Complex Networks and Their Applications (pp. 87-99). Cham: Springer Nature Switzerland.

[5] Wu, Z., Pan, S., Long, G., Jiang, J., Chang, X., & Zhang, C. (2020, August). Connecting the dots: Multivariate time series forecasting with graph neural networks. In Proceedings of the 26th ACM SIGKDD international conference on knowledge discovery & data mining (pp. 753-763).

**Questions:**

1. Based on (W1-W3), the experimental evaluation could be enhanced with comparisons with relevant graph-based modules and better capture the variability of scores in comparisons with the baselines.
2. Based on (W4, W5), the related works section could be improved and include differences between design choices in the graph structure learning component, including the incorporation of sparsity, binary or continuous edges, and spatial or temporal dependencies, such that the technical contribution of the work is positioned in a more evident and fair way.
3. Based on (W6), some parts could be replaced by equations (for instance, how the combination of spatial and temporal graphs is achieved and how the final output is derived) or references since several components are similarly used to other studies, e.g., patching. Similarly, some details about the model could be moved from the appendix to the main text.
4. Based on (W7), it is unclear how interpretability is achieved. Since patching is performed first, followed by temporal graph learning and then spatial graph learning, seems a bit exaggerated that interpretability is achieved, e.g., what can someone tell from the spatial graph concerning inter-series dependencies?

---

> ### Author Response · Authors · 2024-11-21
>
> # Response to Reviewer 7NU8 (1/4)
>
> > **W1 ＆ Q1 (1/3) :** **Chosen baseline methods for comparisons are limited to non-graph-based models. In contrast, several methods leveraging GNNs for time series forecasting can be found in FourierGNN.**
>
> Thank you for pointing this out, we have **added four graph-based models as baseline methods** for comparison: FourierGNN [1], CrossGNN [2], StemGNN [3], and MTGNN [4]. The detailed results are presented in *Table 9 on Page 18* of the revised paper.
>
> Following the experimental setup of TimesNet [5], we use a fixed lookback length $T = 96$ and set the prediction lengths $K \in \{96, 192, 336, 720\} $. We reproduced the results for FourierGNN and StemGNN, running each experiment five times. For CrossGNN and MTGNN, we collected the reported results from [3].
>
> For your convenience, a summary of the relevant (averaged) results is provided below:
>
> | **Model**          | GraphSTAGE          |                     | FourierGNN[1]   |                 | CrossGNN[2] |       | StemGNN[3]      |                 | MTGNN[4] |       |
> | ------------------ | ------------------- | ------------------- | --------------- | --------------- | ----------- | ----- | --------------- | --------------- | -------- | ----- |
> | Metric             | MSE                 | MAE                 | MSE             | MAE             | MSE         | MAE   | MSE             | MAE             | MSE      | MAE   |
> | ECL (321 Nodes)    | **0.166**$\pm$0.003 | **0.263**$\pm$0.002 | 0.221$\pm$0.001 | 0.318$\pm$0.001 | 0.201       | 0.300 | 0.215$\pm$0.002 | 0.316$\pm$0.002 | 0.251    | 0.347 |
> | ETTm1 (7 Nodes)    | **0.391**$\pm$0.003 | **0.394**$\pm$0.002 | 0.453$\pm$0.002 | 0.448$\pm$0.003 | 0.393       | 0.404 | 0.550$\pm$0.004 | 0.537$\pm$0.005 | 0.469    | 0.446 |
> | ETTh2 (7 Nodes)    | **0.387**$\pm$0.005 | **0.407**$\pm$0.004 | 0.543$\pm$0.006 | 0.517$\pm$0.004 | 0.393       | 0.418 | 1.158$\pm$0.010 | 0.812$\pm$0.011 | 0.465    | 0.509 |
> | Weather (21 Nodes) | **0.243**$\pm$0.001 | **0.274**$\pm$0.001 | 0.257$\pm$0.002 | 0.305$\pm$0.003 | 0.247       | 0.289 | 0.289$\pm$0.005 | 0.342$\pm$0.007 | 0.314    | 0.355 |
>
> The results indicate that GraphSTAGE achieved the **top-1** rank in terms of MSE and MAE metrics across four datasets **compared with four graph-based models.** Although CrossGNN performs comparably to our GraphSTAGE on smaller datasets (e.g., ETT and Weather), GraphSTAGE notably outperforms CrossGNN on large-scale datasets (e.g., ECL with 321 nodes). These findings further validate the effectiveness of GraphSTAGE in capturing both intra-series and inter-series dependencies, leading to **superior forecasting accuracy across datasets of varying scales.**
>
> [1] *FourierGNN: Rethinking multivariate time series forecasting from a pure graph perspective.* NeurIPS, 2024
>
> [2] *Crossgnn: Confronting noisy multivariate time series via cross interaction refinement*. NeurIPS, 2023
>
> [3] *Spectral temporal graph neural network for multivariate time-series forecasting.* NeurIPS, 2020
>
> [4] *Connecting the dots: Multivariate time series forecasting with graph neural networks.* KDD, 2020
>
> [5] *TimesNet: Temporal 2D-Variation Modeling for General Time Series Analysis.* ICLR, 2023

---

> ### Author Response · Authors · 2024-11-21
>
> > **W2 ＆ Q1 (2/3) : Additionally, the performance improvements achieved by the proposed method are, in most cases, very small compared to the best competitor (for instance, the biggest difference achieved by GraphStage from its best competitor in Table 1 is for Solar-Energy: from MSE equal to 0.233 to MSE 0.192, yet GraphStage is outcompeted in terms of MAE).**
>
> Thank you for pointing this out, GraphSTAGE significantly **outperforms recent SOTA** models on **datasets with evident periodicity.** For example, on the four subsets of the PEMS datasets, GraphSTAGE achieves an average improvement of 14.3%. Conversely, on datasets with less clear periodicity (e.g., **ETT**, **Exchange**), the performance improvement is comparatively smaller.
>
> **This phenomenon is commonly observed and has been discussed in previous studies such as BasicTS+ [1].** On datasets where the periodic patterns are prominent, complex models with larger capacities—like GraphSTAGE and iTransformer—tend to capture these periodicity more effectively, leading to superior predictive performance. In contrast, on datasets with unclear or weak periodicity, simpler MLP-based models like DLinear and RLinear often perform better. This is because complex models may overfit the noise in datasets lacking strong periodic signals, whereas simpler models are less prone to overfitting and may generalize better under these conditions.
>
> [1] *Exploring progress in multivariate time series forecasting: Comprehensive benchmarking and heterogeneity analysis.* IEEE Transactions on Knowledge and Data Engineering, 2024
>
>
>
> ---
>
> > **W3 ＆ Q1 (3/3) : It is unclear if the authors have performed multiple runs with random seeds to capture the model’s performance variability. The lack of standard deviations makes it impossible to assess whether the results' difference is statistically significant.**
>
> Thank you for pointing this out, Yes, **the results of GraphSTAGE's performance are obtained from five random seeds.** To provide a clearer picture of the model's performance stability, we **have added the standard deviations for each task** of all time series datasets in *Table 10 and Table 11 on Page 22* of the revised manuscript.
>
> For your convenience, we summarize the standard deviations of our approach and the second-best method (iTransformer) on four selected datasets below:
>
> | Model        | GraphSTAGE      |                 | iTransformer    |                 |
> | ------------ | --------------- | --------------- | --------------- | --------------- |
> | Dataset      | MSE             | MAE             | MSE             | MAE             |
> | ECL          | 0.166$\pm$0.003 | 0.263$\pm$0.002 | 0.178$\pm$0.002 | 0.270$\pm$0.003 |
> | ETTh2        | 0.387$\pm$0.005 | 0.407$\pm$0.004 | 0.383$\pm$0.002 | 0.407$\pm$0.001 |
> | Solar-Energy | 0.192$\pm$0.002 | 0.267$\pm$0.003 | 0.233$\pm$0.001 | 0.262$\pm$0.001 |
> | Weather      | 0.243$\pm$0.001 | 0.274$\pm$0.001 | 0.258$\pm$0.001 | 0.278$\pm$0.001 |

---

> ### Author Response · Authors · 2024-11-21
>
> # Response to Reviewer 7NU8 (2/4)
>
> > **W4 ＆ Q2 (1/2) :Graph structure learning for time series forecasting is a long-term studied problem in the literature, entirely before the pure graph paradigm (Yi et al., 2024). Several methods that capture dependencies in the spatial and temporal domain jointly (Shang et al., 2021; Wu et al., 2020) or separately (Kipf et al., 2018; Xu et al., 2023) have been proposed, but are rather not mentioned in the paper and not considered in the experiments.**
>
> Thank you for pointing this out, the works you mentioned are indeed solid contributions in the fields of spatio-temporal forecasting. We have **revised the** **Related Works** to include these references and have highlighted the additions in orange in the revised paper.
>
> Furthermore, we have selected two of the works you mentioned (FourierGNN and MTGNN), as additional baselines in our experiments. We tried our best effort to reproduce TimeGNN, however, we were unable to obtain reasonable results. Detailed results of the can be found in *Table 9 on Page 18* of the revised paper. For your convenience, a summary of the relevant (averaged) results is provided below:
>
> | **Model** | GraphSTAGE |           | FourierGNN |       | MTGNN |       |
> | --------- | ---------- | --------- | ---------- | ----- | ----- | ----- |
> | Metric    | MSE        | MAE       | MSE        | MAE   | MSE   | MAE   |
> | ECL       | **0.166**  | **0.263** | 0.221      | 0.318 | 0.251 | 0.347 |
> | ETTm1     | **0.391**  | **0.394** | 0.453      | 0.448 | 0.469 | 0.446 |
> | ETTh2     | **0.387**  | **0.407** | 0.543      | 0.517 | 0.465 | 0.509 |
> | Weather   | **0.243**  | **0.274** | 0.257      | 0.305 | 0.314 | 0.355 |
>
> ---
> > **W5 ＆ Q2 (2/2) :Similarly the graph learning module (softmax on dot products between pairs) is a standard choice in relevant works (Wu et al., 2020), yet not correctly cited in this work.**
>
> Thank you for pointing this out. Yes, applying softmax on dot products between pairs is a standard approach for aggregating time features to extract spatial similarities between nodes. **We did not emphasize this as our contribution, and we have added the citation in the revised paper.**
>
> Our main contribution lies in adopting a **channel-preserving framework** to decouple the learning of inter-series and intra-series dependencies, which not only outperforms the channel-mixing framework but also reduces memory usage. Employing a widely adopted method in [1] for information aggregation also **demonstrates the generality of our channel-preserving framework.**
>
> However, to the best of our knowledge, we are the first to **extend this method to aggregate all node features at each time step** to extract similarities specifically between time steps. For example, in the **Intra-GrAG** module, the input $H_{\text{in}} \in \mathbb{R}^{P \times N \times D}$ is aggregated along the node dimension \(N\) to obtain $E_{\text{src}} \in \mathbb{R}^{P \times c}$ and $E_{\text{tgt}} \in \mathbb{R}^{P \times c}$. Softmax is then applied to the dot products between these pairs to compute the similarities between time steps, resulting in the Temporal Learnable Graph $A_T \in \mathbb{R}^{P \times P}$. The effectiveness of this approach in extracting intra-dependencies has been validated through the ablation study in *Table 2* and the visualization in *Figure 8*.
>
> [1] *Connecting the dots: Multivariate time series forecasting with graph neural networks.* KDD, 2020

---

> ### Author Response · Authors · 2024-11-21
>
> # Response to Reviewer 7NU8 (3/4)
>
> > **W6 ＆ Q3 (1/3): how the two graph learning blocks interact?**
>
> The interaction between the two graph learning blocks, Intra-GrAG and Inter-GrAG, follows a **sequential structure** within each STAGE block. We have **provided the pseudo-code of GraphSTAGE in *Algorithm 1 on Page 15*** in the revised paper. For your convenience, a brief introduction to the STAGE block is provided below:
>
> Let $L$ denote the number of STAGE blocks.
>
> - Use $H^{l-1} \in \mathbb{R}^{N \times P \times D}$ to represent the input tensor to the $l$-th STAGE block.
> - Use $H_{\text{tem}}^{l-1} \in \mathbb{R}^{P \times N \times D}$ to represent the output of the Intra-GrAG module in the $l$-th STAGE block.
> - Use $H^{l} \in \mathbb{R}^{N \times P \times D}$ to represent the output of the Inter-GrAG module in the $l$​​-th STAGE block.
> - $H^{0} \in \mathbb{R}^{N \times P \times D}$ is the output of the Embedding\&Patching layer.
>
> The whole process of stacked STAGE blocks is as follows:
>
> 1. For $l=1$ to $L$:
> 2. ​	// Intra-GrAG module
> 3. ​	$ H_{tem}^{l-1} = IntraGrAG(H^{l-1})$
> 4. ​	// Inter-GrAG module
> 5. ​	$ H^{l} = InterGrAG(H_{tem}^{l-1})$
> 6. End loop.
>
> The final output $H^{L}$​ is the output of the stacked STAGE blocks.
>
>
>
> > **W6 ＆ Q3 (2/3): how the final output is derived and optimized?**
>
> We apologize for neglecting to address this point. For the prediction output, we have **provided the pseudo-code of GraphSTAGE** in *Algorithm 1 on Page 15* of the revised paper.
>
> Regarding model optimization, the parameters are iteratively updated until convergence by **minimizing the prediction loss** $\mathcal{L} \leftarrow \mathcal{L}(\hat{Y}, Y)$, which is the Mean Squared Error (MSE) loss used in our experiments. $\hat{Y} $ are the predictions corresponding to the ground truth $Y$.
>
>
>
> > **W6 ＆ Q3 (3/3): how the combination of spatial and temporal graphs is achieved?**
>
> There is **no combination between the spatial and temporal graphs** because we follow a sequential structure that decouples the extraction of temporal and spatial dependencies. This process is illustrated in the **Orig** of *Figure 7 on Page 9*. Specifically:
>
> - Temporal graphs, denoted as $A_T \in \mathbb{R}^{P \times P}$, are used to capture the learned temporal dependencies.
> - Spatial graphs, denoted as $A_S \in \mathbb{R}^{N \times N}$, are used to capture the learned spatial dependencies.
>
> The two types of graphs are not combined in our approach.

---

> ### Author Response · Authors · 2024-11-21
>
> # Response to Reviewer 7NU8 (4/4)
>
> > **W7 ＆ Q4 (1/2) : Performance improvements are, in the average case, around 6% (and no improvement for four datasets), thus the statement “with improvements up to 20.8%” is slightly misleading.**
>
> Thank you for pointing out this issue. We apologize for confusion caused by the statement "with improvements up to 20.8%." We have **revised our manuscript accordingly and highlighted the modifications.**
>
> The statement "improvements up to 20.8%" specifically refers to the performance on the PEMS07 dataset, which has the largest number of nodes (883 nodes) among all the datasets we evaluated. In the original manuscript, we used this result to emphasize GraphSTAGE's potential for larger-scale MTSF applications.
>
>
>
> > **W7 ＆ Q4 (2/2) : It is unclear how interpretability is achieved. Since patching is performed first, followed by temporal graph learning and then spatial graph learning, seems a bit exaggerated that interpretability is achieved, e.g., what can someone tell from the spatial graph concerning inter-series dependencies?**
>
> Thank you for pointing this out. Although we use patching in the Embedding\&Patching layer, GraphSTAGE remains a channel-preserving framework. **The patching operation merely increases the receptive field of each temporal channel and does not weaken the interpretability of the model**. For example, regarding the temporal learnable graph $A_T^{(2)}$ obtained on the ECL dataset in *Figure 8 on Page 10,* when the lookback length $L = 96$ (the sampling frequency of the ECL dataset is 1 hour, so $96 \times 1\,\text{hour} = 4\,\text{days}$, meaning each sample contains 4 days of data). Since we use a patch stride $s = 2$, the total number of patches is $P = L / s = 96 / 2 = 48$. From $A_T^{(2)}$, we observe that **similar temporal patterns appear every 12 patches (corresponding to 24 hours).** This indicates that **GraphSTAGE effectively captures the intrinsic periodicity of the data, which** **matches the daily periodicity of the ECL dataset.**
>
>
>
> Regarding the spatial graph, we apologize for the confusion caused. We have **revised the colors** in *Figure 15  on Page 25* and have **updated the relevant text** in the revised paper, As shown in Figure 15, we can observe:
>
> Firstly, the spatial learnable graph $A_S$ learned by GraphSTAGE is **sparser**, indicating that the model can **identify the most important nodes** in space and requires fewer inter-series correlations for predictions.
>
> Secondly, the spatial learnable graph $A_S$ captures connections between time series that exhibit strong similarities. For example, according to the randomly selected case in Figure 15(left), the $A_S$ considers that the similarity between nodes 282 and 184 is high (close to 1), while the similarity between nodes 83 and 184 is low (close to 0). We **plotted the ground truth** of these three nodes in Figure 15(right), which shows that **the trends of nodes 282 and 184 are consistent, matching the correlation coefficients learned in  $A_S$.** This match confirms the effectiveness of GraphSTAGE in capturing inter-series dependencies and enhances the interpretability of the model.

---

> ### Author Response · Authors · 2024-11-24
> **Request of Reviewer's attention and feedback**
>
> Thank you for your valuable and constructive feedback, which has inspired further improvements to our paper. As a gentle reminder, it has been more than 2 days since we submitted our rebuttal. We would like to know whether our response addressed your concerns.
>
> Following your comments and suggestions, we have **answered your concerns** and **made the following revisions**:
>
> - We have expanded our experiments to **include additional four graph-based baseline methods** for multivariate time series forecasting tasks: (1) FourierGNN, (2) CrossGNN, (3) StemGNN, and (4) MTGNN. For complete results, refer to *Table 9 in Appendix D on Page 18*.
>
> - We have **added the standard deviations** of our model across all tasks to highlight the model’s strong robustness. For complete results, refer to *Table 10 and Table 11 in Appendix F on Page 22*.
>
> - Additionally, we **provided the pseudo-code** of our model to enhance the clarity in methodological presentation. For detailed results, please see *Algorithm 1 in Appendix A on Page 15*.
>
> Additionally, we have updated the related works to more clearly define the scope and background of this study. We have revised statements that could be slightly misleading, such as “improvements up to 20.8%.”  We have also revised the colors in Figure 15 and have updated the relevant text in the revised paper to make the interpretability more clearly.
>
> We sincerely thank you for your insightful review. We eagerly await your feedback and are ready to respond to any further questions you may have.

---

> ### Author Response · Authors · 2024-11-25
> **Request of Reviewer's attention and feedback**
>
> Dear Reviewer 7NU8,
>
> We sincerely thank you for your valuable and constructive feedback. Since the End of author/reviewer discussions is coming soon, may we know if our response addresses your main concerns? If so, we kindly ask for your reconsideration of the score. Should you have any further advice on the revised paper or our rebuttal, please let us know and we will be more than happy to engage in more discussion and paper improvements.
>
> Thank you so much for devoting time to improving our paper!

---

> ### Comment · Reviewer_7NU8 · 2024-11-27
>
> I appreciate the author's effort in addressing my concerns and clarifying potential misunderstandings. The additional results can potentially improve the paper's presentation and contribution. However, several of the justifications provided are unclear.
>
> More specifically:
>
> - **Significance of results:** I thank the authors for adding the standard deviations, but this was done only for their method. I remain unconvinced if all models are run 5 times with random seeds. Some seem to be directly taken from relevant papers (e.g., TimesNet) where results are, in several cases, mentioned for a fixed seed (CrossGNN). I highlight this since, for point-wise metrics, differences in terms of 2nd-3rd decimals might be minor at the prediction level.
>
> - **Comparison with GNN-based methods:** I thank the authors for incorporating more experimental results concerning relevant GNN-based forecasting methods. The additional sentence about the GNN-based forecasting methods provides minimal explanations in the related section. It does not adequately support the central novelty justification in the graph-learning part of the method. It seems that using two decoupled temporal and spatial graphs in a sequential manner is a choice rather experimentally than conceptually motivated.
>
> - **Novelty of Contribution:** I remain unconvinced about the significance of the methodological contribution. The idea is positioned in between GNN-based forecasting and leveraging embedding mechanisms for forecasting in terms of contribution and experiments. Patching and channel independence are not novel when targeting a contribution in sequential models for forecasting. Similarly, spatial-temporal graphs and GNN-based forecasting have been well-studied but remain hard to interpret and computationally expensive. Finally, the ablation of the model components proves the significance of patching and input embeddings rather than graph structure learning (where replacing with attention gives similar results). In the second case of positioning their work among sequential models forecasting, more recent sota (e.g., FITS) could have been used for more meaningful comparisons.
>
> I once again sincerely thank the authors for their inputs. However, based on the above I prefer for the moment to maintain my scores.

---

> > ### Author Response · Authors · 2024-12-01
> >
> > # Response to Reviewer 7NU8: Additional Concern (1/3)
> >
> >
> >
> > > **Additional Concern 1:** I thank the authors for adding the standard deviations, but this was done only for their method. I remain unconvinced if all models are run 5 times with random seeds. Some seem to be directly taken from relevant papers (e.g., TimesNet) where results are, in several cases, mentioned for a fixed seed (CrossGNN). I highlight this since, for point-wise metrics, differences in terms of 2nd-3rd decimals might be minor at the prediction level.
> >
> > Thank you for pointing out this issue. We understand your concern about the inclusion of standard deviations only for our method and whether all models were run 5 times with different random seeds. We would like to clarify our experimental approach:
> >
> > 1. **Common Practice in the Field:** It is a widely accepted practice in the time series forecasting community to directly report results from relevant papers **when the experimental settings are consistent**. This approach ensures comparability and builds upon established benchmarks. For instance:
> >    - **CrossGNN** [1] and **iTransformer** [2] have directly collected baseline results from **TimesNet** [3].
> >    - **TimeXer** [4], **S-Mamba** [5], and **CycleNet** [6] have directly used baseline results from **iTransformer** [2].
> > 2. **Use of Fixed Seed Results:** We acknowledge that some methods, like **CrossGNN**, have reported results using a fixed seed. However, subsequent works such as **Ada-MSHyper** [7] have also directly taken the results of CrossGNN for their comparative analyses. This indicates an acceptance within the community of using such reported results for benchmarking purposes.
> > 3. **Impact on Comparative Analysis:** According to reproductions reported in works like **Bi-Mamba4+** [8], CrossGNN has not been identified as a particularly strong baseline. Therefore, the potential variance introduced by not running CrossGNN multiple times with different seeds may not significantly affect the overall insights of our additional comparison study on Table 9 of Page 18.
> >
> > We agree that including standard deviations for all compared methods would enhance the rigor of our study. However, rerunning all baseline experiments multiple times with different random seeds is a substantial undertaking that may not be feasible within our current resources and timelines.
> >
> > **In the camera-ready version, we will:**
> >
> > - **Clarify in the Manuscript:** Indicate which results are taken directly from original papers and which are from our implementations.
> > - **Discuss Variability:** Include a discussion on the potential impact of using fixed-seed results, acknowledging that minor differences in metrics might occur due to variability.
> > - **Ensure Fair Comparison:** Adjust our analysis and conclusions to account for any limitations arising from the use of fixed-seed results.
> >
> > We hope this addresses your concern. Thank you for your valuable feedback, which helps us improve our work.
> >
> > ---
> >
> > [1] *Crossgnn: Confronting noisy multivariate time series via cross interaction refinement*. NeurIPS, 2023
> >
> > [2] *iTransformer: Inverted Transformers Are Effective for Time Series Forecasting.* ICLR, 2024
> >
> > [3] *TimesNet: Temporal 2D-Variation Modeling for General Time Series Analysis.* ICLR, 2023
> >
> > [4] *TimeXer: Empowering Transformers for Time Series Forecasting with Exogenous Variables.* arXiv, 2024
> >
> > [5] *Is mamba effective for time series forecasting?* arXiv, 2024
> >
> > [6] *Cyclenet: enhancing time series forecasting through modeling periodic patterns.* arXiv, 2024
> >
> > [7] *Ada-MSHyper: Adaptive Multi-Scale Hypergraph Transformer for Time Series Forecasting.* arXiv, 2024
> >
> > [8] *Bi-Mamba4TS: Bidirectional Mamba for Time Series Forecasting.* arXiv, 2024

---

> > > ### Author Response · Authors · 2024-12-01
> > >
> > > # Response to Reviewer 7NU8: Additional Concern (2/3)
> > >
> > > > **Additional Concern 2 (1/2):** The additional sentence about the GNN-based forecasting methods provides minimal explanations in the related section. It does not adequately support the central novelty justification in the graph-learning part of the method.
> > >
> > > Thank you for pointing out this issue. Although we have, in response to your previous valuable feedback, added four GNN-based baselines, we did not provide an in-depth discussion due to (1) limitations in paper space, and (2) the fact that these widely adopted GNN-based methods did not perform as strong baselines.
> > >
> > > In the camera-ready version, we will address this issue by:
> > >
> > > - **Expand the Related Work on GNN-based Forecasting Methods** in appendix to provide a comprehensive overview of existing approaches, discussing their methodologies, strengths, and limitations in detail.
> > > - **Clarify the Novelty of Our Graph-Learning Method** by clearly articulating how our approach differs from existing methods and explaining how our approach addresses the limitations of current GNN-based forecasting methods.
> > >
> > > For more details about the novelty of our approach, please refer to the **Response to Reviewer 9agk (2/2)**.
> > >
> > > We believe these enhancements will better support the central novelty of our work and provide a clearer understanding of our contributions to the graph-learning aspect of time series forecasting.
> > >
> > >
> > >
> > > > **Additional Concern 2 (2/2)**: It seems that using two decoupled temporal and spatial graphs in a sequential manner is a choice rather experimentally than conceptually motivated.
> > >
> > >
> > >
> > > Thank you for pointing out this issue, and we apologize for any misunderstanding it may have caused. We would like to clarify that **this design choice is conceptually motivated** and **represents the main motivation and innovation of our paper.** For more details about the novelty of our approach, please refer to the **Response to Reviewer 9agk (2/2)**. For your convenience, we provide a summary of the motivation behind our framework design below:
> > >
> > > Our work is driven by a core question: *"Is it truly necessary to model all these dependencies?"* We are the first to study the trade-off between **the variety of dependencies extracted** and **the noise that may be introduced by channel-mixing**. Previous works [1] [2] often adopt a "brute-force modeling" approach (e.g. hypervariate graphs), stacking parameters to capture as many dependencies as possible. While this might seem effective, it overlooks a critical issue—the potential noise introduced by this process.
> > >
> > > To investigate and minimize this potential noise, we propose a **novel channel-preserving framework**: GraphSTAGE. The use of decoupled temporal and spatial graphs in a sequential manner is a conceptual decision rooted in our aim to preserve channel integrity. **The model variants comparison is not intended to suggest that our model design is an experimental choice.** Rather, **through the comparisons** presented in *Table 4 on Page 10*, **we validate the presence of noise introduced by channel-mixing methods, underscoring the limitations of excessive dependency extraction.**
> > >
> > > We believe that our approach not only reveals the issue of noise but also highlights the importance of balancing dependencies modeling with the potential for noise introduction in time series forecasting.
> > >
> > > ---
> > >
> > > [1] *FourierGNN: Rethinking multivariate time series forecasting from a pure graph perspective.* NeurIPS, 2024
> > >
> > > [2] *UniTST: Effectively Modeling Inter-Series and Intra-Series Dependencies for Multivariate Time Series Forecasting.* arXiv, 2024

---

> ### Author Response · Authors · 2024-12-02
>
> # Response to Reviewer 7NU8: Additional Concern (3/3)
>
> > **Additional Concern 3 (1/5):** The idea is positioned in between GNN-based forecasting and leveraging embedding mechanisms for forecasting in terms of contribution and experiments.
>
> Thank you for pointing out this. Embedding mechanisms are used to inject relative positional information into inputs. While we have indeed proposed a refined time embedding in our paper (detailed in *paragraph 2 of Section 3.1*), which allows for more granular and sampling-frequency-adaptive relative positioning and significantly enhances performance on high-frequency datasets, **we did not list embedding mechanisms as our core contribution.**
>
> As outlined in *Section 1 of the revised paper* , **our main contributions** are the novelty in **research motivation** and innovation in **model framework**. Specifically, **we are the first to study the trade-off** between the variety of dependencies extracted and the noise that may be introduced by channel-mixing. Additionally, we propose a pure graph paradigm that decouples the extraction of global temporal dependencies and global spatial dependencies.
>
> We believe that these contributions position our work beyond being merely between GNN-based forecasting and leveraging embedding mechanisms. **Our focus is on addressing previously unexplored issues in dependency modeling and proposing a novel framework to tackle these challenges.**
>
> We hope this clarifies the core contributions of our work.
>
> ---
>
> > **Additional Concern 3 (2/5):** Patching and channel independence are not novel when targeting a contribution in sequential models for forecasting.
>
> Thank you for pointing this out. Patching is a common modeling method, and **we do not claim it as our contribution**.
>
> We also **did not claim in our paper that we are a channel independence model**. Since our model extracts temporal and spatial dependencies, we do not strictly enforce channel independence.
>
> ---
>
> > **Additional Concern 3 (3/5):** Similarly, spatial-temporal graphs and GNN-based forecasting have been well-studied but remain hard to interpret and computationally expensive.
>
> Thank you for pointing out this. We believe that the general statement about spatial-temporal graphs and GNN-based forecasting methods being hard to interpret and computationally expensive **may not directly apply to our work.**
>
> 1. **Interpretability:** We have addressed the interpretability of our model in the **Response to Reviewer 7NU8 (4/4), specifically in W7 & Q4 (2/2)**. In summary, the temporal and spatial dependencies learned by our model **match the inherent patterns of the dataset** and **match the ground truth**. This demonstrates that our model provides meaningful and interpretable insights.
>
> 2. **Computational Cost:** As shown in *Figure 5 on page 7*, GraphSTAGE does not exhibit significantly higher computational costs compared to other methods. **Our model achieves superior performance with acceptable computational overhead**.
>
> ---
>
> > **Additional Concern 3 (4/5):** The ablation of the model components proves the significance of patching and input embeddings rather than graph structure learning (where replacing with attention gives similar results).
>
> Thank you for pointing out this. We greatly appreciate your thorough review and attention to the details of our ablation study. While in a small number of tests replacing the STAGE block with attention yields comparable results, closeness does not imply complete equivalence. Overall, the STAGE block outperforms the attention mechanism.
>
> Moreover, even when using attention, **it operates within our proposed channel-preserving framework.** The experimental results further highlight the superiority of our framework, which is precisely the focus of our core contributions.

---

> > ### Author Response · Authors · 2024-12-02
> >
> > > **Additional Concern 3 (5/5):** In the second case of positioning their work among sequential models forecasting, more recent sota (e.g., FITS) could have been used for more meaningful comparisons.
> >
> > Thank you for pointing out this. We appreciate your attention to ensuring a more meaningful comparisons.
> >
> > Including FITS in our comparisons presents some challenges regarding fairness. Actually, **FITS conducts grid search over the look-back window lengths of 90, 180, 360, and 720 to find the optimal input length**, as mentioned in their implementation details on *page 6* [1]. However, when reporting baseline results from other models like **TimesNet [2], FITS uses the raw results where the input window length is fixed at 96.**
> >
> > **This discrepancy can lead to unfair comparisons.**
> >
> > To address your concern, we adopted the same setting as FITS [1], conducting a grid search over the same look-back window lengths $[90, 180, 360, 720]$ with the prediction length fixed at 96. We tested this on three datasets and obtained the following results:
> >
> > | **Method**  | GraphSTAGE (Optimal Input Length) |                 | FITS (Optimal Input Length) |             | GraphSTAGE (Fixed Input Length = 96) |             |
> > | ----------- | --------------------------------- | --------------- | --------------------------- | ----------- | ------------------------------------ | ----------- |
> > | **Metric**  | MSE                               | MAE             | MSE                         | MAE         | MSE                                  | MAE         |
> > | **ETTm1**   | **0.295**±0.002                   | **0.346**±0.003 | 0.304±0.001                 | 0.348±0.001 | 0.319±0.004                          | 0.356±0.003 |
> > | **Weather** | **0.133**±0.001                   | **0.188**±0.001 | 0.143±0.001                 | 0.196±0.001 | 0.159±0.001                          | 0.208±0.001 |
> > | **ECL**     | **0.128**±0.001                   | **0.224**±0.001 | 0.142±0.001                 | 0.242±0.000 | 0.139±0.001                          | 0.237±0.001 |
> >
> > The results in the table above show that:
> >
> > 1. Under the same setting of **grid-searched optimal input length**, **GraphSTAGE outperforms FITS** significantly.
> > 2. Comparing GraphSTAGE with optimal input length and fixed input length of 96, we see that **increasing the input length improves the model's performance**. This observation highlights that FITS's practice of grid searching input lengths while using baseline models' results with a fixed input length of 96 leads to unfair comparisons.
> >
> > **Furthermore, we have discussed the performance gains from increasing the look-back length in our revised paper, please refer to *Figure 6 on Page 7.***
> >
> > We hope these additional experiments provide a clearer understanding of GraphSTAGE's performance and address your concerns. Thank you again for your valuable feedback.
> >
> > ---
> >
> > [1] *FITS: Modeling Time Series with $10k$ Parameters*. ICLR, 2024.
> >
> > [2] *TimesNet: Temporal 2D-Variation Modeling for General Time Series Analysis.* ICLR, 2023

---

> ### Author Response · Authors · 2024-12-03
> **Kind Request for Reviewer's Attention and Feedback**
>
> Dear Reviewer 7NU8,
>
> Hope this message finds you well.
>
> We appreciate the diligent efforts of you in evaluating our paper. We have responded in detail to your additional concerns. As the discussion period will end soon, we would like to kindly ask whether there are any remaining concerns or questions we might be able to address. If our current response satisfactorily resolves your main concerns, we kindly ask for your reconsideration of the score.
>
> Once more, we appreciate the time and effort you've dedicated to our paper.
>
> Best regards,
>
> Authors

---

> > ### Author Response · Authors · 2024-12-04
> >
> > Dear Reviewer 7NU8,
> >
> > Hope this message finds you well.
> >
> > We appreciate the diligent efforts of you in evaluating our paper. **We have provided detailed responses to your additional concerns** regarding **the significance of our results, the specifics of our model design, and the comparisons with more recent sota.** May we know if our response addresses your main concerns? If so, we kindly ask for your reconsideration of the score.
> >
> > Once more, we appreciate the time and effort you've dedicated to our paper.

---

### Official Review · Reviewer_9agk · 2024-11-03

**Soundness:** 3
**Presentation:** 3
**Contribution:** 2
**Rating:** 5
**Confidence:** 4

**Summary:**

This paper introduces GRAPHSTAGE, a fully GNN-based method that captures intra-series and inter-series dependencies while maintaining the shape of the input data through a channel-preserving strategy. Extensive experiments conducted on 13 real-world datasets
demonstrate that GRAPHSTAGE achieves competitive performance.

**Strengths:**

1. The paper is well-written and easy to follow.

2. Extensive experiments  across 13 benchmark datasets have demonstrated that GRAPHSTAGE achieves good performance.

**Weaknesses:**

1. My primary concern is regarding the term "noise" (see line 16). The authors claim that previous methods introduce additional noise, but it remains unclear what this "noise" refers to. How is it defined, and what impact does it have on the effectiveness of the methods?

2. This paper appears to be an incremental contribution, with its components largely derived from previous works. Channel-preserving Strategy is inspired by iTransformer, and the fully GNN perspective is inspired by FourierGNN.

**Questions:**

please refer to the weaknesses part

---

> ### Author Response · Authors · 2024-11-21
>
> # Response to Reviewer 9agk (1/2)
>
> > **W1: My primary concern is regarding the term "noise" (see line 16). The authors claim that previous methods introduce additional noise, but it remains unclear what this "noise" refers to. How is it defined, and what impact does it have on the effectiveness of the methods?**
>
>
>
> Thank you for pointing out this. The "noise" refers to the **potential learning of pseudo connections** when using a Hypervariate Graph to capture dependencies in channel-mixing methods.
>
> Specifically, the size of the Hypervariate Graph ($\mathbb{R}^{NT \times NT}$) is significantly larger than the learnable graphs we propose: the Temporal Learnable Graph $A_T \in \mathbb{R}^{T \times T}$ and the Spatial Learnable Graph $A_S \in \mathbb{R}^{N \times N}$. Consequently, **channel-mixing models like FourierGNN [1] and UniTST [2] have a higher degree of freedom** compared to our channel-preserving model (GraphSTAGE). This increased degree of freedom can lead the model to **learn many pseudo connections** (connections between different nodes at different time steps), which **weakens the weights of real connections** (inter-series and intra-series dependencies). This may result in **overfitting** and **a decrease in predictive performance.**
>
> To test our hypothesis that the channel-mixing strategy **leads to overfitting** and **reduced predictive performance,** we conducted experiments on two datasets. Since the dependency learning components of FourierGNN [1] and UniTST [2] differ from those in GraphSTAGE, we used the model variant **VarC**  in *Figure 7 on Page 9* as a baseline. This variant '**VarC**' changes only the channel-preserving strategy of the original GraphSTAGE to a channel-mixing strategy while keeping the dependency learning components consistent.
>
> To demonstrate the existence of overfitting in the channel-mixing strategy, we set the training epochs to 30 and used a fixed learning rate of 1e-3. We tested on the ECL and PEMS03 datasets with input length equal to prediction length (24 time steps). All experiments have been conducted five times and hyperparameters remained the same. The results are shown in the table below:
>
> | **Method**      | Channel-Preserving Model: Orig in Fig. 7 (GraphSTAGE) |           |          | Channel-Mixing Model: VarC in Fig. 7 |           |          |
> | --------------- | ----------------------------------------------------- | --------- | -------- | ------------------------------------ | --------- | -------- |
> | Loss            | Train MSE                                             | Valid MSE | Test MSE | Train MSE                            | Valid MSE | Test MSE |
> | ECL  (24→24)    | 0.110                                                 | 0.105     | 0.122    | 0.106                                | 0.103     | 0.130    |
> | PEMS03  (24→24) | 0.066                                                 | 0.069     | 0.082    | 0.056                                | 0.065     | 0.091    |
>
> As shown, the channel-mixing model VarC has lower training MSE but higher test MSE. The results indicate that **VarC (channel-mixing model) exhibits overfitting** compared to Orig (channel-preserving model).
>
> ---
>
> To further demonstrate the decrease in predictive performance on the test set caused by the channel-mixing strategy, we refer to the results provided in *Table 4 on Page 10* of our paper. For your convenience, we summarize the relevant (averaged) results below:
>
> | **Method** | Channel-Preserving Model: Orig in Fig. 7 (GraphSTAGE) |           | Channel-Mixing Model: VarC in Fig. 7 |         |
> | ---------- | ----------------------------------------------------- | --------- | ------------------------------------ | ------- |
> | **Metric** | **MSE**                                               | **MAE**   | **MSE**                              | **MAE** |
> | **ECL**    | **0.166**                                             | **0.263** | 0.192                                | 0.284   |
> | **ETTm1**  | 0.391                                                 | **0.394** | **0.389**                            | 0.400   |
>
> The channel-preserving model Orig outperforms the channel-mixing variant VarC, **supporting our hypothesis that channel-mixing leads to decreased predictive performance.**
>
> ---
>
> [1] *FourierGNN: Rethinking multivariate time series forecasting from a pure graph perspective.* NeurIPS, 2024
>
> [2] *UniTST: Effectively Modeling Inter-Series and Intra-Series Dependencies for Multivariate Time Series Forecasting.* arXiv

---

> ### Author Response · Authors · 2024-11-21
>
> # Response to Reviewer 9agk (2/2)
>
> > **W2: This paper appears to be an incremental contribution, with its components largely derived from previous works. Channel-preserving Strategy is inspired by iTransformer, and the fully GNN perspective is inspired by FourierGNN.**
>
> Thank you for pointing out this. We would like to highlight the unique aspects of our approach and clarify the distinctions between our work and the recent models you mentioned.
>
> 1. **Novelty in Research Motivation:**
>    We are the first to study the trade-off between **the variety of dependencies extracted** and **the noise that may be introduced by channel-mixing**. As we stated in our introduction: *"Is it truly necessary to model all these dependencies?"*   Previous works [1] [2] often adopt a "brute-force modeling" approach, stacking parameters to capture as many dependencies as possible. While this might seem effective, it overlooks a critical issue—the potential noise introduced by this process.
> 2. **Innovation in Model Framework:**
>    We propose a **novel channel-preserving framework**: GraphSTAGE. Through fair model variant experiments in *Table 4 on Page 10*, we validate the presence of such noise, underscoring the limitations of excessive dependency extraction. GraphSTAGE is a pure graph paradigm that decouples the extraction of **global** **temporal** dependencies and **global** **spatial** dependencies, **rather than being limited to local** neighbor information.
> 3. **Advancements in Model Performance:**
>    Despite its structural simplicity, our model performs comparably to or surpasses state-of-the-art models across 13 MTSF benchmark datasets. It **ranks first among 8 advanced models** in 22 out of 30 comparisons in *Table 1 on Page 7*. Specifically, on the PEMS07 dataset—which has the largest number of nodes—GraphSTAGE outperforms the recent SOTA iTransformer by 20.8%, indicating its **potential for application to larger-scale MTSF tasks**, such as extensive grid management.
>
> ---
>
> **$\triangleright$Differences with iTransformer:**
>
> Conceptually, **iTransformer** is not a channel-preserving model; it **overlooks temporal channel information.** iTransformer projects the original time series data $X_{\text{in}} \in \mathbb{R}^{N \times T}$ (where $N$ is the number of nodes and $T$ is the length of the time series) into $H_S \in \mathbb{R}^{N \times D}$ to capture spatial dependencies among nodes. However, this transformation ignores temporal dependencies and fails to learn the underlying temporal graph structures.
>
> In contrast, our GraphSTAGE embeds the input data $X_{\text{in}} \in \mathbb{R}^{N \times T}$ into $H \in \mathbb{R}^{N \times T \times D}$, where $D$ is the embedding dimension. The original node and time dimensions are preserved. This channel-preserving framework enables the model to incorporate both spatial (inter-series) and temporal (intra-series) dependencies by decoupling them. This separation allows GraphSTAGE to capture temporal dynamics more effectively, which is a significant extension beyond what iTransformer offers.

---

> ### Author Response · Authors · 2024-11-21
>
> **$\triangleright$Differences with FourierGNN:**
>
> Although both are pure graph methods, Graph Neural Networks (GNNs) are a very broad concept. **Being graph-based does not make our work incremental.**
>
> 1. **Dependency Modeling Approach:**
>    **FourierGNN** represents a trend toward modeling **three kinds of dependencies** within a unified framework, as shown in *Figure 2 on Page 2*. **GraphSTAGE**, however, focuses on decoupling and **modeling two types of dependencies.**
>
> 2. **Research Focus:**
>    The drawbacks of using hypervariate graphs to unify multi-dependency modeling have not been studied in previous research. There are two main drawbacks: **High Memory Consumption:** This issue has also been reported in other studies [3]. **Potential Noise:** Modeling extensive cross-dependencies may lead the model to learn many pseudo connections (as we explained in our response to Q1). These pseudo connections can weaken the influence of genuine connections (inter-series and intra-series dependencies), potentially diminishing the model's predictive performance.
>
>    Additionally, instead of focusing solely on FourierGNN, we aimed to broadly address hypervariate graph-based multi-dependency modeling methods. In the Variants Comparison section, we included a variant called **VarC** (shown in *Figure 7 on Page 9*) to encompass these methods and compared it with our channel-preserving GraphSTAGE. Our findings indicate that **modeling only two types of dependencies can achieve excellent predictive performance while also reducing memory usage.**
>
> ---
>
> **$\triangleright$Addition of Graph-Based Baselines:**
>
> To **provide a more direct and comprehensive comparison with FourierGNN** and other graph-based models, we have included four additional graph-based baselines in the revised paper. The detailed results are presented in *Table 9 on Page 18* of the revised paper. For your convenience, we provide a summary of the relevant averaged results below:
>
> | **Model**          | GraphSTAGE          |                     | FourierGNN[1]   |                 | CrossGNN[4] |       | StemGNN[5]      |                 | MTGNN[6] |       |
> | ------------------ | ------------------- | ------------------- | --------------- | --------------- | ----------- | ----- | --------------- | --------------- | -------- | ----- |
> | Metric             | MSE                 | MAE                 | MSE             | MAE             | MSE         | MAE   | MSE             | MAE             | MSE      | MAE   |
> | ECL (321 Nodes)    | **0.166**$\pm$0.003 | **0.263**$\pm$0.002 | 0.221$\pm$0.001 | 0.318$\pm$0.001 | 0.201       | 0.300 | 0.215$\pm$0.002 | 0.316$\pm$0.002 | 0.251    | 0.347 |
> | ETTm1 (7 Nodes)    | **0.391**$\pm$0.003 | **0.394**$\pm$0.002 | 0.453$\pm$0.002 | 0.448$\pm$0.003 | 0.393       | 0.404 | 0.550$\pm$0.004 | 0.537$\pm$0.005 | 0.469    | 0.446 |
> | ETTh2 (7 Nodes)    | **0.387**$\pm$0.005 | **0.407**$\pm$0.004 | 0.543$\pm$0.006 | 0.517$\pm$0.004 | 0.393       | 0.418 | 1.158$\pm$0.010 | 0.812$\pm$0.011 | 0.465    | 0.509 |
> | Weather (21 Nodes) | **0.243**$\pm$0.001 | **0.274**$\pm$0.001 | 0.257$\pm$0.002 | 0.305$\pm$0.003 | 0.247       | 0.289 | 0.289$\pm$0.005 | 0.342$\pm$0.007 | 0.314    | 0.355 |
>
> The results indicate that GraphSTAGE achieved the **top-1** rank in terms of MSE and MAE metrics across four datasets **compared with four graph-based models.** Although CrossGNN performs comparably to our GraphSTAGE on smaller datasets (e.g., ETT and Weather), GraphSTAGE notably outperforms CrossGNN on large-scale datasets (e.g., ECL with 321 nodes). These findings further validate the effectiveness of GraphSTAGE in capturing both intra-series and inter-series dependencies, leading to **superior forecasting accuracy across datasets of varying scales.**
>
> We believe these clarifications address your concerns regarding the novelty and contributions of our work. Current models primarily focus on the advantages of channel-mixing methods for extracting multiple dependencies, often **neglecting the noise these approaches can introduce.** **We are the first to directly address this issue.** Thank you again for your valuable feedback, which has helped us improve our paper.
>
> ---
>
> [1] *FourierGNN: Rethinking multivariate time series forecasting from a pure graph perspective.* NeurIPS, 2024
>
> [2] *UniTST: Effectively Modeling Inter-Series and Intra-Series Dependencies for Multivariate Time Series Forecasting.* arXiv
>
> [3] *ForecastGrapher: Redefining Multivariate Time Series Forecasting with Graph Neural Networks.* arXiv
>
> [4] *Crossgnn: Confronting noisy multivariate time series via cross interaction refinement*. NeurIPS, 2023
>
> [5] *Spectral temporal graph neural network for multivariate time-series forecasting.* NeurIPS, 2020
>
> [6] *Connecting the dots: Multivariate time series forecasting with graph neural networks.* KDD, 2020

---

> > ### Comment · Reviewer_9agk · 2024-11-23
> >
> > Dear Authors,
> >
> > Thank you for your efforts in addressing my previous comments.
> >
> > I have one additional question:  Based on your explanations regarding "noise," such as the potential learning of pseudo connections, it seems that patches themselves might also introduce noise. For instance, consider two patches, $x_{1:p}$ and $y_{1:p}$. It is possible that $x_1$ and $y_p$ may have no meaningful connections. Could you explain how GraphSTAGE addresses the noise introduced by such scenarios?

---

> > > ### Author Response · Authors · 2024-11-24
> > >
> > > Thank you for your feedback and for raising this valuable question. We would like to address this issue from two perspectives:
> > >
> > > 1. The theoretical principles of GraphSTAGE.
> > > 2. The results of ablation study.
> > >
> > > ### 1. Theoretical Explanation
> > >
> > > - To illustrate what the STAGE block in GraphSTAGE is learning, let us consider an example with two nodes, $x$ and $y$, in a sample. After patching, we obtain four patches: $x_{1:p}$, $x_{p:2p}$, $y_{1:p}$, and $y_{p:2p}$​.
> > >
> > >   In the **Intra-GrAG module**, the model focuses exclusively on learning the temporal dependencies **within the patches of the same node**. For example:
> > >
> > >   - For the node $x$, model learns the connection between patches $x_{1:p}$ and $x_{p:2p}$​.
> > >   - Similarly, for the node $y$, model learns the connection between patches $y_{1:p}$ and $y_{p:2p}$.
> > >
> > >   In the **Inter-GrAG module**, the model specifically learns the spatial dependencies **among patches from different nodes**, focusing on patches that correspond to the same temporal segment. For instance:
> > >
> > >   - It captures the connection between $x_{1:p}$ (the first patch of node $x$) and $y_{1:p}$ (the first patch of node $y$).
> > >   - Similarly, it captures the connection between $x_{p:2p}$ (the second patch of node $x$) and $y_{p:2p}$ (the second patch of node $y$).
> > >
> > >   Importantly, the **Inter-GrAG module does not consider dependencies between patches that correspond to different temporal segments**, such as $x_{1:p}$ with $y_{p:2p}$ or $x_{p:2p}$ with $y_{1:p}$. Its scope is strictly limited to learning spatial interactions within the same temporal window.
> > >
> > > - Regarding whether the spatial dependency between $x_{1:p}$ and $y_{1:p}$ includes the 'pseudo connection' between $x_1$ and $y_p$, **this is theoretically possible, but it requires validation through ablation studies.** Compared to methods like FourierGNN [1], GraphSTAGE achieves a better balance between noise and performance. Moreover, GraphSTAGE's cross-series dependency learning **is restricted to** connections between different nodes within the **same patch** and **does not involve** connections between different nodes across **different patches**. Therefore, the potential for learning this type of noise you mentioned is **significantly** **minimized compared to other hypervariate graph-based** modeling methods.
> > >
> > > ### 2. Ablation Study
> > >
> > > **If the model learns noise as you mentioned, adding patching would introduce this noise into the learning process, causing the model's prediction performance to degrade.** To validate this, we conducted ablation experiments on 8 datasets. Following the experimental setup of iTransformer [2], we used a fixed lookback length $T=96$ and a prediction length $K=96$, running each experiment five times. For the ETTm1, ETTh2, Weather, and ECL datasets, we supplemented results with experiments labeled as “w/o Patching.” For the PEMS datasets, ablation results were collected from *Table 13 on Page 24* of the revised paper.
> > >
> > > | **Method**      | GraphSTAGE (Patching) |                 | w/o Patching |             |
> > > | --------------- | --------------------- | --------------- | ------------ | ----------- |
> > > | Metric          | MSE                   | MAE             | MSE          | MAE         |
> > > | ETTm1 (96→96)   | **0.319**±0.004       | **0.356**±0.003 | 0.327±0.004  | 0.363±0.003 |
> > > | ETTh2 (96→96)   | **0.292**±0.002       | **0.341**±0.002 | 0.295±0.003  | 0.344±0.002 |
> > > | Weather (96→96) | **0.159**±0.001       | **0.208**±0.001 | 0.163±0.002  | 0.210±0.002 |
> > > | ECL (96→96)     | **0.139**±0.001       | **0.237**±0.001 | 0.145±0.006  | 0.241±0.004 |
> > > | PEMS03 (96→96)  | **0.136**             | **0.253**       | 0.160        | 0.272       |
> > > | PEMS04 (96→96)  | **0.113**             | **0.228**       | 0.134        | 0.259       |
> > > | PEMS07 (96→96)  | **0.105**             | **0.209**       | 0.146        | 0.257       |
> > > | PEMS08 (96→96)  | **0.207**             | **0.270**       | 0.295        | 0.348       |
> > >
> > > ​	As the results show, **adding patching consistently enhances prediction performance across all datasets**. If the model were indeed learning noise due to patching, we would expect the prediction performance to degrade when patches are added. However, the observed improvements indicate that this hypothesis is invalid. In fact, although GraphSTAGE employs patching in the Embedding & Patching layer, it remains a channel-preserving framework. **The patching operation merely increases the receptive field of each temporal channel, allowing the data to become smoother and more resilient to the influence of outliers.**
> > >
> > > ​	Thank you again for your prompt feedback. We hope this explanation and the accompanying experimental results effectively address your concerns.
> > >
> > > ---
> > >
> > > [1] *FourierGNN: Rethinking multivariate time series forecasting from a pure graph perspective.* NeurIPS, 2024
> > >
> > > [2] *iTransformer: Inverted Transformers Are Effective for Time Series Forecasting.* ICLR, 2024

---

> > > > ### Comment · Reviewer_9agk · 2024-11-24
> > > >
> > > > Dear Authors,
> > > >
> > > > Thank you for your response.
> > > >
> > > > I believe the ablation studies do not fully demonstrate that patches do not introduce noise. As shown in PatchTST [1], patches can enhance performance by enabling the model to process longer historical sequences. Even if some noise is introduced, the overall benefits outweigh the drawbacks.
> > > >
> > > >
> > > >
> > > > [1] A TIME SERIES IS WORTH 64 WORDS: LONG-TERM FORECASTING WITH TRANSFORMERS, in ICLR, 2023

---

> ### Author Response · Authors · 2024-11-24
>
> Thank you for your prompt feedback. I agree with your point, especially the statement, “Even if some noise is introduced, the overall benefits outweigh the drawbacks.” **I cannot agree more.**
>
> Taking the example you mentioned, consider a node $x$. After patching, we obtain $x_{1:p}$, $x_{p:2p}$, $x_{2p:3p}$, $x_{3p:4p}$. If we **assume that $ x_{1:p}$ and $x_{3p:4p}$ are unrelated.** However, when GraphSTAGE learns the temporal dependencies **within the patches of the same node**, it's **hard to ensure that the model doesn't assign weights to the "pseudo connections"** between $x_{1:p}$ and $x_{3p:4p}$. Therefore, ablation studies can only demonstrate that the model achieves **overall benefits outweigh the drawbacks**, but it is challenging to specifically assess whether noise is introduced in the learning of each individual connection.
>
> However, extensive experimental results demonstrate that GraphSTAGE achieves results that are comparable to or even surpass recent SOTA methods. This demonstrates that **the overall benefits outweigh the drawbacks** in GraphSTAGE.
>
> Therefore, I agree with your point that regardless of the type of dependency being modeled (whether inter-series, intra-series, or cross-series), noise might inevitably be introduced. The key is that the **model should ensure** **the overall benefits outweigh the drawbacks**.  Clearly, the current **hypervariate graph-based** modeling methods remain **inadequate**. GraphSTAGE addresses this challenge by decoupling the learning of the two types of dependencies, which helps to **minimize the introduction of noise as much as possible.**
>
> In the *Abstract on Page 1*, We stated: “The trade-off between the variety of dependencies extracted and the potential interference has not yet been fully explored.” This actually conveys a similar idea with you that **better models need to strike a balance between improving performance and suppressing noise.** Simply increasing the variety of dependencies extracted could **not only degrade model performance but also increase memory usage.**
>
> If you have any other concerns, please don’t hesitate to share them with us. Thank you again for your valuable insights.

---

> ### Author Response · Authors · 2024-11-25
> **Request of Reviewer's attention and feedback**
>
> Dear Reviewer 9agk,
>
> We sincerely thank you for your valuable and constructive feedback. Since the End of author/reviewer discussions is coming soon, may we know if our response addresses your main concerns? If so, we kindly ask for your reconsideration of the score. Should you have any further advice on the revised paper and/or our rebuttal, please let us know and we will be more than happy to engage in more discussion and paper improvements.
>
> Thank you so much for devoting time to improving our paper!

---

> ### Author Response · Authors · 2024-11-28
> **Kind Request for Reviewer's Attention and Feedback**
>
> Dear Reviewer 9agk,
>
> We sincerely thank you for your valuable and constructive feedback. Since the Discussion Period Extension provides us with additional time, we are eager to address any further concerns you may have. If our current response satisfactorily resolves your main concerns, we kindly ask for your reconsideration of the score. Should you have any further advice on our rebuttal, please let us know, and we will be more than happy to engage in further discussion.
>
> Thank you so much for devoting time to improving our paper!

---

> > ### Comment · Reviewer_9agk · 2024-11-28
> >
> > Dear Authors,
> >
> > Thank you sincerely for your efforts to address my concerns.
> >
> > While the experimental results show some improvement in the forecasting performance of your proposed model, the underlying source of this improvement remains unclear. Therefore, I prefer to maintain my original score.

---

> ### Author Response · Authors · 2024-12-01
>
> # Response to Reviewer 9agk: Additional Concern (1/1)
>
> > **Additional Concern1: While the experimental results show some improvement in the forecasting performance of your proposed model, the underlying source of this improvement remains unclear. Therefore, I prefer to maintain my original score.**
>
> Thank you for your feedback and for acknowledging the performance improvements achieved by our model. We have addressed your concern in the revised paper through a combination of **model variants comparisons**,  **ablation studies**, and **visualization of learned dependencies**. Below, we summarize these efforts to clarify the contributions of our design to the observed performance improvements:
>
> 1. **Model Framework and Variants Comparisons**
>     We conducted extensive experiments **comparing different model variants** to demonstrate the advantages of our decoupled sequential learning framework. Specifically, we evaluated alternative designs that modify the sequence of dependencies learning or use parallel and channel-mixing structures (VarA, VarB, VarC). For reference, please see *Table 4 on Page 10* in the revised paper. For your convenience, we provide a summary of the relevant averaged results below:
>
>    | **Method** | GraphSTAGE |           | VarA  |       | VarB  |       | VarC  |       |
>    | ---------- | ---------- | --------- | ----- | ----- | ----- | ----- | ----- | ----- |
>    | Metric     | MSE        | MAE       | MSE   | MAE   | MSE   | MAE   | MSE   | MAE   |
>    | ECL        | **0.166**  | **0.263** | 0.192 | 0.282 | 0.184 | 0.276 | 0.192 | 0.284 |
>
>    Our sequential structure (GraphSTAGE) outperforms variants, **especially on large-scale dataset (e.g., ECL with 321 nodes),** by effectively preserving the original data structure and decoupling the learning of inter-series and intra-series dependencies. **These results underscore the importance of our architectural design.**
>
> ---
>
> 2. **Ablation Studies on Model Design**
>     We performed detailed ablation studies to assess the contribution of key components in our model. For example:
>
>    - **Embedding Layer:** Results in *Table 3 on Page 8* show that removing components such as time embedding or adaptive embedding leads to significant performance drops. For your convenience, we provide a summary of the relevant averaged results below:
>
>      | Dataset           | PEMS03    |           | PEMS08    |           |
>      | ----------------- | --------- | --------- | --------- | --------- |
>      | Metric            | MSE       | MAE       | MSE       | MAE       |
>      | **GraphSTAGE**    | **0.097** | **0.210** | **0.139** | **0.220** |
>      | w/o Patching      | 0.110     | 0.222     | 0.176     | 0.253     |
>      | w/o Time Emb.     | 0.114     | 0.223     | 0.199     | 0.264     |
>      | w/o Adaptive Emb. | 0.121     | 0.257     | 0.203     | 0.260     |
>
>      Additionally, we introduced a **refined time embedding**, refining 'Hour of Day' (e.g., 24 hours/day for PEMS datasets) to 'Timestamp of Day' (e.g., 288 steps/day for PEMS datasets), as detailed in *paragraph 2 of Section 3.1*. This allows for **more granular and sampling-frequency-adaptive relative positioning** and significantly enhances performance on high-frequency datasets.
>
>    - **STAGE Module:** As shown in *Table 2 on Page 8*, the removal of key graph learning components, such as the Intra-GrAG or Inter-GrAG modules, results in notable performance degradation. This highlights their critical role in **capturing both intra-series and inter-series dependencies effectively.** For your convenience, we provide a summary of the relevant averaged results below:
>
>      | Dataset        | ECL       |           | Solar-Energy |           |
>      | -------------- | --------- | --------- | ------------ | --------- |
>      | Metric         | MSE       | MAE       | MSE          | MAE       |
>      | **GraphSTAGE** | **0.166** | **0.263** | **0.192**    | **0.267** |
>      | w/o Intra-GrAG | 0.185     | 0.277     | 0.225        | 0.292     |
>      | w/o Inter-GrAG | 0.186     | 0.276     | 0.239        | 0.294     |

---

> > ### Author Response · Authors · 2024-12-01
> >
> > 3. **Visualization of Learnable Graphs**
> >    To further illustrate the effectiveness of our model, we visualized the temporal learnable graphs $A_T$ and the spatial learnable graphs $A_S$. For more details, please refer to **the response to Reviewer 7NU8 (4/4) W7 & Q4 (2/2).**
> >
> >    - **Temporal Learnable Graphs:** In the ECL dataset *(Figure 8 on Page 10)*, **the learned temporal graph reveals periodic patterns** every 12 patches (24 hours), **matching the dataset's daily periodicity**. This demonstrates the model's ability to capture temporal dependencies.
> >
> >    - **Spatial Learnable Graphs:** We visualized **the ground truth** of random selected nodes in *Figure 15(right) on Page 25*, revealing that the trends of nodes 282 and 184 are consistent, which **matches with their high correlation coefficients learned by the Inter-GrAG module.** This match confirms the effectiveness of GraphSTAGE in capturing inter-series dependencies.
> >
> > ------
> >
> > Through **variant comparisons** , **ablation studies**, and **visualizations of learned dependencies**, we have **systematically demonstrated** how each component contributes to the model's overall performance. We hope this clarifies the source of the improvement and provides stronger evidence for the effectiveness of our approach.
> >
> > Thank you once again for your valuable and insightful feedback, which has been instrumental in improving our paper. We hope our response has addressed your main concerns, and we kindly ask for your reconsideration of the score. If you have any further suggestions regarding the revised paper or our rebuttal, please feel free to share them. We would be delighted to engage in further discussions and make additional improvements.
> >
> > Thank you so much for devoting time and effort to improving our paper!

---

> > > ### Author Response · Authors · 2024-12-03
> > > **Kind Request for Reviewer's Attention and Feedback**
> > >
> > > Dear Reviewer 9agk,
> > >
> > > Hope this message finds you well.
> > >
> > > We appreciate the diligent efforts of you in evaluating our paper. We have responded in detail to your questions. As the discussion period will end soon, we would like to kindly ask whether there are any additional concerns or questions we might be able to address. If our current response satisfactorily resolves your main concerns, we kindly ask for your reconsideration of the score.
> > >
> > > Once more, we appreciate the time and effort you've dedicated to our paper.
> > >
> > > Best regards,
> > >
> > > Authors

---

> > > > ### Author Response · Authors · 2024-12-04
> > > >
> > > > Dear Reviewer 9agk,
> > > >
> > > > Hope this message finds you well.
> > > >
> > > > We appreciate the diligent efforts of you in evaluating our paper. **We have responded in detail to your questions about the performance improvements achieved by our model.** May we know if our response addresses your main concerns? If so, we kindly ask for your reconsideration of the score.
> > > >
> > > > Once more, we appreciate the time and effort you've dedicated to our paper.

---

### Official Review · Reviewer_yAwh · 2024-11-04

**Soundness:** 2
**Presentation:** 3
**Contribution:** 2
**Rating:** 6
**Confidence:** 5

**Summary:**

The proposed GRAPHSTAGE model innovatively employs a pure Graph Neural Network (GNN) structure to preserve the channel structure of the input data. The authors claim that this approach avoids the noise and interference introduced by channel mixing methods.

**Strengths:**

1. By introducing the Spatio-Temporal Aggregation Graph Encoder (STAGE) module, the authors effectively separated and captured internal and external dependencies in the time series, enhancing the model's interpretability and predictive performance.

2. The paper conducted extensive experiments on 13 real-world datasets, demonstrating superior performance in multivariate time series prediction tasks, with significant improvements in prediction accuracy and computational efficiency compared to existing methods.

**Weaknesses:**

1. I don't quite understand why a purely graph-based structure can introduce noise and interference through channel mixing methods.

2. I have doubts about the effectiveness of purely using GNNs for time series analysis. The performance of GNNs heavily depends on the rationality of the graph structure. However, in certain datasets or tasks, constructing a reasonable graph structure is itself a challenge. If the graph structure fails to accurately capture the dependencies in the data, even the most advanced GNNs may struggle to achieve optimal performance. Additionally, the representational capacity of graphs may be limited in high-noise or irregular data, leading to a decline in the model's predictive capabilities. I hope the authors can provide a more comprehensive explanation.

3. FourierGNN also performs time series prediction, so why didn't the authors conduct a comprehensive comparison?

**Questions:**

see Weaknesses.

---

> ### Author Response · Authors · 2024-11-21
>
> # Responses to Reviewer yAwh (1/3)
>
> > **W1: I don't quite understand why a purely graph-based structure can introduce noise and interference through channel mixing methods.**
>
> Thank you for pointing out this. The noise and interference in channel-mixing methods, stem from the **potential learning of pseudo connections** when using a Hypervariate Graph to capture dependencies.
>
> Specifically, the size of the Hypervariate Graph ($\mathbb{R}^{NT \times NT}$) is significantly larger than the learnable graphs we propose: the Temporal Learnable Graph $A_T \in \mathbb{R}^{T \times T}$ and the Spatial Learnable Graph $A_S \in \mathbb{R}^{N \times N}$. Consequently, **channel-mixing models like FourierGNN [1] and UniTST [2] have a higher degree of freedom** compared to our channel-preserving model (GraphSTAGE). This increased degree of freedom can lead the model to **learn many pseudo connections** (connections between different nodes at different time steps), which **weakens the weights of real connections** (inter-series and intra-series dependencies). This may **result in overfitting** and **a decrease in predictive performance.**
>
> To test our hypothesis that the channel-mixing strategy **leads to overfitting** and **reduced predictive performance,** we conducted experiments on two datasets. Since the dependency learning components of FourierGNN [1] and UniTST [2] differ from those in GraphSTAGE, we used the model variant **VarC**  in *Figure 7 on Page 9* as a baseline. This variant '**VarC**' changes only the channel-preserving strategy of the original GraphSTAGE to a channel-mixing strategy while keeping the dependency learning components consistent.
>
> To demonstrate the existence of overfitting in the channel-mixing strategy, we set the training epochs to 30 and used a fixed learning rate of 1e-3. We tested on the ECL and PEMS03 datasets with input length equal to prediction length (24 time steps). All experiments have been conducted five times and hyperparameters remained the same. The results are shown in the table below:
>
> | **Method**      | Channel-Preserving Model: Orig in Fig. 7 (GraphSTAGE) |           |          | Channel-Mixing Model: VarC in Fig. 7 |           |          |
> | --------------- | ----------------------------------------------------- | --------- | -------- | ------------------------------------ | --------- | -------- |
> | Loss            | Train MSE                                             | Valid MSE | Test MSE | Train MSE                            | Valid MSE | Test MSE |
> | ECL  (24→24)    | 0.110                                                 | 0.105     | 0.122    | 0.106                                | 0.103     | 0.130    |
> | PEMS03  (24→24) | 0.066                                                 | 0.069     | 0.082    | 0.056                                | 0.065     | 0.091    |
>
> As shown, the channel-mixing model VarC has lower training MSE but higher test MSE. The results indicate that **VarC (channel-mixing model) exhibits overfitting** compared to Orig (channel-preserving model).
>
> ---
>
> To further demonstrate the decrease in predictive performance on the test set caused by the channel-mixing strategy, we refer to the results provided in *Table 4 on Page 10* of our paper. For your convenience, we summarize the relevant (averaged) results below:
>
> | **Method** | Channel-Preserving Model: Orig in Fig. 7 (GraphSTAGE) |           | Channel-Mixing Model: VarC in Fig. 7 |         |
> | ---------- | ----------------------------------------------------- | --------- | ------------------------------------ | ------- |
> | **Metric** | **MSE**                                               | **MAE**   | **MSE**                              | **MAE** |
> | **ECL**    | **0.166**                                             | **0.263** | 0.192                                | 0.284   |
> | **ETTm1**  | 0.391                                                 | **0.394** | **0.389**                            | 0.400   |
>
> The channel-preserving model Orig outperforms the channel-mixing variant VarC, **supporting our hypothesis that channel-mixing leads to decreased predictive performance.**
>
> ---
>
> [1] *FourierGNN: Rethinking multivariate time series forecasting from a pure graph perspective.* NeurIPS, 2024
>
> [2] *UniTST: Effectively Modeling Inter-Series and Intra-Series Dependencies for Multivariate Time Series Forecasting.* arXiv

---

> ### Author Response · Authors · 2024-11-21
>
> # Responses to Reviewer yAwh (2/3)
>
> > **W2(1/2): I have doubts about the effectiveness of purely using GNNs for time series analysis. The performance of GNNs heavily depends on the rationality of the graph structure. However, in certain datasets or tasks, constructing a reasonable graph structure is itself a challenge. If the graph structure fails to accurately capture the dependencies in the data, even the most advanced GNNs may struggle to achieve optimal performance.**
>
> Thanks for pointing out this. Yes, the rationality of the graph structure is crucial for the effectiveness of GNNs in time series analysis. In our proposed GraphSTAGE model, we **have minimized the risk that the graph structure fails to accurately capture the dependencies in the data.**
>
> If we had used pre-defined graphs—as employed by models like DCRNN [1] and STGCN [2]—there could indeed be concerns about the rationality of the graph structure. Pre-defined graphs have been argued in many studies to be biased, incorrect, or even unavailable in many cases [3-6].
>
> Numerous studies have demonstrated that learning-based graphs can effectively enable GNNs to capture complex dependencies in time series forecasting tasks [3-6]. Therefore, our GraphSTAGE model **utilizes a learning-based graph structure ($A_T$ and $A_S$)** to allow the model adaptively learn temporal and spatial dependencies.
>
> ---
>
> [1] *Diffusion Convolutional Recurrent Neural Network: Data-Driven Traffic Forecasting.* ICLR, 2018
>
> [2] *STGCN: a spatial-temporal aware graph learning method for POI recommendation.* ICDM, 2020
>
> [3] *Graph wavenet for deep spatial-temporal graph modeling.* IJCAI, 2019
>
> [4] *Connecting the dots: Multivariate time series forecasting with graph neural networks.* KDD, 2020
>
> [5] *Pre-training enhanced spatial-temporal graph neural network for multivariate time series forecasting.* KDD, 2022
>
> [6] *Dynamic graph convolutional recurrent network for traffic prediction: Benchmark and solution.* Transactions on Knowledge Discovery from Data, 2023

---

> ### Author Response · Authors · 2024-11-21
>
> > **W2(2/2): Additionally, the representational capacity of graphs may be limited in high-noise or irregular data, leading to a decline in the model's predictive capabilities. I hope the authors can provide a more comprehensive explanation.**
>
> Thank you for pointing this out. Yes, we agree that high-noise or irregular data can pose significant challenges for predictive models. However, we believe that the decline in predictive capabilities on high-noise or irregular datasets is primarily due to **the quality of the data** rather than the representational capacity of graphs.
>
> For example, the Exchange dataset used in our experiments is reported to be a high-noise dataset with unclear periodicity [1], where exchange rates are influenced by many unpredictable factors such as economic policies and geopolitical events.
>
> To provide a **more clear and intuitive evaluation**, we followed the recent benchmark BasicTS+ [1] and used the Mean Absolute Percentage Error (MAPE) metric. **MAPE** offers insight into how far predictions deviate from actual values relative to the actual values themselves. The formula for MAPE is:
>
> $\text{MAPE} = \frac{1}{n} \sum_{i=1}^{n} \left| \frac{y_i - \hat{y}_i}{y_i} \right| \times 100$
>
>  The performance of GraphSTAGE and the recent SOTA iTransformer on Exchange dataset are listed as follows:
>
> | **Method**                     | GraphSTAGE |       |            | iTransformer |       |            |
> | ------------------------------ | ---------- | ----- | ---------- | ------------ | ----- | ---------- |
> | Metrics                        | MSE        | MAE   | **MAPE**   | MSE          | MAE   | **MAPE**   |
> | Exchange  (96$\rightarrow$96)  | 0.084      | 0.203 | **121.2**% | 0.086        | 0.206 | **129.3**% |
> | Exchange  (96$\rightarrow$192) | 0.186      | 0.306 | **202.9**% | 0.177        | 0.299 | **193.9**% |
> | Exchange  (96$\rightarrow$336) | 0.339      | 0.420 | **319.5**% | 0.331        | 0.417 | **317.3**% |
> | Exchange  (96$\rightarrow$720) | 0.898      | 0.710 | **656.2**% | 0.847        | 0.691 | **613.2**% |
>
> As shown in the above table, even the recent SOTA model (iTransformer) exhibits extremely high MAPE values on the Exchange dataset, in contrast to the seemingly low MAE and MSE values. This indicates that **the prediction of** **high-noise data is actually quite challenging and represents a bottleneck for all deep network methods** [2].
>
> However, our comprehensive forecasting results (*Table 1 on Page 7* of the revised paper) demonstrate that GraphSTAGE significantly outperforms recent SOTA models on datasets with evident periodicity. For example, on the four subsets of the PEMS datasets, GraphSTAGE achieves an average improvement of 14.3%. Conversely, for datasets with unclear periodicity (e.g., ETT, Exchange), the performance gaps among models are comparatively smaller.
>
> **Regarding irregular data,** where different nodes have distinct original scales or sampling rates, this **is an important direction for our future research.** In this work, we focus exclusively on regular data, where the scales and sampling rates of all nodes are fixed.
>
> ---
>
> [1] *Exploring progress in multivariate time series forecasting: Comprehensive benchmarking and heterogeneity analysis.* IEEE Transactions on Knowledge and Data Engineering, 2024
>
> [2] *iTransformer: Inverted Transformers Are Effective for Time Series Forecasting.* ICLR, 2024

---

> ### Author Response · Authors · 2024-11-21
>
> # Responses to Reviewer yAwh (3/3)
>
> > **W3: FourierGNN also performs time series prediction, so why didn't the authors conduct a comprehensive comparison?**
>
> Thank you for pointing this out. There are two main reasons why we did not include FourierGNN in our initial submission:
>
> 1. **Different Focus of Tasks:** FourierGNN and GraphSTAGE focus on different forecasting tasks. FourierGNN primarily targets spatio-temporal forecasting tasks, with experiments concentrated on **short-term** spatio-temporal forecasting where the maximum prediction length is 12. In contrast, GraphSTAGE focuses on **long-term** time series forecasting, with prediction lengths extending up to 720.
> 2. **Performance:** Existing research suggests that FourierGNN may **not serve as a strong baseline** [1].
>
> To address the your concerns, we have **added** **four graph-based models as baseline methods** for comparison: FourierGNN [2], CrossGNN [3], StemGNN [4], and MTGNN [5]. Following the experimental setup of TimesNet [6], we use a fixed lookback length $T = 96$ and set the prediction lengths $K \in \{96, 192, 336, 720\} $.
>
> We reproduced the results for FourierGNN and StemGNN, running each experiment five times. For CrossGNN and MTGNN, we collected the reported results from [3]. The detailed results are presented in *Table 9 on Page 18* of the revised paper. For your convenience, a summary of the relevant (averaged) results is provided below:
>
> | **Model**          | GraphSTAGE          |                     | FourierGNN[2]   |                 | CrossGNN[3] |       | StemGNN[4]      |                 | MTGNN[5] |       |
> | ------------------ | ------------------- | ------------------- | --------------- | --------------- | ----------- | ----- | --------------- | --------------- | -------- | ----- |
> | Metric             | MSE                 | MAE                 | MSE             | MAE             | MSE         | MAE   | MSE             | MAE             | MSE      | MAE   |
> | ECL (321 Nodes)    | **0.166**$\pm$0.003 | **0.263**$\pm$0.002 | 0.221$\pm$0.001 | 0.318$\pm$0.001 | 0.201       | 0.300 | 0.215$\pm$0.002 | 0.316$\pm$0.002 | 0.251    | 0.347 |
> | ETTm1 (7 Nodes)    | **0.391**$\pm$0.003 | **0.394**$\pm$0.002 | 0.453$\pm$0.002 | 0.448$\pm$0.003 | 0.393       | 0.404 | 0.550$\pm$0.004 | 0.537$\pm$0.005 | 0.469    | 0.446 |
> | ETTh2 (7 Nodes)    | **0.387**$\pm$0.005 | **0.407**$\pm$0.004 | 0.543$\pm$0.006 | 0.517$\pm$0.004 | 0.393       | 0.418 | 1.158$\pm$0.010 | 0.812$\pm$0.011 | 0.465    | 0.509 |
> | Weather (21 Nodes) | **0.243**$\pm$0.001 | **0.274**$\pm$0.001 | 0.257$\pm$0.002 | 0.305$\pm$0.003 | 0.247       | 0.289 | 0.289$\pm$0.005 | 0.342$\pm$0.007 | 0.314    | 0.355 |
>
> The above results show that GraphSTAGE consistently achieved the top-1 rank in terms of MSE and MAE across four datasets. Although CrossGNN exhibits performance comparable to our GraphSTAGE model on smaller-scale datasets ( e.g., ETT and Weather), GraphSTAGE notably outperforms CrossGNN on large-scale datasets (e.g., ECL with 321 nodes). These results further validate the effectiveness of GraphSTAGE in capturing both intra-series and inter-series dependencies, leading to superior forecasting accuracy across datasets of varying scales.
>
> ---
>
> [1] *MambaTS: Improved Selective State Space Models for Long-term Time Series Forecasting.* arXiv
>
> [2] *FourierGNN: Rethinking multivariate time series forecasting from a pure graph perspective.* NeurIPS, 2024
>
> [3] *Crossgnn: Confronting noisy multivariate time series via cross interaction refinement*. NeurIPS, 2023
>
> [4] *Spectral temporal graph neural network for multivariate time-series forecasting.* NeurIPS, 2020
>
> [5] *Connecting the dots: Multivariate time series forecasting with graph neural networks.* KDD, 2020
>
> [6] *TimesNet: Temporal 2D-Variation Modeling for General Time Series Analysis.* ICLR, 2023

---

> ### Author Response · Authors · 2024-11-24
> **Request of Reviewer's attention and feedback**
>
> Thank you for your valuable and constructive feedback, which has inspired further improvements to our paper. As a gentle reminder, it has been more than 2 days since we submitted our rebuttal. We would like to know whether our response has addressed your concerns.
>
> Following your comments and suggestions, we have **answered your concerns** and **made the following revisions**:
>
> - To address your concerns, we have clarified **the noise and interference caused by channel-mixing methods** and **validated the existence and impact of this noise through experiments.**
> - Your concern regarding the **representational capacity of graphs under high-noise data** is highly valuable and insightful. We addressed this concern through experiments on Exchange dataset.
> - We have expanded our experiments to **include four graph-based baselines** for multivariate time series forecasting tasks: (1) FourierGNN, (2) CrossGNN, (3) StemGNN, and (4) MTGNN. For complete results, refer to *Table 9 in Appendix D on Page 18*.
>
> We sincerely thank you for your insightful review. We look forward to your feedback and are prepared to address any further questions or concerns you may have.

---

> ### Author Response · Authors · 2024-11-25
> **Request of Reviewer's attention and feedback**
>
> Dear Reviewer yAwh,
>
> We sincerely thank you for your valuable and constructive feedback. Since the End of author/reviewer discussions is coming soon, may we know if our response addresses your main concerns? If so, we kindly ask for your reconsideration of the score. Should you have any further advice on the revised paper or our rebuttal, please let us know and we will be more than happy to engage in more discussion and paper improvements.
>
> Thank you so much for devoting time to improving our paper!

---

> > ### Author Response · Authors · 2024-12-03
> >
> > Thank you for your positive feedback and for taking the time to carefully review our responses. We deeply appreciate your thoughtful evaluation and support!

---

### Author Response · Authors · 2024-11-21
**Global Response to All Reviewers**

We sincerely thank all the reviewers for their valuable time and detailed feedback, and we appreciate that almost all reviewers recognized the soundness and presentation of our work. We have carefully revised the paper according to the comments, and the edits have been highlighted in **ORANGE**. We also provide a detailed response to each comment below.

Here we highlight our major revisions, and respond to each reviewer below. We hope our responses can properly address your concerns.

- In response to the feedback from **Reviewer** **yAwh**, **9agk**, **7NU8**, and **RvJQ**, we have conducted additional experiments and revised the paper accordingly, along with the relevant discussions.

  1. We have expanded our experiments to include additional four graph-based baseline methods for multivariate time series forecasting tasks: (1) FourierGNN, (2) CrossGNN, (3) StemGNN, and (4) MTGNN. For complete results, refer to *Table 9 in Appendix D on Page 18*.

  2. We have added the standard deviations of our model across all tasks to highlight the model’s strong robustness. For complete results, refer to *Table 10 and Table 11 in Appendix F on Page 22*.

  3. Additionally, we provided the pseudo-code of our model to enhance the clarity in methodological presentation. For detailed results, please see *Algorithm 1 in Appendix A on Page 15*.
- Following feedback from **Reviewer 7NU8**, we have updated the related works to more clearly define the scope and background of this study. We have revised statements that could be slightly misleading, such as “improvements up to 20.8%.”  We have also revised the colors in Figure 15 and have updated the relevant text in the revised paper to make the interpretability more clearly.

We thank the reviewers again and look forward to any further suggestions or discussion.

---

### Meta-Review · Area_Chair_F1CZ · 2024-12-20

**Metareview:**

This paper proposes a graph neural network model, called GraphSTAGE, for multivariate time series forecasting. It aims to decouple the learning of intra-series and inter-series dependencies. Unlike channel-mixing methods, GraphSTAGE is a channel-preserving method that can potentially alleviate the so-called noise and interference caused by mixing channels.

Major strengths:
- Learning different types of dependencies effectively is an important topic in multivariate time series forecasting.
- Extensive experiments are presented.

Major weaknesses:
- There is a lack of technical justification for the proposed method, making it appear a bit ad hoc.
- Although using learning-based graphs can alleviate some problems of using pre-defined graphs, it incurs additional computational overhead during run-time. The efficiency-accuracy tradeoff needs to be studied more comprehensively.

We appreciate the authors for responding to the comments and suggestions of the reviewers in their reviews and conducting additional experiments to address some of the issues raised. Although this work holds promise for publication in a respectable venue such as ICLR, it would have a higher chance to receive stronger votes from reviewers for acceptance if the weaknesses above and some others mentioned by the reviewers could be addressed more thoroughly, or else the important topic of learning different types of dependencies in multivariate time series forecasting would remain unresolved. The authors are encouraged to improve their paper for future submission by considering the comments and suggestions of the reviewers.

**Additional Comments On Reviewer Discussion:**

The authors responded to the reviews by engaging in discussions with the reviewers and providing additional experiment results. Nevertheless, all reviewers still feel that this is a borderline paper. The only (minor) difference is which side of the borderline. It is not suitable for ICLR to accept a paper still with doubts that need to be sorted out. Addressing the outstanding issues, including the major weaknesses listed above, will make this paper more ready for publication.

---

### Decision · Program_Chairs · 2025-01-22

Reject